# Segment Anything for Microscopy

Anwai Archit [1], Luca Freckmann[1], Sushmita Nair[1], Nabeel Khalid [2,3], Paul Hilt[13], Vikas Rajashekar[2], Marei Freitag [1], Carolin Teuber [1], Melanie Spitzner[4], Constanza Tapia Contreras[4], Genevieve Buckley [5], Sebastian von Haaren[6], Sagnik Gupta[1], Marian Grade[4,7], Matthias Wirth[4,7,8,9,10], Günter Schneider[4,7,11], Andreas Dengel[2,3], Sheraz Ahmed[2] & Constantin Pape [1,12] ✉

Accurate segmentation of objects in microscopy images remains a bottleneck for many researchers despite the number of tools developed for this purpose. Here, we present Segment Anything for Microscopy (μSAM), a tool for segmentation and tracking in multidimensional microscopy data. It is based on Segment Anything, a vision foundation model for image segmentation. We extend it by fine-tuning generalist models for light and electron microscopy that clearly improve segmentation quality for a wide range of imaging conditions. We also implement interactive and automatic segmentation in a napari plugin that can speed up diverse segmentation tasks and provides a unified solution for microscopy annotation across different microscopy modalities. Our work constitutes the application of vision foundation models in microscopy, laying the groundwork for solving image analysis tasks in this domain with a small set of powerful deep learning models.

Identifying objects in microscopy images, such as cells and nuclei in light microscopy (LM) or cells and organelles in electron microscopy (EM) is one of the key tasks in image analysis for biology. The large variety of modalities and different dimensionalities (two or three dimensions, time) make these identification tasks challenging and so far require different approaches. The state-of-the-art methods are deep learning based and have in the past years dramatically improved cell and nucleus segmentation in LM[1–3], cell, neuron and organelle segmentation in EM[4–7] and cell tracking in LM[8,9]. Most of these methods provide pretrained models and yield high-quality results for new data similar to their training data. However, due to limited generalization capabilities of the underlying deep learning approaches, quality degrades for data dissimilar to the original training data and they can only be improved by retraining. Generating annotations for retraining relies on manual work

and is time consuming. Some approaches for semiautomatic annotation based on manual correction of initial segmentation results exist[1]. These are still time consuming if the initial results are of low quality. Furthermore, a unified method that addresses diverse segmentation tasks in different modalities like LM and EM is missing.

Vision foundation models have recently been introduced for image analysis tasks in natural images, echoing developments in natural language processing[10–12]. These models are based on vision transformers[13] and are trained on very large datasets. They can be used as a flexible backbone for different analysis tasks. Among the first successful vision foundation models was CLIP[10], which combines images and language, and underlies many generative image models[14]. More recently, foundation models targeting segmentation have been introduced[11,12]. Among them Segment Anything Model[11] (SAM), which was trained on a large

[1]Georg-August-University Göttingen, Institute of Computer Science, Goettingen, Germany. [2]German Research Center for Artificial Intelligence, Kaiserslautern, Germany. [3]RPTU Kaiserslautern-Landau, Kaiserslautern, Germany. [4]Department of General, Visceral and Pediatric Surgery, University Medical Center Göttingen, Göttingen, Germany. [5]Ramaciotti Centre for Cryo-Electron Microscopy, Monash University, Melbourne, Victoria, Australia. [6]Georg-August-University Göttingen, Campus Institute Data Science, Goettingen, Germany. [7]CCC-N (Comprehensive Cancer Center Lower Saxony), Göttingen, Germany. [8]Department of Hematology, Oncology and Cancer Immunology (Campus Benjamin Franklin), Charité Universitätsmedizin, Berlin, Germany. [9]German Cancer Consortium (DKTK), Heidelberg, Germany. [10]Max Delbrück Center, Berlin, Germany. [11]Clinical Research Unit 5002, KFO5002, University Medical Center, Göttingen, Germany. [12]Cluster of Excellence 'Multiscale Bioimaging: from Molecular Machines to Networks of Excitable Cells' (MBExC), Georg-August-University Göttingen, Göttingen, Germany. [13]Independent Researcher: Paul Hilt. ✉e-mail: constantin.pape@informatik.uni-goettingen.de

**a**

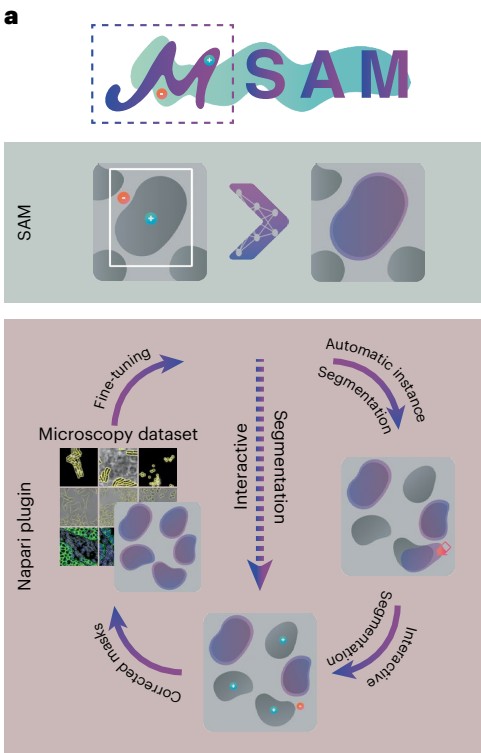

**b**

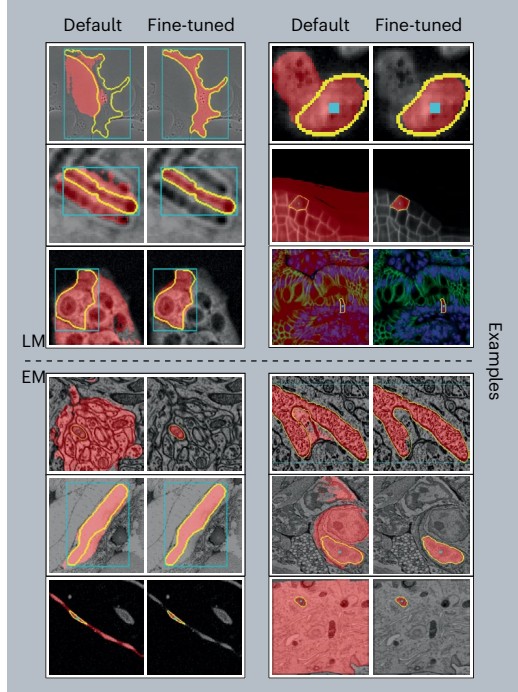

**Fig. 1 | Overview of µSAM. a,** We provide a napari plugin for segmenting multidimensional microscopy data. This tool uses SAM, including our improved models for LM and EM (see **b**). It supports automatic and interactive segmentation as well as model retraining on user data. The drawing sketches a complete workflow based on automatic segmentation, correction of the segmentation masks through interactive segmentation and model retraining

based on the obtained annotations. Individual parts of this workflow can also be used on their own, for example, only interactive segmentation can be used as indicated by the dashed line. **b,** Improvement of segmentation quality due to our improved models for LM (top) and EM (bottom). Blue boxes or blue points show the user input, yellow outlines show the true object and red overlay depicts the model prediction.

labeled dataset and achieves impressive interactive segmentation performance for a wide range of image domains. The application of such foundation models in microscopy has so far been limited, but their potential in this domain has already been identified[15].

Here, we introduce Segment Anything for Microscopy, called µSAM in the following, that improves and extends SAM for microscopy data. Our main contributions are:

- A training procedure to fine-tune SAM, including a new decoder that provides improved instance segmentation results.
- Improved models for LM and EM segmentation that perform considerably better than the default SAM models in their respective domains.
- A tool for interactive and automatic data annotation, provided as a napari[16] plugin. This tool can use the default SAM models, our LM and EM models or custom models fine-tuned by users.

Figure 1 shows a high-level overview of µSAM and examples for improved segmentation results. Prior work has already investigated SAM for biomedical applications, for example, in medical imaging[17], histopathology[18] and neuroimaging[19]. However, these studies were limited to the default SAM and did not implement retraining for their respective domains, which is crucial according to our findings. Retraining SAM for other domains has been investigated for a narrow interactive segmentation task in medical image data[20]. Using SAM as the basis for automatic segmentation has been investigated for histopathology[21] and using it for cell segmentations has been investigated based on prior object detection[22]. None of this prior work combines retraining of the full interactive segmentation capabilities with improved automatic segmentation in a single model as in our contribution.

Compared to established segmentation and tracking tools, µSAM is more versatile because its pretrained models cover both LM and EM, covering a wide range of segmentation tasks. It supports two-dimensional (2D) and volumetric segmentation as well as tracking in the same tool. It combines interactive and automatic segmentation using the same underlying model. As a result, both aspects of the model are improved during fine-tuning, which can massively speed up data annotation. In contrast, the in-the-loop training mode of CellPose 2 (ref. 23), which has pioneered integrated data annotation and training, relies on manual pixel-level correction. In summary, our tool's main distinguishing features are its applicability to diverse segmentation tasks across different modalities and dimensionalities and its fast annotation speed thanks to its interactive segmentation capability. We demonstrate these aspects in three user studies where we find competitive performance with CellPose[23] for cell segmentation, clearly improved performance compared to ilastik carving[24] for volumetric segmentation and compare to TrackMate[9] for tracking. Overall, our contribution shows the promise of vision foundation models to unify image analysis solutions in bioimaging. Our tool is available at https://github.com/computational-cell-analytics/micro-sam/.

## Results

We compare the default SAM with models that we fine-tune for different microscopy segmentation tasks. First, we study interactive and automatic segmentation on the LIVECell[25] dataset. Then, we train and evaluate generalist models, encompassing training on multiple datasets, for cell and nucleus segmentation in LM as well as for mitochondrion and nucleus segmentation in EM. In the following, we refer to the original models provided by Kirilov et al.[11] as 'default' models, models that we have fine-tuned on a single dataset as 'specialist' models, and

models we have fine-tuned on multiple datasets as 'generalist' models. Note that training a single model that consistently improves across different microscopy modalities is not feasible given the current SAM architecture (see 'EM' sections and 'Discussion' for details). Hence, we train separate generalist models for LM and EM. We further investigate fine-tuning SAM in resource-constrained settings. Then, we introduce our user-friendly tool, implemented as a napari[16] plugin, for interactive and automatic data annotation for (volumetric) segmentation and tracking. We compare it to established tools in three user studies for cell segmentation in LM, nucleus segmentation in EM and nucleus tracking in LM.

### Fine-tuning SAM improves cell segmentation

In ref. [11], SAM is introduced as a model for interactive segmentation: it predicts an object mask based on point, box or mask annotations. The point annotations can be positive (part of the object) or negative (not part of the object). The model was trained on a very large dataset of natural images with object annotations. The authors also introduce a method for automatic instance segmentation called automatic mask generation (AMG) based on covering an image with a grid of points, using all of them as point annotations for SAM and filtering out unlikely or overlapping masks. They evaluate interactive and automatic segmentation on a wide range of tasks, including an LM dataset[26]. See the Methods for an overview of the SAM functionality. The original microscopy experiment and our evaluation of default SAM show a remarkable generalization to microscopy, despite the fact that the original training set contains predominantly natural images. However, we noticed several shortcomings of the models for microscopy. For example, SAM segments clusters of cells as a single object as seen in Fig. 1b. To improve SAM for application to our domain, we implement an iterative training scheme to enable fine-tuning on new datasets. This approach reimplements the original training method, which has so far not been made open source. Furthermore, we add a new decoder to the model that predicts foreground as well as distances to object centers and boundaries to then obtain an automatic instance segmentation via post-processing. We refer to this approach as AIS. The additional decoder can be trained in conjunction with the rest of SAM. Both AIS and the training methodology are explained in more detail in the Methods ('AIS' and 'Training').

We investigate our fine-tuning method on LIVECell[25], one of the largest publicly available datasets for cell segmentation. Figure 2a shows the mean segmentation accuracy[27] (higher is better; see the Methods for details) for the default and LIVECell specialist model, using a separate test set for evaluation. Here, we evaluate interactive segmentation by simulating user annotations based on segmentation ground truth. We derive either a box annotation (red bars) or a positive point annotation (green bars) from the ground truth, corresponding to iteration 0 in the figure. Then we sample both a positive and a negative point from incorrect areas in the prediction, the positive point where the prediction is missing and the negative point where it should not be, and then rerun the model with the additional annotations. This process is repeated seven times (iterations 1–7) and in each iteration the newly sampled points are used as additional point inputs.

We also compare automated segmentation via AMG and AIS (only available after fine-tuning) and provide the results from a CellPose model trained on LIVECell for reference. We use SAM based on a large vision transformer (ViT-L) and train it for 250,000 iterations on the training split of LIVECell. We found that ViT-L provides the best trade-off between runtime and quality; see Fig. 5a for a comparison of runtimes with different model sizes and Extended Data Fig. 1 for an evaluation of segmentation results. The results show a clear improvement due to fine-tuning across all settings. Interactive segmentation with the specialist models is clearly better than any of the automated segmentation results, whereas it only reaches the performance of CellPose after several correction iterations for the default model.

The specialist models also achieve a consistent improvement when provided with more annotations, which is not the case for the default model. This is partly because we do not use the mask prediction as additional model inputs; see Supplementary Fig. 1 for details. Instance segmentation with AMG drastically improves, and segmentation with AIS, which is only available after fine-tuning, is on par with CellPose.

We investigate different fine-tuning strategies in Fig. 2b, where we fine-tune only parts of the SAM architecture, freezing all other weights. Here, we perform the same evaluation experiments as in Fig. 2a. For a more concise presentation, we only report the interactive segmentation results for a single point prompt ('point'), for a single box prompt ('box'), for the last iteration when starting from a point prompt ('I$_P$', corresponding to the green bar at iteration 7 in Fig. 2a) and for the last iteration when starting from a box prompt ('I$_B$', corresponding to the red bar at iteration 7 in Fig. 2a). The results show that fine-tuning the image encoder has the biggest impact, and fine-tuning the complete model shows the best overall performance. In Fig. 1c, we fine-tune the model with only a subset of the available training data, using the data splits defined in the LIVECell publication. The results show that the majority of improvement is achieved with training data fractions of 2%, 4% and 5%. Overall, the results on LIVECell offer the following conclusions:

1. Fine-tuning SAM clearly improves the segmentation quality for a given dataset.
2. Fine-tuning all parts of the model yields the best results. Consequently, we train the full model in all further experiments.
3. Most of the improvements for a given dataset can be achieved with a rather small fraction of the training set. We investigate this in more detail in Fig. 5b.

### An LM generalist model improves across diverse conditions

Our next goal is to train a generalist model for LM that improves segmentation performance for this modality and can thus serve as a replacement to the default SAM. While the previous experiments have shown that fine-tuning on data from a given image setting improves performance, we have not yet shown that it leads to improved generalization. To train the generalist, we assemble a large and diverse training set based on published datasets, including LIVECell[25], DeepBacs[28], TissueNet[2], NeurIPS CellSeg[29], PlantSeg (Root)[30] and Nucleus DSB[26], using a version of this dataset excluding histopathology images provided by StarDist[3], and eight datasets from the Cell Tracking Challenge[31].

**Fig. 2 | Results on LIVECell. a**, Comparison of the default SAM with our fine-tuned model. The bar plot shows the mean segmentation accuracy for interactive segmentation, starting from a single annotation, either a single positive point (green) or a box (red). We then iteratively add a pair of point annotations, one positive, one negative, derived from prediction errors to simulate interactive annotation. The lines indicate the performance for automated instance segmentation methods—AMG (yellow), AIS (dark green) and CellPose (red)— using a CellPose model trained on LIVECell. Evaluation is performed on the test set defined in the LIVECell publication[25]. **b**, Comparison of partial model fine-tuning. The *x* axis indicates which part(s) of the model are updated during training: the image encoder, the mask decoder and/or the prompt encoder. We evaluate AIS (dark green, striped), AMG (yellow), segmentation from a single point annotation (light green, corresponding to the green bar at iteration 0 in **a**), from iterative point annotations I$_P$ (green, corresponding to the green bar at iteration 7 in **a**), from a box annotation (magenta, corresponding to the red bar at iteration 0 in **a**) and from a box annotation followed by correction with iterative point annotations I$_B$ (red, corresponding to the red bar at iteration 7 in **a**). Training the image encoder has the biggest impact and fine-tuning all model parts yields the best overall results. **c**, Evolution of segmentation quality for increasing size of the training dataset, using the same evaluation and color coding as in **b**. All results in this figure use a model based on ViT-L. Extended Data Fig. 1 explains the model parts and shows results for models of different sizes.

We also train specialist models on five of these individual datasets. Figure 3a compares the segmentation performance for default, specialist and generalist models. In all cases, the evaluation is done on a test split that is not used for training. We see clear improvements of both specialist and generalist models compared to the default model. The generalist model overall performs similar or better than the specialist, except for AIS on LIVECell. We include automatic segmentation results for CellPose as a reference, using specialist models for LiveCell and

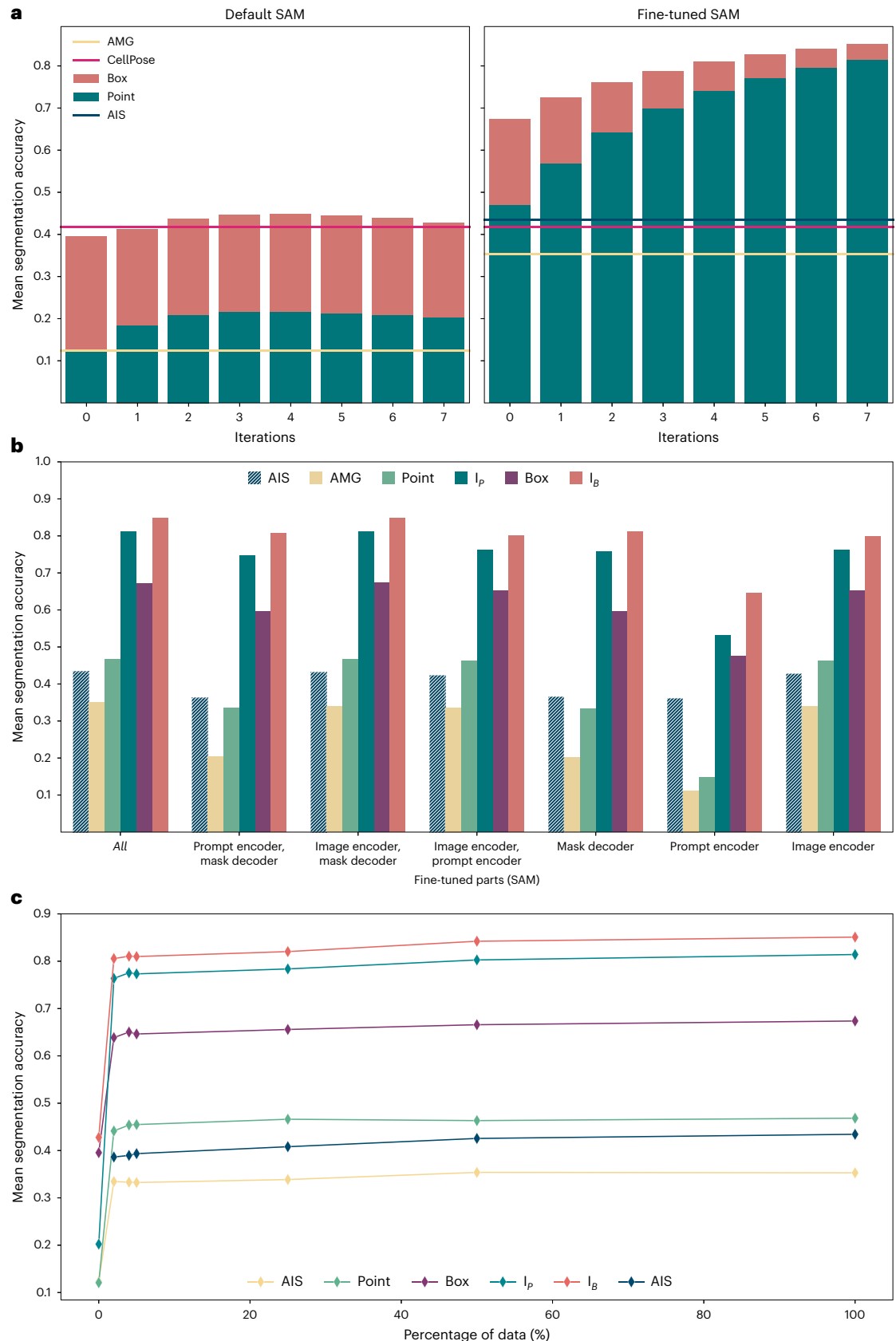

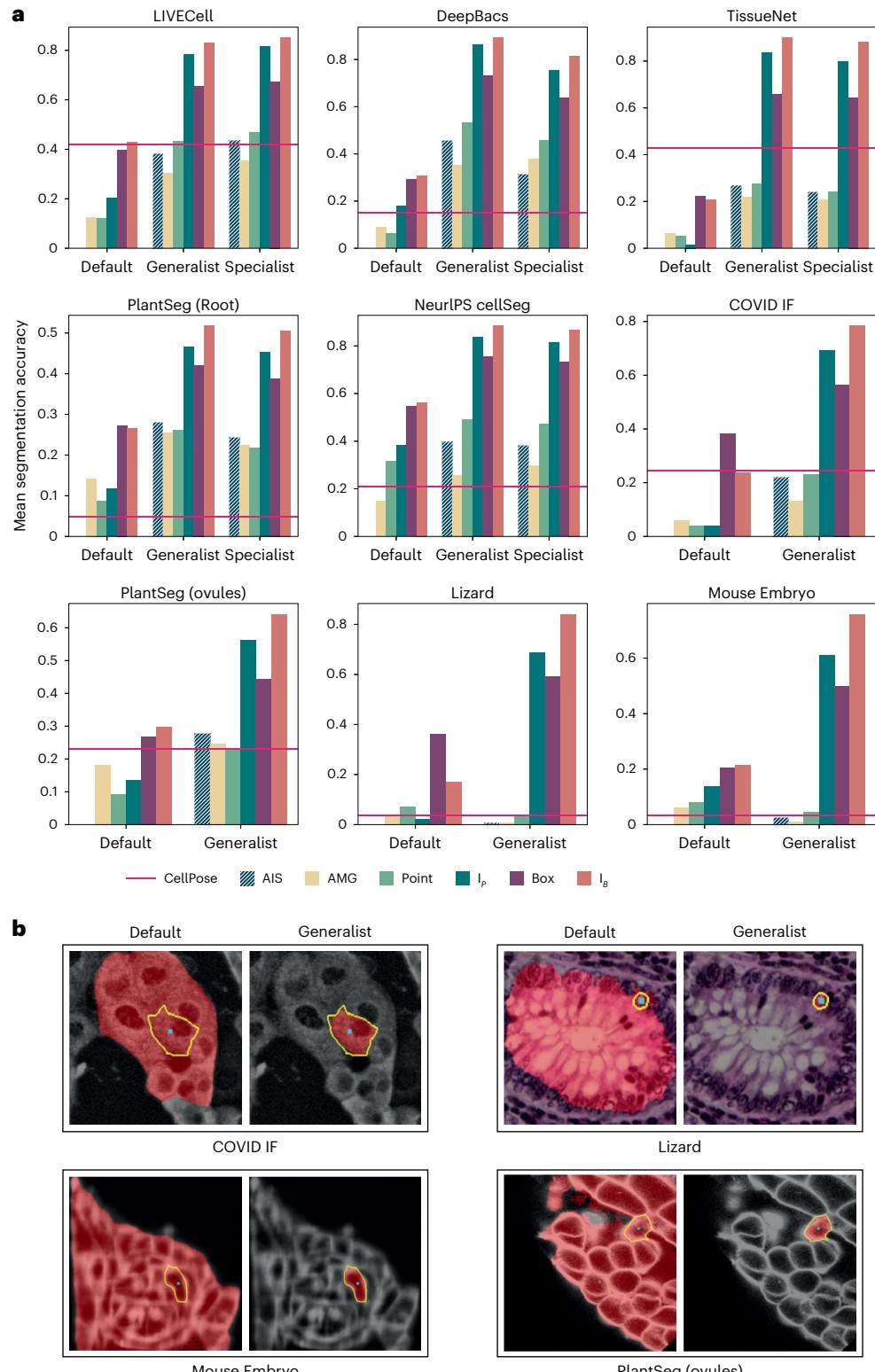

**Fig. 3 | Generalist LM model. a**, Comparison of the default SAM with our generalist and specialist models. We use the same evaluation procedure as in Fig. 2b,c. The red line indicates the performance of CellPose (specialist models for LIVECell and TissueNet, cyto2 model otherwise). Datasets LIVECell, DeepBacs, TissueNet, PlantSeg (root) and NeurIPS CellSeg are part of the training set (evaluated on a separate test split) and datasets COVID IF, PlantSeg (ovules), Lizard and Mouse Embryo contain image settings not directly represented in training. **b**, Qualitative segmentation results with the default SAM and our LM generalist model. The cyan dot indicates the point annotation, the yellow outline highlights the true object and the red overlay represents the model prediction.

TissueNet and the 'cyto2' model otherwise. Automatic segmentation via AIS performs on par or better than CellPose except for TissueNet. We believe that this difference is partly due to the fact that TissueNet contains two channels, which do not map well to the RGB inputs of SAM (Methods). Please note that the comparisons on DeepBacs, PlantSeg (Root) and NeurIPS CellSeg are heavily biased in our favor, because

our model was trained on the training splits of the respective datasets, unlike the CellPose cyto2 model.

To study whether the generalist model improves generalization to other microscopy settings, we apply it to datasets that are not directly represented in the training set. We choose the datasets COVID IF[32] containing immunofluorescence data, PlantSeg (ovules)[30] containing plant cells imaged with confocal fluorescence microscopy, Lizard[33] containing histopathology images and Mouse Embryo[34] containing mammalian cells imaged with confocal fluorescence microscopy. Results for the default SAM and our generalist model as well as the model cyto2 in CellPose are also shown in Fig. 3a. The improvement of our generalist over the default model is clear for all datasets. AIS is better than AMG in almost all cases and CellPose and AIS show overall comparable segmentation accuracy. For Lizard, which contains a different modality compared to the training data and Mouse Embryo, which represents a particularly difficult problem due to small cell sizes, none of the automated segmentation approaches work well, but interactive segmentation yields good results and is improved by the generalist. Figure 3b shows example comparisons of segmentation with the default model and generalist model. We report the results for ViT-L, trained for 250,000 iterations for our models. Extended Data Fig. 2 shows results for SAM models of different sizes, including results on additional datasets, Extended Data Figs. 3 and 4 show qualitative examples and Supplementary Fig. 2b shows examples for the automatic segmentation results. An overview of the LM datasets is given in Supplementary Table 1.

Overall, these experiments demonstrate that a generalist model for a given domain clearly improves segmentation quality. We provide such a generalist model for LM. It supports both interactive and automatic segmentation, achieving comparable automatic segmentation quality to CellPose, the state-of-the-art for automatic cell segmentation. Note that we do not claim that our model is better than CellPose for automatic segmentation, but that it provides similar quality for most practical settings, while also enabling interactive segmentation.

### Improved mitochondria segmentation in EM

We further investigate training a generalist model for EM. This is more challenging compared to LM, because in EM membrane-bound structures are labeled unspecifically rather than having a specific stain for a cellular component. Consequently, the segmentation tasks in EM are more diverse and structures can have a hierarchical composition, for example, an organelle inside a cellular compartment. This makes training a model for generic EM segmentation more challenging. Hence, we focus on training a model for the segmentation of mitochondria and nuclei, for which large public datasets exist. We make use of the MitoLab[5] and MitoEM[35] datasets for mitochondria and PlatyEM[4] for nuclei. We refer to this model as EM generalist in the following, but want to make clear that it reliably improves EM segmentation only for mitochondria, nuclei and other roundish organelles. Due to the limitation of not being able to provide a unified model for EM, we also refrain from exploring a unified generalist model for both EM and LM.

We compare the default SAM and our EM generalist model on test splits of the training datasets and on additional test datasets: Lucchi[36] containing mitochondria imaged in FIBSEM, two MitoLab[5] test datasets containing mitochondria in volume EM (Fly Brain) and transmission electron microscopy (TEM), UroCell[37] containing mitochondria in FIBSEM and VNC containing mitochondria in serial-section TEM. We also include NucMM (Mouse)[38], which contains nuclei imaged in high-energy X-ray, an imaging modality that shares similarity with EM. See Fig. 4 for quantitative and qualitative results. We see a clear improvement for interactive segmentation due to fine-tuning for all datasets. For automatic mitochondrion segmentation, we also compare to MitoNet[5] and find that its performance is overall comparable to AIS and AMG, with results varying across datasets. The advantage of AIS

over AMG is not as clear as for LM. This is likely because AMG works better for well-separated objects, like mitochondria in EM, compared to densely packed objects, like cells in LM. In practice, AIS is preferable in most cases due to its lower runtime (Fig. 5a). Note that we don't claim that our method is superior to MitoNet for automatic mitochondrion segmentation, but rather that it provides comparable quality while also enabling interactive segmentation. Extended Data Fig. 5 shows results for additional datasets and for different model sizes. Extended Data Figs. 6 and 7 show additional qualitative results, and Supplementary Table 2 lists an overview of the EM datasets.

We also perform experiments for other organelles and structures in EM. We segment cilia and microvilli with our model (see the results for Sponge EM and Platynereis (Cilia) in Extended Data Figs. 5–7) and find that our EM generalist model overall performs better compared to the default SAM. We also study segmentation for endoplasmic reticulum (ER) and neurites (Extended Data Fig. 8). We find that our EM generalist only provides marginal benefits or is detrimental in these cases, which is due to the different morphology of ER compared to mitochondria/nuclei and the fact that the model prefers to segment organelles over the surrounding cellular compartment. We train specialist models for both cases, which clearly improve the performance for the given segmentation task.

Overall, we find that training a model for improved organelle segmentation in EM is feasible, and we provide a generalist model for mitochondrion and nucleus segmentation, which can also improve results for other organelles of similar morphology. Training an even more general EM model should be possible given a suitable dataset, but for training a true generalist model that improves segmentation for both cellular compartments and organelles, a semantically aware model and training procedure is required. However, our fine-tuning methodology can be used to train specialist models for a given EM segmentation task and our annotation tools (see below) can be used for fast data annotation to provide the required training data, making our contribution also valuable for EM segmentation tasks where our EM generalist model does not offer benefits.

### Resource-constrained settings for inference and fine-tuning

One of our main goals is to build a user-friendly tool for interactive and automatic microscopy segmentation. As a preparation, we investigate how SAM can be used in resource-constrained settings, for example, on a user laptop or a regular workstation, for inference and fine-tuning. First, we compare the inference times for all relevant operations: computing image embeddings, inference for one object with a box or point annotation and automatic segmentation via AMG and AIS, for CPU and GPU (Fig. 5a). For Point, Box, AMG and AIS, we measure the runtime excluding the embedding computation. Runtimes are much smaller on the GPU, but interactive segmentation with points or boxes is feasible on the CPU in around 30 ms per object, given precomputed embeddings. We also see a big speedup of AIS compared to AMG. The main advantage of a GPU is strongly decreased runtime for embedding computation and faster automatic segmentation, especially for multidimensional data (see below). We also compare the runtimes for different sizes of the image encoder, including ViT Tiny (ViT-T)[39]. Given the trade-off between runtimes and segmentation accuracy (see Extended Data Figs. 1, 2 and 5 for an extensive comparison of the segmentation quality across model size), we recommend using ViT Base (ViT-B) or ViT-L models. Using ViT Huge (ViT-H) does generally not yield better results but incurs a higher computational cost. If runtime is an issue, ViT-B can be used with only a small penalty on segmentation quality. ViT-T is much faster and yields good results for simple segmentation tasks but has severely degraded quality for others. To provide a comparison to established tools, we have also measured the runtime of CellPose with the same hardware. It takes circa 0.3 s to segment an image with the GPU and 1.5 s with the CPU. Compared to this, the runtime of embedding computation and AIS, which is the relevant

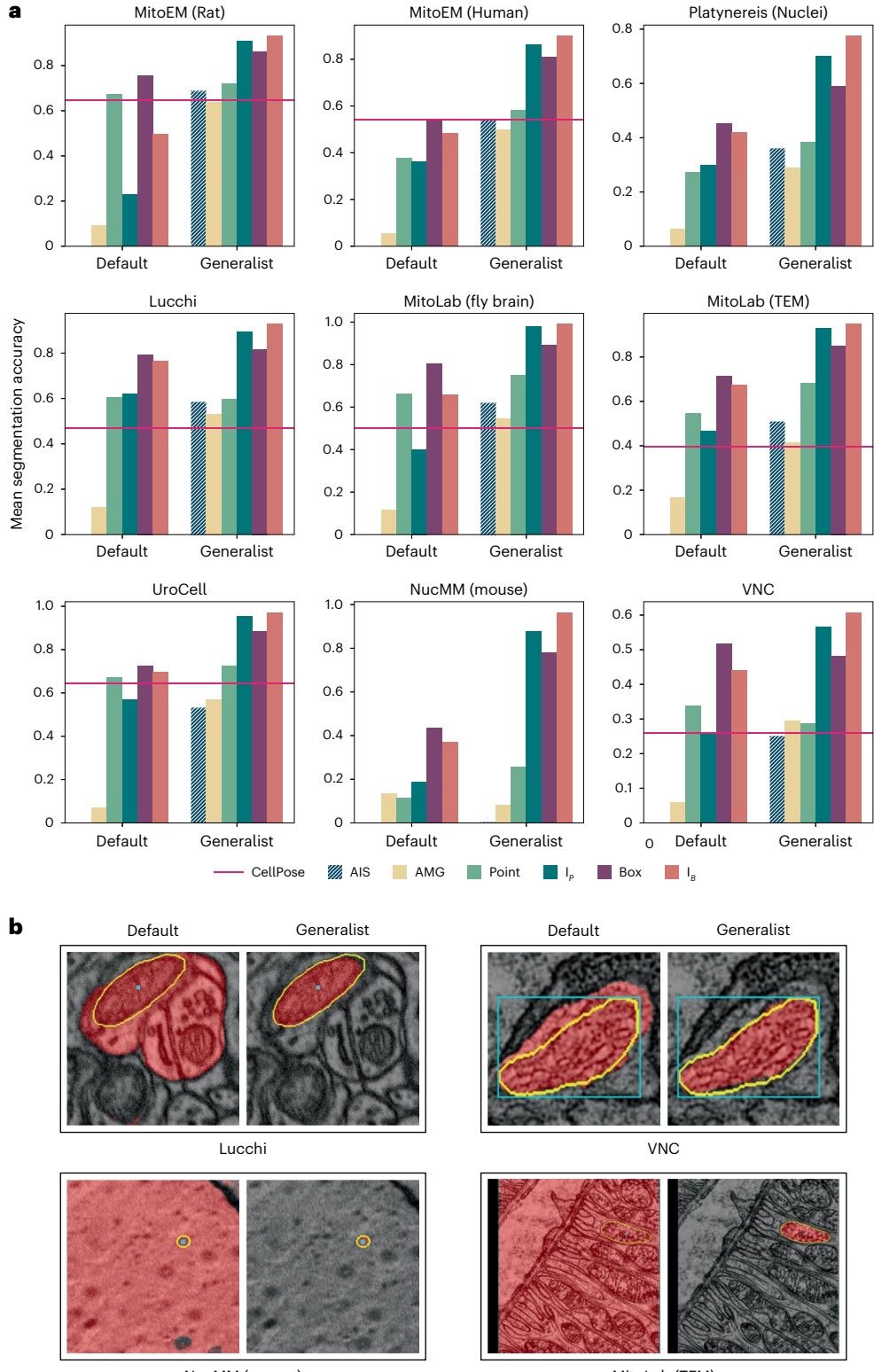

**Fig. 4 | EM model. a**, Comparison of the default SAM and our EM generalist that was trained to improve mitochondrion and nucleus segmentation. Of the nine datasets, MitoEM (rat), MitoEM (human) and Platynereis (nuclei) are part of the training set (evaluation is done on separate test splits), while the others are not. We follow the same evaluation procedure as before. We provide the results of MitoNet (red line) as a reference for automatic mitochondrion segmentation. All experiments are done in 2D. **b**, Qualitative comparisons of segmentation results with default SAM and our EM generalist, using the same color coding as in Fig. 3b.

measure for automatic segmentation, takes 0.2–1.2 s on the GPU and 1.5–7.5 s on the CPU, depending on the model size.

We further investigate model fine-tuning in resource-constrained settings. While our LM or EM generalist models improve quality in many settings, they may not be sufficient for the user's needs or may not match the modality of their data. To enable further improvement for a specific task, we investigate fine-tuning on the COVID IF data (also used in Fig. 3). We study how it behaves for a small number of annotated

images and fine-tuning on the CPU (Fig. 5b) and for other hardware configurations (Supplementary Fig. 3), starting from either the default or the LM generalist model. To enable training with limited resources, we use early stopping and find the best hyperparameters that enable training for the given hardware configuration (Extended Data Fig. 10b,c). We also study parameter-efficient training using LoRA[40], which holds the promise of faster training (see dashed result lines in previously mentioned figures and Supplementary Fig. 4c for an extensive evaluation of training with LoRA). We find that training on only a few images with a CPU is feasible and that it improves the model clearly for a given task. Training with LoRA results in longer training times in most cases because the model needs more iterations to converge. Training on the CPU in this setting took 5.3 h, while training on the GPU took 30 min. The overview of training times for different hardware configurations is given in Extended Data Fig. 10d.

Overall, we find that applying and training SAM in resource-constrained settings is feasible. However, the runtimes for computing image embeddings and training are larger compared to architectures based on convolutional neural networks, especially when using the CPU. We also find that fine-tuning on a few annotated images, which can be quickly generated with our annotation tools (next section), clearly improves results. Starting fine-tuning from our models can provide clear benefits. In cases where our models are worse than default SAM, for example, for neurite segmentation in EM, it will likely be better to start from a default model, so users should choose the model that performs best for their task as the starting point.

### µSAM enables fast data annotation for microscopy

We provide a tool for interactive and automatic data annotation, making use of the models and knowledge described in the previous sections. To make the tool easily available to biologists, we implemented it as a napari[16] plugin. Napari is a Python-based viewer for multidimensional image data that is popular for microscopy image analysis. We provide five different functionalities within our tool: (i) for 2D image segmentation, (ii) for volumetric segmentation, (iii) for tracking in time-series data, (iv) for high-throughput segmentation of multiple images and (v) for fine-tuning. They are implemented as separate plugin widgets. The annotation widgets (i–iv) support interactive segmentation based on user-provided point or box annotations and automatic segmentation based on AIS or AMG (except for the tracking widget (iii)). To enable interactive usage, we implement precomputation and caching of image embeddings, tiled interactive and automatic segmentation and efficient recomputation of the automatic segmentation given parameter changes. We also support interactive segmentation for volumetric data and interactive tracking for time series by projecting masks to adjacent slices or frames and rerunning SAM with the derived annotations. For volumetric data, we implement automated segmentation by running AIS or AMG per slice and merging the results across slices in a post-processing step (Extended Data Fig. 9). The fine-tuning widget (v) allows users to choose the model and training parameters that best fit their hardware and then fine-tune a model on their own data. We also provide the underlying functionality as a Python library so that users with computational

knowledge can implement training scripts and so that developers can build upon our extensions to the original SAM functionality. See the Methods for the details. The Supplementary Videos explain the tool usage and it is documented at https://computational-cell-analytics.github.io/micro-sam/micro_sam.html.

We study our tool for three representative annotation tasks: organoid segmentation in brightfield microscopy, nucleus segmentation in EM and nucleus tracking in fluorescence microscopy and compare them to established software for the respective annotation tasks. Further details about the experimental setup for the user studies can be found in the Methods and Supplementary Information.

### User study 1: Brightfield organoid segmentation

For 2D annotation, we study organoid segmentation in brightfield images. Growing organoids is a common experimental technique for studying tissues, for example, in cancer research. Organoid segmentation enables studying growth and morphology. Here, we use an internal dataset to compare different annotation approaches, comparing our tool with CellPose and manual annotation. The results of the study are summarized in Fig. 6a. In our tool, we compare using the default SAM ('µSAM (default)'), our LM generalist ('µSAM (LM generalist)') and a model fine-tuned on user annotations ('µSAM (fine-tuned)'). For all these models, we first run automatic segmentation, which we then correct using interactive segmentation. We use ViT-B as the image encoder for all models. For CellPose we use the cyto2 model ('CellPose (default)'), in-the-loop training starting from cyto2 ('CellPose (HIL)') and annotation with the model obtained after in-the-loop training ('CellPose (fine-tuned)'). Here, we also first run automatic segmentation and then correct it using manual annotation. These experiments are performed with the CellPose GUI. For each method, we report the average annotation time per object, the quality of the annotations compared against consensus annotations ('mSA Ann.') and the segmentation quality measured on a separate test split of the organoid dataset ('mSA test'). The latter measure evaluates model generalization after fine-tuning. All experiments are done by five different annotators and, we use standard deviations over annotators to compute errors. The image in Fig. 6a shows an automatic segmentation result from the default SAM model and a fine-tuned model. We can derive several observations from the results in Fig. 6a: the default SAM model provides better interactive segmentation results than the LM generalist for this data. This is because interactive segmentation with the generalist yields masks that are too big. This bias was likely introduced by the generalist's training data, which did not include organoid-like data. We plan to address this by extending the generalist's training data in the future. Note that annotation times with the generalist are faster, because it yields a better automatic segmentation. However, due to its better annotation quality, we continue with the default SAM model for the rest of the user study. When comparing the pretrained SAM models with CellPose, we find slightly faster annotation times, but also decreased annotation quality compared to the consensus. Annotation is much faster than manual in all cases. After fine-tuning, annotation time and quality is better with both µSAM and CellPose and is similar for both tools. Finally, a clear difference can be seen in the results for

**Fig. 5 | Inference and training in resource-constrained settings. a**, Runtimes for computing embeddings, running AIS and AMG (per image) and segmenting an object via point or box annotation (per object) on a CPU (Intel Xeon, 16 cores) and GPU (Nvidia RTX5000, 16 GB VRAM). We run AIS, AMG, point and box annotation with precomputed embeddings. We report the average runtime for 10 different images for Embeddings, AIS and AMG, measuring the runtime for each image five times and taking the minimum. For Point and Box, we report the average runtime per object, averaged over the objects in 10 different images. **b**, Improvements due to fine-tuning a ViT-B model when training on 1, 2, 5 or 10 images of the COVID IF dataset on the CPU (same CPU as in **a**). We compare using the default SAM and our LM generalist model as starting points and evaluate the segmentation results on 36 test images (not part of any of the training sets). We use early stopping. Dotted lines indicate results obtained with LoRA[40] using a rank of 4. Otherwise all model parameters are updated, as in previous experiments; we refer to this as full fine-tuning (FFT) in the caption. See Extended Data Fig. 10d for training times of different hardware setups. Note that we use the segmentation accuracy evaluated at an intersection over union (IOU) threshold of 50%, as the metric here, because we found that mean segmentation accuracy was too stringent for the small objects to meaningfully compare improvements. **c**, Qualitative automatic segmentation results before and after fine-tuning on 10 images for the default SAM (comparing AMG before and AIS after fine-tuning) and our LM generalist (comparing AIS before and after fine-tuning).

generalization to the test split: the μSAM models improve consistently, whereas the CellPose models deteriorate, although starting from a better initial result. We are not sure what causes this effect, but we have qualitatively observed that it's because the fine-tuned CellPose

models find fewer organoids in the test set. Overall, we find that there is no clearly better tool for this dataset: CellPose has a better initial segmentation quality, provides similar annotation speed and quality after fine-tuning, but generalizes worse to similar data. We also want to

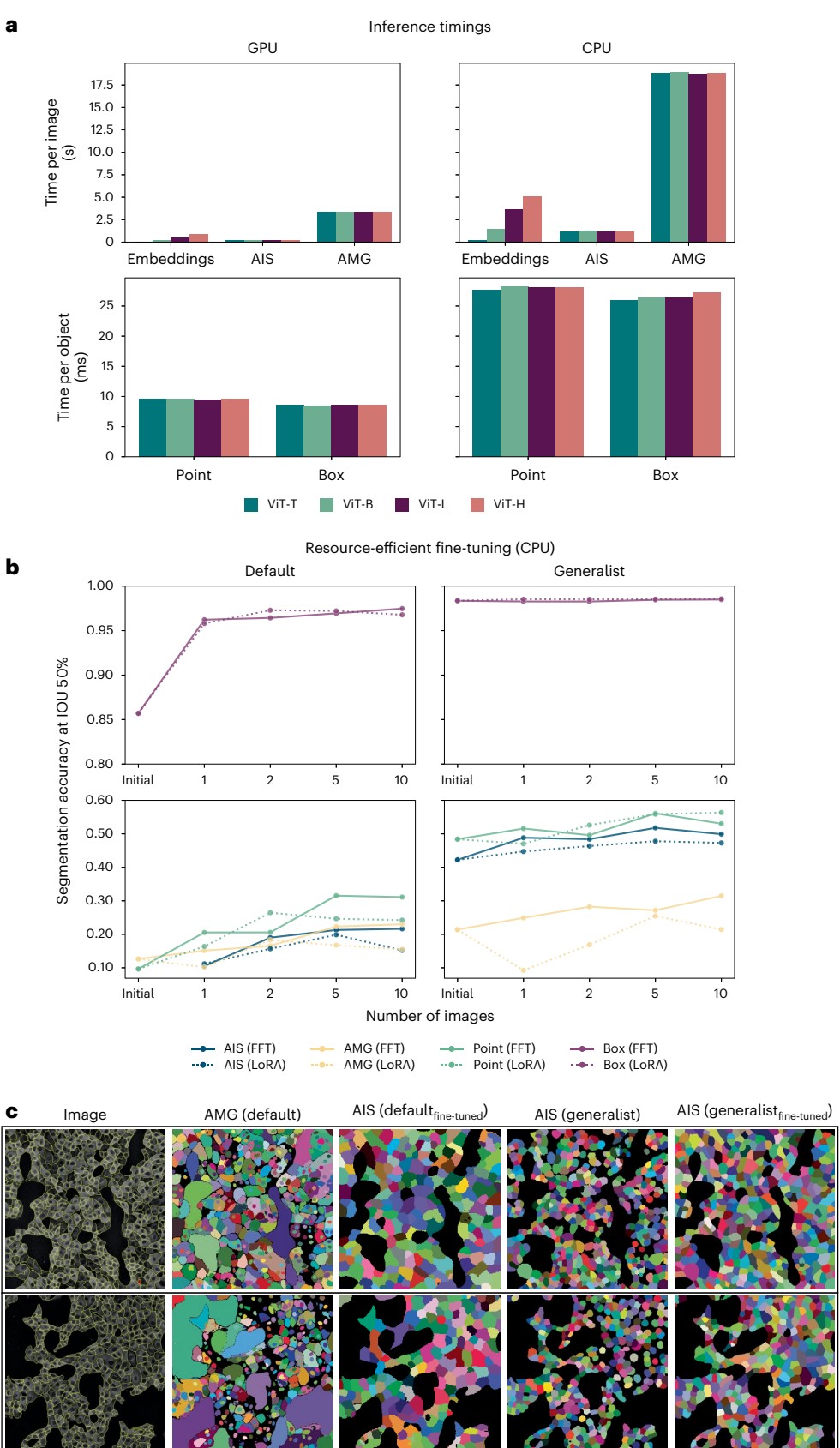

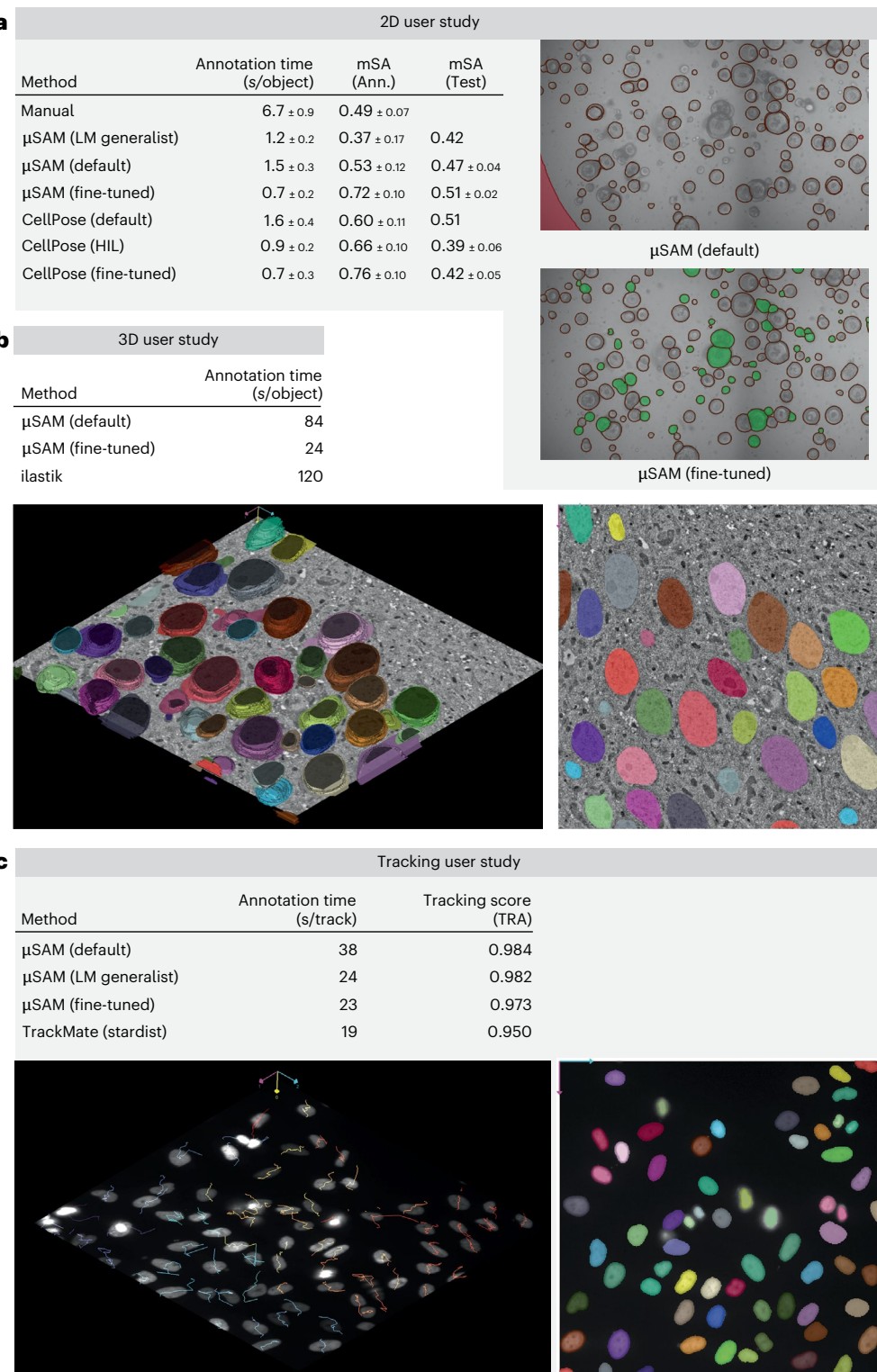

**a** 2D user study

| Method | Annotation time (s/object) | mSA (Ann.) | mSA (Test) |
|---|---|---|---|
| Manual | 6.7 ± 0.9 | 0.49 ± 0.07 | |
| μSAM (LM generalist) | 1.2 ± 0.2 | 0.37 ± 0.17 | 0.42 |
| μSAM (default) | 1.5 ± 0.3 | 0.53 ± 0.12 | 0.47 ± 0.04 |
| μSAM (fine-tuned) | 0.7 ± 0.2 | 0.72 ± 0.10 | 0.51 ± 0.02 |
| CellPose (default) | 1.6 ± 0.4 | 0.60 ± 0.11 | 0.51 |
| CellPose (HIL) | 0.9 ± 0.2 | 0.66 ± 0.10 | 0.39 ± 0.06 |
| CellPose (fine-tuned) | 0.7 ± 0.3 | 0.76 ± 0.10 | 0.42 ± 0.05 |

μSAM (default)

μSAM (fine-tuned)

**b** 3D user study

| Method | Annotation time (s/object) |
|---|---|
| μSAM (default) | 84 |
| μSAM (fine-tuned) | 24 |
| ilastik | 120 |

**c** Tracking user study

| Method | Annotation time (s/track) | Tracking score (TRA) |
|---|---|---|
| μSAM (default) | 38 | 0.984 |
| μSAM (LM generalist) | 24 | 0.982 |
| μSAM (fine-tuned) | 23 | 0.973 |
| TrackMate (stardist) | 19 | 0.950 |

**Fig. 6 | User studies of the μSAM annotation tools. a**, Segmentation of organoids imaged in brightfield microscopy with μSAM, CellPose and manual annotation. We compare different models for μSAM and CellPose; see the text and Methods for details. We report the average annotation time per object, quality of annotations when compared to consensus annotations and segmentation quality evaluated on a separate test dataset. All experiments are done by five annotators and errors correspond to standard deviations over annotator results. The entries 'μSAM (LM generalist)' and 'CellPose (default)' in the 'mSA (test)' column are obtained from evaluating the initial models; the other results in this column are obtained from evaluating models trained on user annotations. The two images on the right compare the automated segmentation result (without correction) obtained from 'μSAM (default)' and 'μSAM (fine-tuned)'. **b**, Segmentation of nuclei in volume

EM. The table compares the average annotation time per object for μSAM, using a default model and a model fine-tuned for this data, with ilastik carving. For the fine-tuned model, we start annotation from an initial 3D segmentation provided by the model; otherwise, we annotate each object interactively. The image below shows the result after correction for the fine-tuned model. **c**, Tracking of nuclei in fluorescence microscopy. The table lists the average annotation time per track for μSAM, using three different models, and TrackMate, as well as the tracking quality, measured by the tracking accuracy score (TRA). For μSAM, each lineage is tracked interactively; 'fine-tuned' is trained specifically for this data. TrackMate provides an automatic tracking result, based on nucleus segmentation from StarDist, which is then corrected. The image below illustrates the tracking annotation obtained with μSAM (fine-tuned).

stress that results are data dependent and will differ for other datasets depending on performance of the initial models.

## User study 2: volume EM nucleus segmentation

For the three-dimensional (3D) annotation tool, we study nucleus segmentation in volume EM, using an internal dataset from the fruit fly larva brain, for which we also have ground-truth annotations for several small blocks. Segmentation of nuclei or other large organelles in volume EM is an important task for analyzing cellular morphology and differentiating cell types based on phenotypic criteria[4]. Here, we compare interactive nucleus segmentation with µSAM and with ilastik carving[24]. Carving uses a seeded graph watershed to segment objects in 3D from user annotations. This method is not based on deep learning, but is still one of the most commonly used approaches for interactive 3D segmentation, for example, Gallusser et al.[41] use it to generate training annotations. In µSAM, we first annotate the data with the default ViT-B model, which worked slightly better than the EM generalist model, likely due to differences in resolution to our training data. In this case, we did not use automatic segmentation since it did not yield good results. We also fine-tune a model on another small block with ground-truth data. For this model, automatic 3D segmentation (based on AIS) yields good results. Figure 6b shows the annotation time per object and an illustration of an annotated block. Annotation with µSAM is faster than ilastik when using the default model and much faster when using a fine-tuned model, for which we can correct automatic segmentation results rather than interactively segmenting every object.

## User study 3: fluorescence microscopy nuclei tracking

We study the tracking annotation tool on a dataset of nuclei imaged in fluorescence microscopy from Schwartz et al.[42], using every third frame to make the task more challenging. We compare annotation via µSAM with the most recent version of TrackMate[9], which has integrated support for deep learning-based segmentation tools, including StarDist[3]. Figure 6c shows the results for four different approaches: interactive segmentation with our tool, using the default SAM, the LM generalist model and a model fine-tuned for this data, as well as TrackMate with StarDist. We report annotation times and quality of the annotations compared to ground truth. Note that our tool and TrackMate work quite differently for tracking: in our tool, each lineage has to be tracked interactively, whereas TrackMate automatically tracks the nuclei based on the segmentation from StarDist, followed by manual correction. Here, we see a clear advantage of the LM generalist model over the default SAM; it tracks the nuclei better during interactive annotation. Fine-tuning of this model on a separate time series does not speed up tracking further. Compared to TrackMate, our method is a bit slower, which is because we currently do not automatically track objects, but yields annotations of higher quality. We aim to implement automatic tracking that can be used as a starting point for correction, based on initial frame-by-frame segmentations from AIS, and expect a major speedup from this extension.

## Discussion

We have introduced a method to fine-tune SAM for microscopy data, used it to provide generalist models for LM and EM and extensively compared these to the default SAM and reference methods for automatic segmentation. We have also implemented a napari plugin for interactive and automatic segmentation. Our quantitative experiments and user studies show that our contribution can speed up data annotation and automatic segmentation for a diverse set of applications. Our contribution also marks the application of vision foundation models in microscopy. We expect future work to build on it and extend the application of such models to further improve object identification tasks and address other image analysis problems.

We compare our method to established tools for segmentation and tracking and show competitive or improved performance. However, we expect that further improvements toward usability and performance can be made by integrating parts of our methods with other tools. For example, our models and interactive segmentation functionality could be integrated with CellPose, MitoNet or other methods for automatic instance segmentation that enable users to fine-tune, combining faster data annotation with more efficient architectures for processing large datasets. To enable such integration, we have developed our annotation tool as a napari plugin so that they can be used in combination with other napari-based software, published our models on BioImage.IO[43] to offer them in a standard format and also provided a well-documented Python library. Our models can already be used within Deep MIB[44] and QuPath[45,46], which offer preliminary support for SAM. Integration with other tools that support interactive annotation, such as ilastik[24] or TrackMate[9], is also desirable.

We also plan to improve and extend µSAM across several dimensions. In the near future, we plan to train further models for biomedical applications, in particular a generalist EM model for organelle segmentation leveraging the data provided by Open Organelle[7] and models for other modalities such as histopathology data. We also want to implement automated tracking to speed up annotation with the tracking tool. To enable more efficient fine-tuning, we plan to extend the investigations into parameter-efficient training approaches to more recent methods than LoRA[40], which may provide faster training times in our setting. In addition, more efficient architectures[47] could replace the transformer-based encoder to reduce the computational cost for inference and training. To move toward a universal model for microscopy instance segmentation, we plan to also investigate how SAM (or similar models) can be made semantically aware, to enable ambiguous segmentation cases as in EM, how it can be extended to full 3D segmentation and how a unified model for several domains (LM, EM) can be trained.

While our contribution provides versatile and powerful functionality for interactive and automatic microscopy segmentation, it has some limitations compared to established approaches, mainly due to the larger computational footprint of vision transformers. While interactive data annotation is possible due to the modular design of SAM (enabling precomputation of image embeddings), automated processing of large datasets is not as efficient compared to CNN-based approaches such as CellPose or MitoNet. Furthermore, fine-tuning the SAM models on new data takes longer, especially on the CPU, so we do not provide 'human-in-the loop' fine-tuning as in CellPose, where the model is updated after each annotated image, but rather enable users to fine-tune through a separate user interface or scripts. The computational cost also prevents us from building a 3D segmentation approach that operates on orthogonal slices, as is done by CellPose and MitoNet; we process volumetric or time-series data slice by slice instead and use post-processing to avoid the resulting artifacts. Some artifacts due to 2D inference can still occur.

Our comparisons to CellPose and MitoNet are meant to provide a reference for automatic segmentation tools as they are available to a user. While we have done our best to compare to these methods fairly, we did not retrain them on our model's training data (which would be very challenging for the large dataset SAM is initially trained on). We do not claim superior performance compared to them; rather, we provide similar automatic segmentation quality for most practical purposes with the added benefit of interactive segmentation and support for more data modalities. Similarly, the user studies we conduct have many degrees of freedom, so depending on user experience and use case, the conclusion about tool suitability will vary. Nevertheless, we believe that these studies provide important context for the application of our tool in practice, and we have designed them to provide as fair a comparison as possible. In addition, we do not yet provide a single model that works equally well for multiple microscopy domains, but rather provide three sets of models (LM generalist, EM generalist for mitochondria and nuclei, default SAM) with different strengths. We have added a section

in our documentation to guide users through choosing the correct model for their application (https://computational-cell-analytics.github.io/micro-sam/micro_sam.html#choosing-a-model). We believe that, despite these limitations, μSAM offers the most versatile solution to (interactive) microscopy segmentation currently available and we are optimistic that the developments outlined herein will eventually address its limitations.

## Online content

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

## Methods

### SAM

SAM is a vision foundation model for interactive segmentation. It was introduced by Kirilov et al.[11]. Here, we briefly summarize its main functionality. It solves interactive segmentation tasks by predicting an object mask for annotations describing a given object in the input image. The annotations can be a bounding box, points (positive and/or negative) or a low-resolution mask. These annotations are called 'prompts' in the SAM publication; we use the terms 'prompt' and 'annotation' interchangeably. Kirilov et al. also describe segmentation based on text annotations, but the published version of the model does not include this feature. For a new image, the model predicts an embedding, which corresponds to a vector per pixel in a downscaled representation of the input, with the image encoder. The image encoder is a vision transformer[13] and SAM comes in three variants with different sizes of the encoder, using the ViT-B, ViT-L or ViT-H architectures (ordered by increasing model size). We also include a version using the smaller ViT-T, which was introduced by MobileSAM[39]. The image encoder contains the majority of the model's parameters. It has to be applied only once per image, enabling fast recomputation of object masks if the annotations change in interactive segmentation. The other parts of the model are the prompt encoders that encode the user annotations and the mask decoder that predicts the object mask and IOU score based on the image embedding and the encoded annotations. The IOU score corresponds to an estimate for the mask quality. To deal with the ambiguity of a single point annotation, which could refer both to an object or a part thereof, SAM predicts three different masks for this case. See Extended Data Fig. 1a for an overview of the SAM architecture.

The model is trained on a large labeled dataset of natural images that is constructed iteratively by annotators who correct the outputs of SAM trained on a previous version of this dataset. The model is then evaluated on a broad range of segmentation tasks and shows remarkable generalization performance to images from different domains. The authors also implement a method for AIS, termed AMG. It covers the input image with a grid of points and predicts masks for all points. The predicted masks are post-processed to retain only high-quality predictions. This involves filtering out masks with a low IOU prediction, and masks with a low stability score, which is computed based on the change of the masks when thresholded at different logit values. Finally non-maximum suppression is applied to remove overlapping predictions.

SAM was trained on RGB images, so the image encoder expects image data with three channels as input. To process microscopy images, which mostly have a single channel, we duplicate this channel three times. We found that this approach works well and assume that SAM was also trained on grayscale images using the same approach. Applying the model to data with a different number of channels, for example, two for a nuclear and cytosol stain such as in TissueNet, was more challenging. We tried two approaches: (i) appending an empty channel and (ii) averaging the two channels to obtain a single channel that is then duplicated three times. Both approaches have disadvantages: in the first case, the image statistics are altered compared to training by adding an empty channel, while in the second case, information is lost by averaging. We found that the second approach worked better and applied it in the relevant experiments. Note that this approach is detrimental compared to using both channels independently and constitutes a limitation when applying the current SAM architecture to multichannel images. The image is resized to 1,024 × 1,024 pixels before being passed into the image encoder.

### AIS

We extend the original SAM architecture with an additional decoder for predicting an AIS. This decoder is based on UNETR[48]. It consists of four blocks of two convolutional layers, each followed by a transposed convolution for upsampling. Each block receives the image encoder output as additional input. The output of the decoder has the same spatial dimensions as the input image. It predicts three output channels: the distance to the object center, the distance to the object boundary and foreground probabilities. The distances are normalized per object; see Supplementary Fig. 2a for a depiction of the targets used for training. We compute an instance segmentation based on them using a seeded watershed, using the implementation from scikit-image[49]. Both distance channels are used to derive seeds by finding connected regions with the center distance below a threshold parameter and the boundary distance above a threshold parameter. In addition to these seeds, the watershed uses the distance predictions as a heightmap and the thresholded foreground predictions as a mask. We have chosen this approach to segment complex object morphology with a rather simple procedure: using the boundary distances prevents merging narrow adjacent objects that would be falsely joined if only the center distances were used. Conversely, using the center distances prevents falsely splitting non-convex objects that have multiple connected regions in the thresholded boundary distance predictions. We call this approach AIS. This segmentation procedure is inspired by other approaches that use distance predictions for instance segmentation, for example, StarDist[3] or CellPose[1], but it uses a simpler post-processing logic.

We have validated our approach by comparing it to two other segmentation methods: predicting boundaries and foreground followed by watershed and predicting affinities followed by Mutex Watershed[50]. We trained a UNETR model based on the SAM ViT-B encoder on LIVECell for all three approaches, using 10,000 training iterations and otherwise using the same hyperparameters as described in the next section. We found that the distance-based approach (mean segmentation accuracy of 0.39) performed better than predicting affinities (0.36) and boundaries (0.31). We have further compared how our segmentation method works when using different network architectures. For this, we compare the UNETR architecture with a UNet[51] and a simpler architecture based on SAM that reuses the SAM image encoder and mask decoder to predict the foreground and distance channels for instance segmentation. The results are shown in Supplementary Fig. 4a,b. In summary, we see that the SAM-based architectures provide a big advantage for small training datasets, as long as their weights are initialized with a pretrained model, and that the UNETR architecture with convolutional decoder has an advantage over using the SAM mask decoder for this task.

Note that our segmentation approach also shares some similarities with CellVIT[21], which uses a SAM encoder for AIS in histopathology. However, CellVIT does not preserve the interactive segmentation capabilities of SAM. We have also evaluated its instance segmentation approach, which is based on predicting distance gradients, but found that it does not work well for touching objects.

### Training

To fine-tune SAM models, we implement and make available an iterative training scheme following the description in Kirilov et al.[11]. Note that the training algorithm by Kirilov et al. has so far not been released. Other tools that fine-tune SAM, for example, MedSAM[20], rely on a simpler training heuristic that only fine-tunes SAM for a specific kind of prompt, for example, box prompts. We have found that such approaches improve the segmentation quality for the given prompt type, but that they hamper it for other prompts (see also below). To provide a model for interactive segmentation, it is thus crucial to follow a similar training procedure as that used for training the initial SAM.

The training algorithm requires image data and corresponding ground-truth segmentations for the objects of interest. During training, we iterate over the complete training set several times in so-called epochs. In a single iteration, we sample a minibatch, corresponding to multiple images and the corresponding ground truth, apply the image encoder, derive prompts from the ground truth that are then passed to the prompt encoder and predict objects with the mask decoder.

We then compute the loss between predictions and ground truth and update the network weights via backpropagation and gradient descent. Compared to regular training approaches for instance segmentation, a single iteration is more complex as it is made up of multiple sub-iterations to mimic interactive segmentation. In more detail, a training iteration follows these steps:

1. Sample a minibatch containing input images and ground truth from the training set.
2. Sample a fixed number of objects from the ground truth. Training with all objects in a given image would require too much memory.
3. Predict the embeddings for the sampled image(s) with the encoder. Note that the encoder depends only on the image data and not on the prompts.
4. Perform the following steps for all sampled objects in a batched fashion:
   a. Sample a random point from the object, which is used as a positive input point, or use the bounding box of the object as the prompt.
   b. Predict the mask and expected IOU value for the given input with SAM. If the model is presented with a single point annotation, predict three output masks; otherwise, predict a single output mask. See the previous section for the motivation of this approach.
   c. Compute the loss between the predicted and ground-truth object as well as the loss between the estimated and true IOU score. If three objects were predicted, only the loss of the object with highest IOU prediction is taken into account.
   d. Sample two new points: a positive one where the model predicted background but where there should be foreground (according to the ground truth), and a negative one for the reverse case. If such points cannot be sampled because there is no region with missing foreground predictions or vice versa, we sample a random positive/negative point from within/outside the object.
   e. Present the combined annotations from the previous steps, that is, all points sampled so far and the box annotation if used in the first step, to the model. We also use the mask prediction from the previous step as an additional prompt with a probability of 50%; see next paragraph for more details on this step.
   f. Compute the mask and IOU loss for the current prediction.
5. Steps 4d–f are repeated for a fixed number of times (we use eight sub-iterations), all losses are accumulated; backpropagation and gradient descent are performed based on the average loss over all sub-iterations and update all parts of the model (image encoder, prompt encoder, mask decoder).

The goal of this training procedure is for the model to iteratively improve segmentation masks and provide a valid mask output for any input annotation. We implement it as described in Kirilov et al. with the exception of the mask sampling in step 4e. Here, the original training scheme uses the previous model prediction as mask input for the next sub-iteration every time rather than sampling it. We found that this approach leads the model to 'rely' on the presence of a mask prompt when multiple point annotations are given, resulting in degraded performance if this mask prompt is not given. To enable both settings, segmentation with multiple point prompts with or without a mask prompt, we introduce the aforementioned sampling procedure. See Supplementary Fig. 1 for a quantitative comparison of iterative segmentation with and without mask prompt using the default SAM and a model fine-tuned with our training implementation. We have also experimented with simpler training schemes that do not involve multiple sub-iterations and that instead only sample boxes and/or fixed numbers of point annotations from the ground truth. We found

that this approach leads to worse results for iterative segmentation; the model does not work well for interactive correction of the model predictions. Even simpler training approaches, like only training to segment from a box prompt as is done in MedSAM[15], lead to a further degradation of the model's capacity for interactive segmentation.

To train the segmentation decoder (see previous section), we interleave a training iteration for interactive segmentation and an iteration for automatic segmentation. Here, we make use of the same image and ground truth as sampled for interactive segmentation. We derive the target channels for the decoder from the ground truth: center and boundary distances as well as foreground map (see previous section and Supplementary Fig. 2a). We then compute the loss between these targets and the decoder predictions and update the weights of the image encoder and segmentation decoder based on it. We have also explored two other training strategies where we first train the model for interactive segmentation and then for AIS, trying both updating the weights of the image encoder and keeping them frozen. We found that training interactive and AIS jointly leads to the best results; the other strategies lead to diminished results either for interactive segmentation (if the image encoder weights are updated) or for AIS (if the encoder weights are frozen).

For the validation steps during training, we rely on a simpler procedure for interactive segmentation where we sample a bounding box and a fixed number of points per object, using the average Dice score between ground-truth and predicted objects as a metric. For automatic segmentation, we use the same loss function as in training as a metric, and add up the metric values for interactive and automatic validation. All experiments reported in this paper rely on fine-tuning the weights provided by the SAM publication; in some experiments, we further fine-tune our models. Our training method could also be used to train a model from randomly initialized weights. However, we expect this approach to drastically increase training times and thus did not pursue it.

We use the following settings and hyperparameters for training:

- We use a batch size of two, that is, two images and the corresponding ground truth are sampled per batch. In cases where we train with constrained resources (Fig. 5a and Extended Data Fig. 10) we use a batch size of 1. Further training hyperparameters are documented in Extended Data Fig. 10c.
- We train all models with a patch shape of 512 × 512 pixels; some training datasets contain smaller images, which are zero padded to match this shape. The only exception is the LIVECell specialist, which we have trained with a patch shape of 520 × 704 (the full image shape).
- We use the Dice loss to compare ground-truth objects and mask predictions for interactive and automatic segmentation.
- We use the L2 loss to compare true and predicted IOU scores.
- We use the ADAM optimizer[52] with an initial learning rate of $10^{-5}$. We also investigated the impact of learning rate and optimizer, trying the learning rates $5 \times 10^{-4}$, $10^{-4}$, $5 \times 10^{-5}$, $10^{-5}$ and $5 \times 10^{-6}$ as well as using ADAMW[53] instead of ADAM. We found that using higher learning rates than $10^{-5}$ led to worse results and did not find an effect of the other parameter choices.
- For training the decoder outputs for AIS, we use the average Dice loss over the three predicted channels, center and boundary distances as well as foreground predictions, masking the loss in the background for the two distance channels. Somewhat counterintuitively, we found that using Dice as the loss function for the distance predictions works better than using the L2 loss. Note that the distance channels are normalized to the range [0, 1], so the Dice loss is well defined.
- We lower the learning rate when the validation metric plateaus (ReduceLROnPlateau).
- For most experiments, we train the models for 250,000 iterations and use the epoch that achieves the best validation metric.

We found that the EM and LM generalist models, especially ViT-L and ViT-H, which are both trained on large and diverse datasets, kept improving late in training, and use this setup for all experiments where we train and compare to these models (Figs. 3a and 4a and Extended Data Figs. 1b, 2, 5 and 8).

- For investigating which model parts to fine-tune (Fig. 1b) and investigating lower training data fractions on LIVECell (Fig. 1c and Extended Data Fig. 1c), we train the models with early stopping for a maximum of 100,000 iterations. For resource-constrained settings (Fig. 5b and Extended Data Fig. 10a) and the user study (Fig. 6), we also use early stopping and train for a maximum of 100 epochs.

- Unless stated otherwise, the models were trained on a A100 GPU with 80 GB of VRAM, where training a model for 250,000 iterations took about 6 days. We provide an overview of all hardware configurations used for our experiments in Extended Data Fig. 10c and list representative training times with early stopping in Extended Data Fig. 10d.

For the implementation, we reuse the code from Kirilov et al. wherever possible and implement the additional training logic with PyTorch[54] and torch-em[55], a PyTorch-based library for deep learning applied to microscopy also developed by us.

## Inference and evaluation

To quantitatively evaluate SAM for interactive segmentation, we mimic user-based segmentation with the model, following a similar logic as described in the previous section.

We derive prompts from the ground-truth objects, run the model with the image and these prompts as input and evaluate the predicted masks. We do not compute any loss functions and don't accumulate gradients. We implement two different evaluation approaches—one where the mask from the previous iteration is always used, one where it is not used (step 4e; see also the evaluation in Supplementary Fig. 1). We evaluate the results for each of the sub-iterations individually by computing the mean segmentation accuracy compared to the ground-truth masks (see below). In Fig. 2a, Extended Data Fig. 1b and Supplementary Fig. 1, we report the mean segmentation accuracy for each individual sub-iteration, stopping after seven iterations. We distinguish the cases where we start from a box prompt (red bars) or from a single point prompt (green bars). For all other figures, we only report the mean segmentation accuracy for the zeroth sub-iteration (that is, segmentation based only on the initial box or point prompt) and the respective last sub-iteration. Note that the point prompts are sampled randomly (subject to prediction errors in previous sub-iterations); we investigate the influence of this randomness in Extended Data Fig. 1b.

We evaluate models for automatic segmentation by computing the mean segmentation accuracy (see below) between model prediction and ground-truth masks. When evaluating AMG, we found that it was crucial to also optimize two of its hyperparameters: the IOU and stability thresholds that are used for filtering out low-quality predictions (see also the first Methods section). While the default settings work well for the original SAM models, they have to be lowered for the fine-tuned models. Presumably, this is because these models are better calibrated to the actual prediction quality for objects in microscopy, which is lower compared to natural images. To efficiently perform a grid search, we precompute the predicted object masks and then evaluate the hyperparameter ranges to be tested. The parameter search is performed on a separate validation set, and the best setting found is applied to the test set. For AIS, we determined the best parameters for the threshold applied to center and boundary distances similarly via grid search. In the annotation tool (next section), the best values for AIS and AMG parameters are automatically set for the selected model.

For comparisons with CellPose, we use the most suitable CellPose model for the given data (at the time of running the experiments), corresponding to the CellPose specialist models for LIVECell and TissueNet and the cyto2 model otherwise. We have used these models with default settings and ran prediction with the CellPose Python library. For MitoNet, we use the napari plugin for 2D segmentation with the MitoNet_v1 model with default parameters. Note that the MitoNet Python library was not available open source at the time of running the experiments, so we resorted to using the napari plugin (https://github.com/volume-em/empanada-napari/) instead.

We evaluate segmentation results with the mean segmentation accuracy. The segmentation accuracy, SA($t$), was introduced in Everingham et al.[27] and is defined in terms of true positives, TP($t$), false positives, FP($t$), and false negatives, FN($t$), at IOU threshold $t$ as: SA($t$) = TP($t$)/(TP($t$) + FP($t$) + FN($t$)). TP($t$), FP($t$) and FN($t$) are computed by matching segmentation and ground truth on a per-object level and counting matches with a higher IOU value than $t$ as TP($t$), unmatched objects in the prediction as FP($t$) and unmatched objects in the ground truth as FN($t$). The mean segmentation accuracy is then computed by averaging SA($t$) over thresholds $t$ in the range from 0.5 to 0.95 with increments of 0.05. We compute this score per image and then average it over all images of a given evaluation dataset. This metric has been popularized for microscopy by the DSB Nucleus Segmentation Challenge[26] and has recently been studied in depth by Hirling et al.[56] in the context of microscopy segmentation. The mean segmentation accuracy is a stringent evaluation criterion, because it includes the evaluation at high IOU thresholds, which penalize even small deviations from the ground-truth objects. For this reason, we use the less stringent SA (0.5) measure for some experiments where we found that it was too strict for a meaningful evaluation.

To evaluate the quality of tracking results in the user study, we use the tracking metric introduced by the Cell Tracking Challenge[31]. This metric matches the graph defined by the ground-truth tracking annotations and the graph defined by the predicted tracking result to each other and then counts errors in this matching. We use the implementation provided by the 'traccuracy' repository (https://github.com/Janelia-Trackathon-2023/traccuracy/).

## Interactive annotation tools and Python library

We extend the core functionality of SAM to support caching of precomputed image embeddings, tiled computation of image embeddings and multidimensional segmentation based on projecting prompts to adjacent slices/time frames. We implement this functionality in our μSAM Python library, using scipy[57] and scikit-image[49] to implement additional image processing logic. We also use the scientific Python libraries numpy[58], pandas[59] and matplotlib[60] to implement our library and to perform additional data analysis and plotting for this paper. Our Python library also implements the training and evaluation functionality described in the previous sections. It further contains the implementation of our napari plugin, which implements five different widgets: for 2D annotation, for 3D annotation, for high-throughput image annotation, for tracking and for model training. A more detailed description of our implementation for this functionality can be found in the Supplementary Information.

## User study

We perform three different user studies to demonstrate the usefulness of our napari tool for 2D segmentation, 3D segmentation and tracking. The first user study is performed by five different annotators and we compare μSAM and CellPose for annotating organoids in brightfield images. We perform this user study with multiple annotators in order to study the difference of annotation performance between users and compare different annotation modes for both tools. In the second user study, a single annotator segments nuclei in 3D EM, comparing μSAM with ilastik carving. In the last user study, nuclei are tracked in

fluorescence microscopy by a single annotator, comparing interactive tracking with μSAM and tracking and correction with TrackMate. A detailed description of the user studies can be found in the Supplementary Information.

**Reporting summary**
Further information on research design is available in the Nature Portfolio Reporting Summary linked to this article.

## Data availability
We use publicly available datasets for most experiments (Supplementary Tables 1 and 2). We generate new datasets for the 2D and 3D user studies. The dataset for the 2D user study consists of brightfield microscopy images of organoids and is available on Zenodo via https://doi.org/10.5281/zenodo.14036956 (ref. 61). The dataset for the 3D user study consists of volume EM blocks, in which we have annotated nuclei. We have deposited this dataset on Zenodo via https://doi.org/10.5281/zenodo.14037020 (ref. 62).

## Code availability
Our software is available on GitHub under a permissive open-source license at https://github.com/computational-cell-analytics/micro-sam/. It is documented at https://computational-cell-analytics.github.io/micro-sam/micro_sam.html. The version at submission of this manuscript is 1.1.1. Our LM and EM generalist models are available on BioImage.IO and Zenodo. Please refer to our model documentation for the DOIs of individual models. Additional models are deposited on Zenodo and the corresponding links are given at https://computational-cell-analytics.github.io/micro-sam/micro_sam.html#finetuned-models. The tables and code for generating quantitative plots are available on GitHub. Additional code for the analysis of the 2D annotation user study is available at https://github.com/computational-cell-analytics/user-study-v3/.

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

## Acknowledgements
We express our gratitude to Sartorius for their support of this research through the Quantitative Cell Analytics Initiative (QuCellAI) and thank all partners involved in the initiative for their contributions and valuable insights. We also gratefully acknowledge the computing time granted by the Resource Allocation Board and provided on the supercomputer Lise and Emmy at NHR@ZIB and NHR@Göttingen as part of the NHR infrastructure. The calculations for this research were conducted with computing resources under the project nim00007. The work of A.A. was funded by the Deutsche Forschungsgemeinschaft (DFG, German Research Foundation) - PA 4341/2-1. The work of L.F. and Su.N. was funded by the DFG under Germany's Excellence Strategy - EXC 2067/1-390729940. This work was also supported by the Google Research Scholarship 'Vision Foundation Models for Bioimage Segmentation'. The work of G.B. was supported by Microscopy Australia (ROR: 042mm0k03) at the Ramaciotti Centre for Cryo-Electron Microscopy, Monash University (ROR: 02bfwt286), enabled by NCRIS. The work of M.W. was supported by Deutsche Forschungsgemeinschaft - WI 6148/1-1, German Cancer Aid – 70115444, and Hector-Foundation M2408. We thank T. Lüddecke for helpful discussions on designing training objectives for SAM. In addition, we acknowledge the groups of G. Schneider and M. Grade at University Medicine Göttigen as well as M. Pankratz at University of Bonn for providing data for the user studies. In addition, we acknowledge the group of M. Pankratz at University of Bonn for providing data for the user studies.

## Author contributions
C.P., A.A., S.A., A.D. and N.K. conceptualized the work. A.A., C.P., L.F., G.B. and P.H. implemented the software. A.A., C.P., S.N., V.R., N.K., S.G., C.T. and M.F. performed experiments. M.S., C.T.C., M.G., M.W. and G.S. provided experimental methods, resources and data. A.A., S.V.H. and C.P. visualized the results. C.P. and A.A. drafted the manuscript. A.A., L.F., S.N., N.K., P.H., V.R., M.F., C.T., G.B., S.V.H., S.G., A.D., S.A. and C.P. reviewed and edited the final manuscript.

## Funding

## Competing interests
The authors declare no competing interests.

## Additional information
**Extended data** is available for this paper at https://doi.org/10.1038/s41592-024-02580-4.

**Correspondence and requests for materials** should be addressed to Constantin Pape.

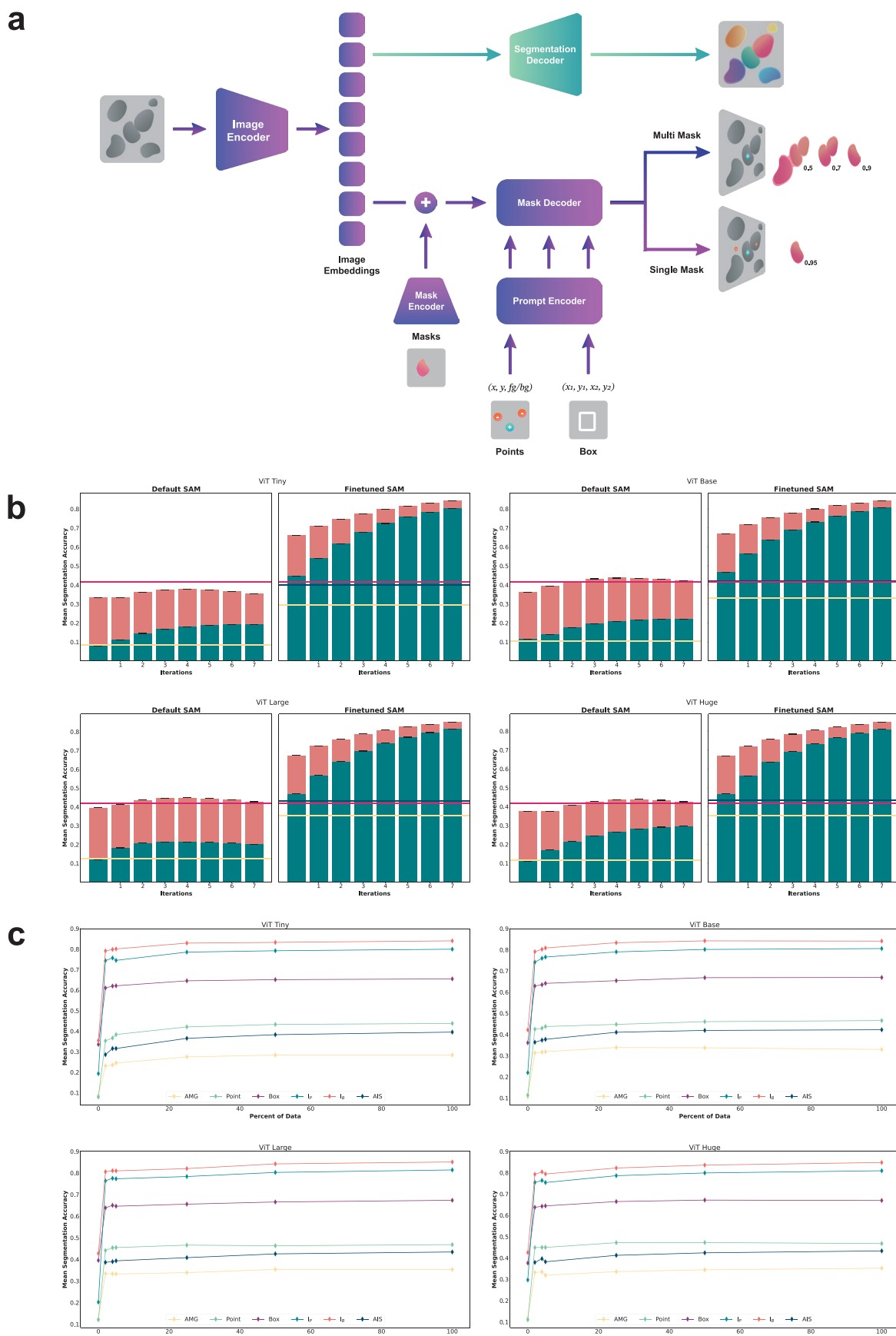

**Extended Data Fig. 1 | See next page for caption.**

**Extended Data Fig. 1 | SAM Architecture and extended LIVECell results.** SAM architecture and extended results on LIVECell. **a**. SAM takes the image and object annotations as input and predicts mask(s) and IOU score(s). The image encoder computes the embeddings, which are independent of the annotations, the prompt encoders encode the mask, point and/or box annotations and the mask decoder predicts the output mask(s) and score(s). In the case of annotation with a single point, the model predicts three potential output masks to deal with ambiguity; for example predicting the individual object highlighted by the point in the example or also predicting the objects touching it. The predicted score gives the confidence for the correctness of the mask. **b**. Results for SAM (default and fine-tuned) on LIVECell with different image encoder sizes (ViT-T, ViT-B, ViT-L, ViT-H). We use the same experimental set-up as in Fig. 2a. The black error bars indicate the standard deviation over five independent runs of the interactive segmentation evaluation procedure. Note that this procedure includes randomness because it samples prompts to correct the segmentation masks according to segmentation errors from previous iterations. **c**. Training on reduced LIVECell datasets for all image encoder sizes; same experimental set-up as Fig. 2c with different image encoder sizes.

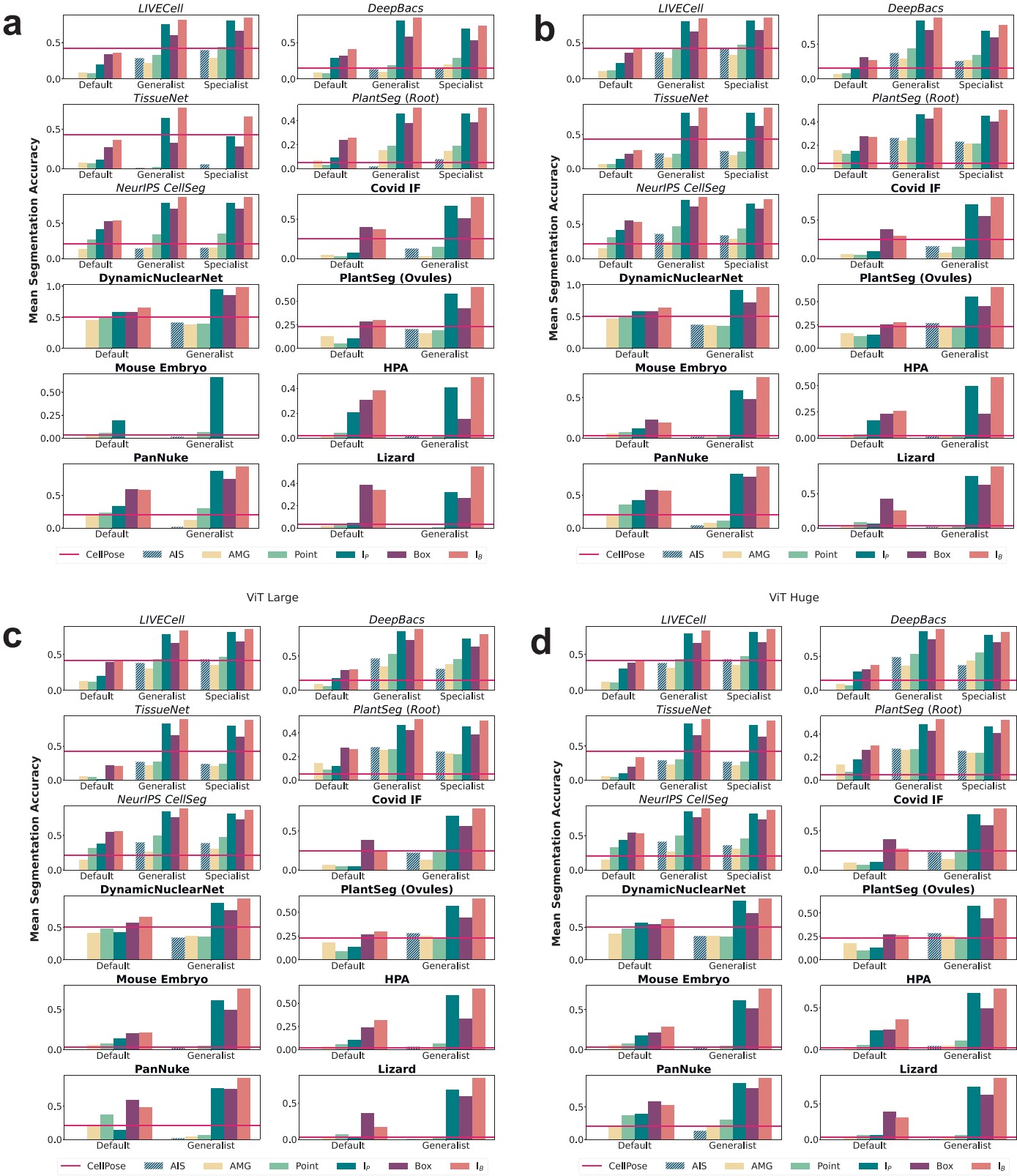

**Extended Data Fig. 2 | Extended quantitative evaluation for light microscopy models.** Comparison of default SAM, LM generalist, and specialist models as well as CellPose. Same experimental set-up as in Fig. 3a, but we compare on additional datasets and report the results for all image encoder sizes (**a - d**). See Supplementary Table 1 for dataset references.

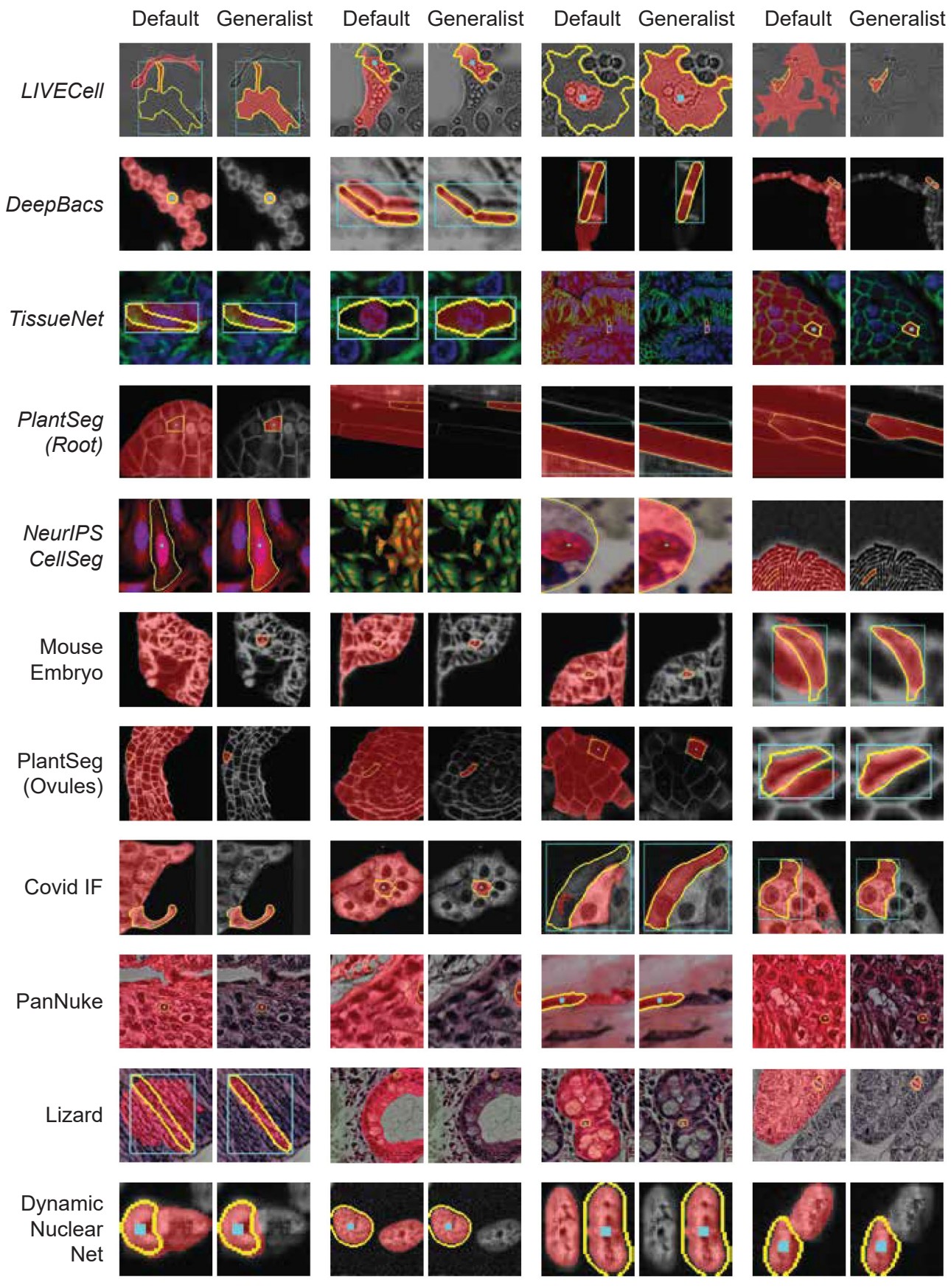

**Extended Data Fig. 3 | See next page for caption.**

**Extended Data Fig. 3 | Qualitative interactive segmentation results for light microscopy I.** Qualitative comparison of interactive segmentation for the default SAM and our LM generalist. For both the model based on ViT-L is used. Cyan shows the input point or box annotation, yellow the correct object and red the model prediction. We select examples with the best improvement in IOU score of the generalist compared to the default model to highlight typical improvements. The most consistent improvement is that the generalist correctly segments individual cells in clusters, whereas the default model segments the whole cluster. This figure serves to give an impression of how the interactive segmentation is improved; the quantitative improvement can be seen in Fig. 3a and Sup. Figure 2.

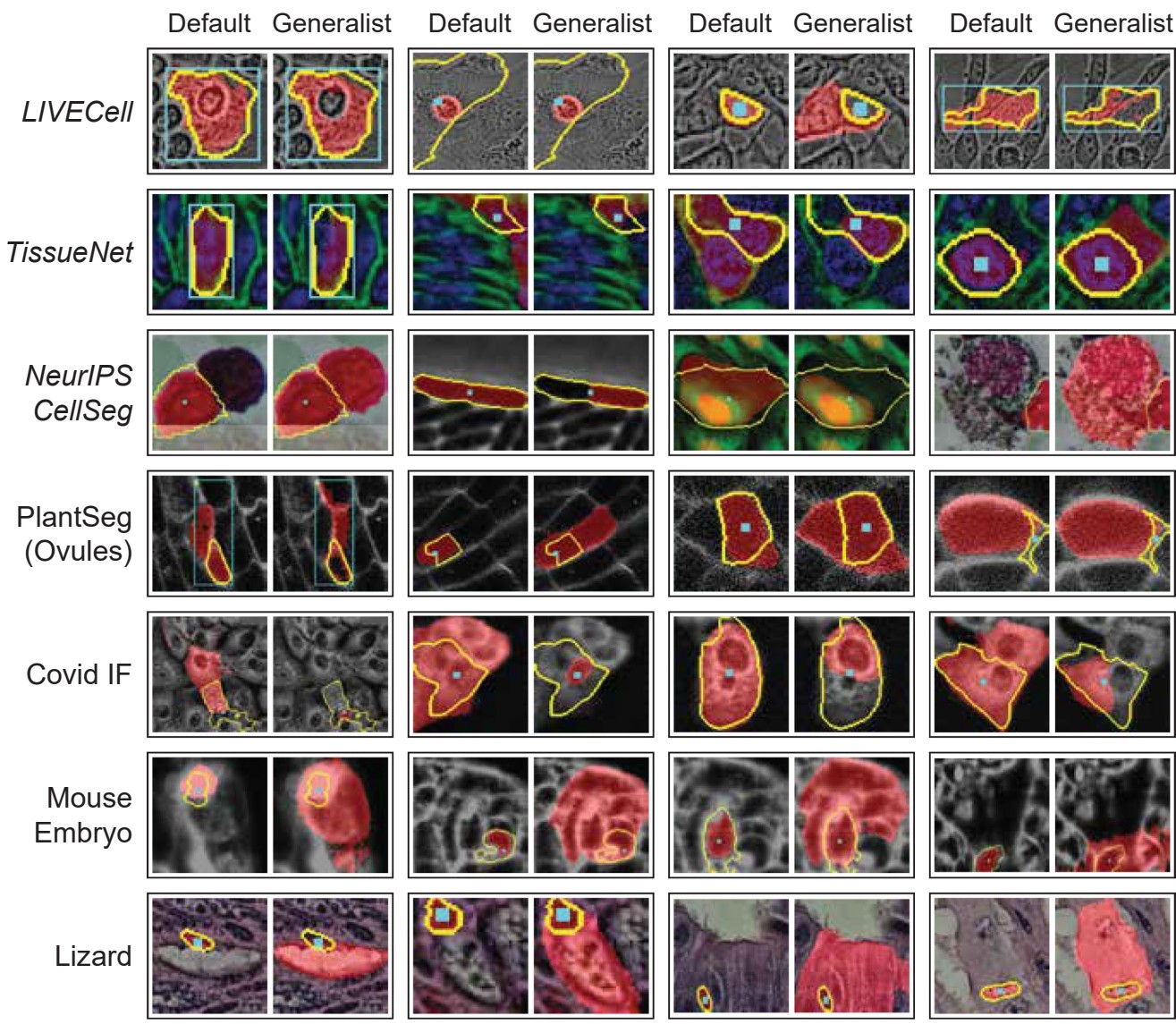

**Extended Data Fig. 4 | Qualitative interactive segmentation results for light microscopy II.** Qualitative comparison of interactive segmentation for the default SAM and our LM generalist (ViT-L). Opposite approach to Extended Data Fig. 3: we show the objects where the decrease in IOU is largest comparing the generalist and default model. Here, we see a few different effects: in some cases the generalist model segments several nearby cells (proving an exception to the general behavior observed previously) for point annotations, in other cases the segmentation quality is lower because the generalist segments smaller substructures. This systematic effect can also be observed for Covid IF, where the generalist often segments only the nucleus, which is discernible from the rest of the cell, rather than the full cell. Note that the quantitative segmentation quality for all these datasets is clearly higher for the generalist model as shown in Fig. 3 and Extended Data Fig. 2.

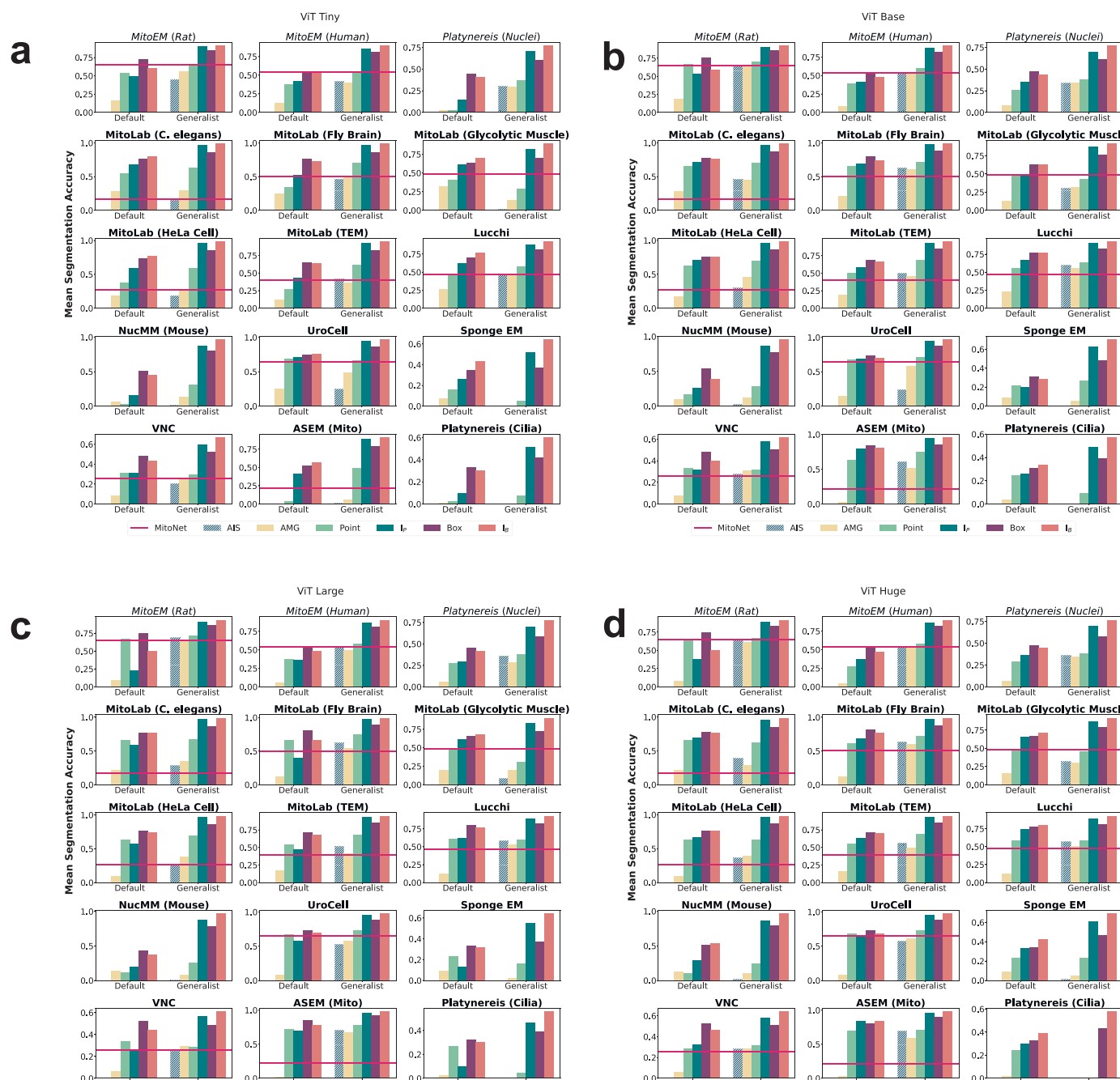

**Extended Data Fig. 5 | Extended quantitative evaluation for electron microscopy models.** Comparison of the default SAM and our EM generalist, with MitoNet as reference for automatic mitochondrion segmentation. We use the same experimental set-up as in Fig. 3 but give results for all image encoder sizes (**a - d**) and additional datasets. Note that the datasets **Sponge EM** and **Platynereis (Cilia)** evaluate segmentation for cilia and microvilli, which the generalist models were not trained for. They still yield improved results (except for segmentation with a single point prompt). See Supplementary Table 2 for dataset references.

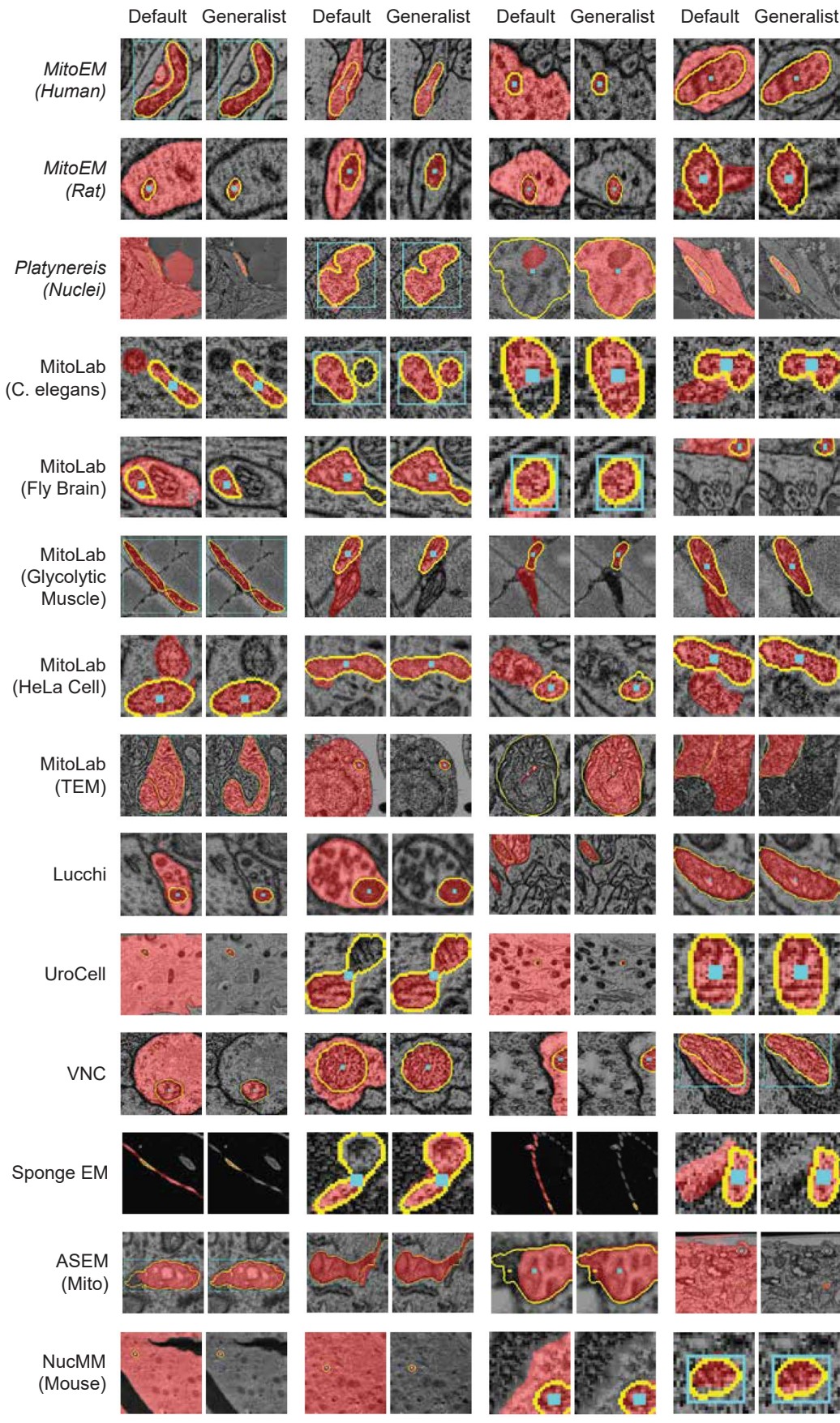

**Extended Data Fig. 6 | See next page for caption.**

**Extended Data Fig. 6 | Qualitative interactive segmentation results for electron microscopy I.** Qualitative comparison of interactive segmentation for the default SAM and our EM generalist (ViT-L). Cyan shows the input point or box annotation, yellow the correct object and red the model prediction. We select examples with the best improvement from the generalist model (see also Extended Data Fig. 3). The generalist model overall adheres better to the object boundaries and for single point annotations segments the selected organelle instead of the surrounding compartment. It also avoids segmenting touching objects. This figure serves to give an impression of how the interactive segmentation is improved; the quantitative improvement can be seen in Fig. 4a and Extended Data Fig. 5.

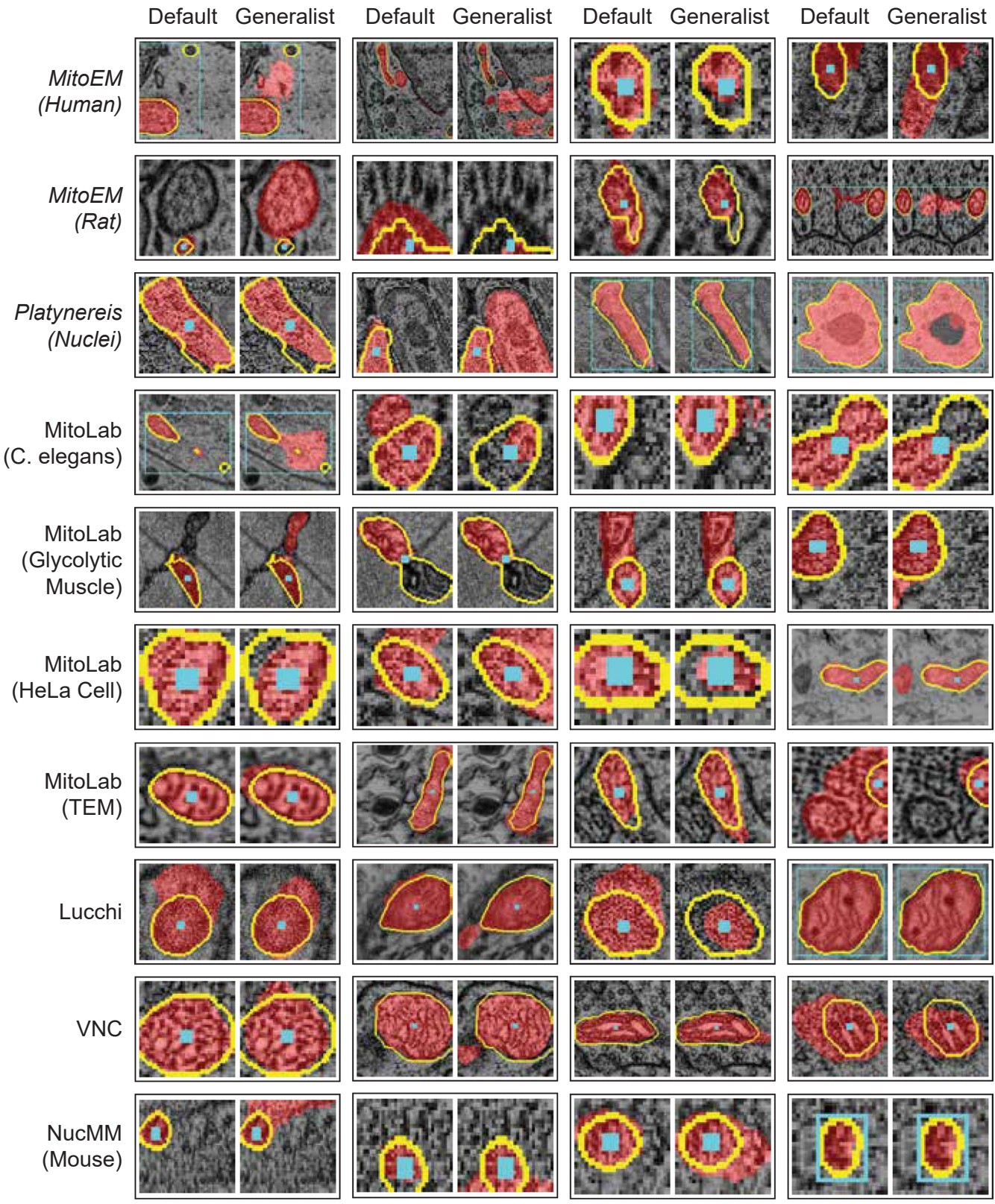

**Extended Data Fig. 7 | Qualitative interactive segmentation results for electron microscopy II.** Qualitative comparison of interactive segmentation for the default SAM and our EM generalist (ViT-L). Opposite approach to Extended Data Fig. 6: we show the objects with the largest disadvantage for the generalist model (see also Extended Data Fig. 4). Note that the quantitative segmentation quality for all these datasets is better with the generalist as shown in Fig. 4 and Extended Data Fig. 5.

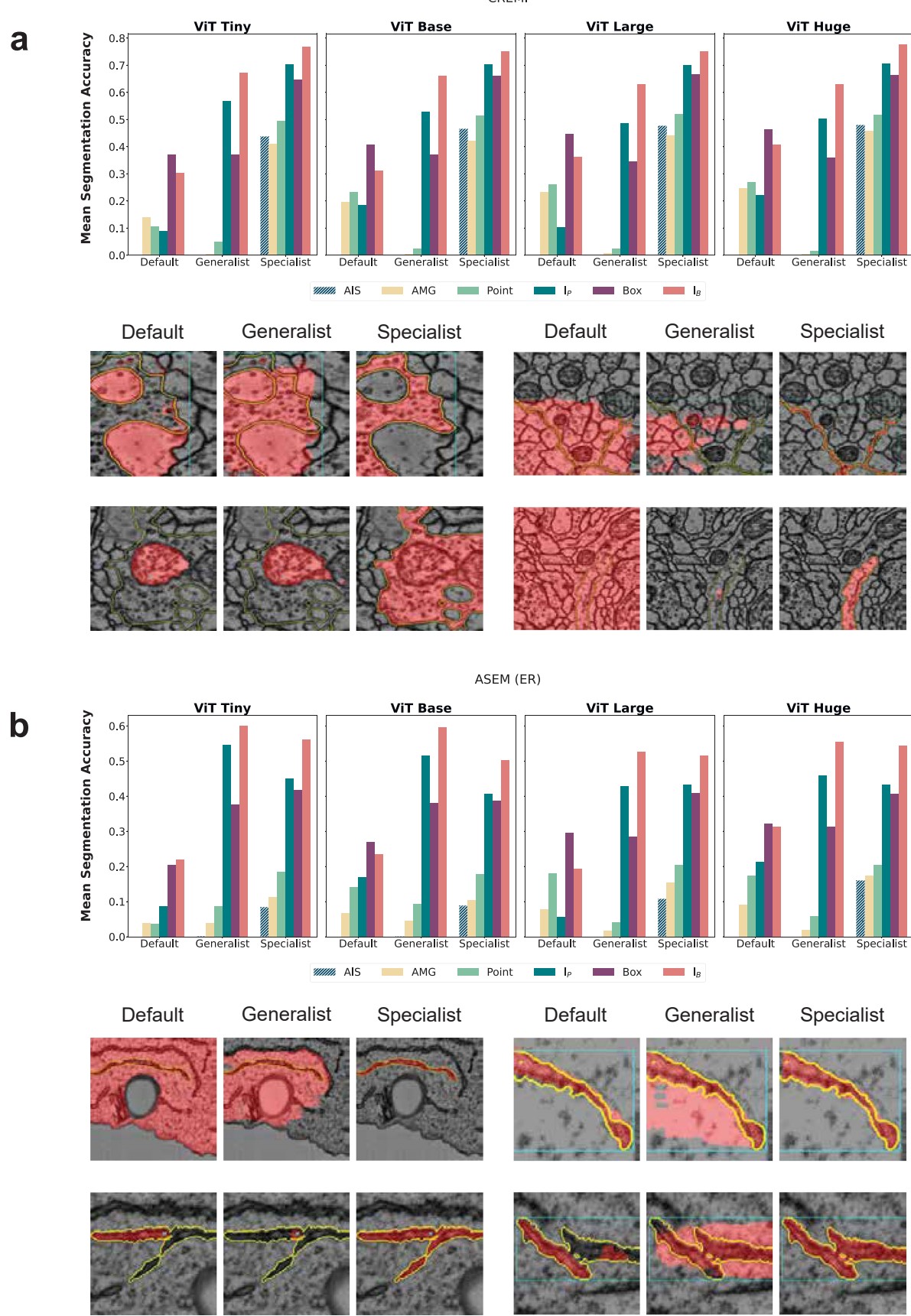

**Extended Data Fig. 8 | See next page for caption.**

**Extended Data Fig. 8 | Segmentation results for neuron and other organelle segmentation in electron microscopy.** Segmentation of other structures in EM. **a**. Segmentation of neurites in EM using the CREMI[58] dataset. We compare the default SAM, our EM generalist and a specialist model. The specialist is fine-tuned starting from default SAM on a separate training split; the models are evaluated on the same test split; the evaluation is in 2D and follows the usual approach. The images below compare qualitative results for interactive segmentation with the three models. All models are based on ViT-L. We see that the generalist overall decreases the segmentation quality for this task because it was trained to segment organelles rather than membrane compartments like neurites. Only interactive segmentation after correction ($I_P$ and $I_B$) is improved, which can be partly explained by the effect discussed in Supplementary Fig. 1. The specialist model clearly improves the segmentation results across all settings.

**b**. Endoplasmic reticulum (ER) segmentation. We follow the same strategy as in **a**, but for segmenting ER instead of neurites, using the ASEM dataset from Gallusser et al.[41]. Here, we somewhat surprisingly observe that the two smaller models (ViT-T, ViT-B) perform better than the two larger models in some settings. Annotation quality with a single point and AMG quality decrease for the generalist compared to the default model, but annotation with a box improves or does not change much (depending on the model). Interactive segmentation ($I_P$ and $I_B$) improves. In summary the generalist does not have a clear advantage over the default model. Training a specialist, with the default model as starting point, improves results in all settings compared to the default model and is better than or on par with the generalist in almost all settings, except for interactive segmentation with ViT-T and ViT-B.

**a**

| Light Microscopy | | | | Electron Microscopy | | | |
|---|---|---|---|---|---|---|---|
| Dataset | Model | Method | SA50 | Dataset | Model | Method | SA50 |
| PlantSeg (Ovules) | µSAM (Default) | Interactive | 0.383 | Lucchi | µSAM (Default) | Interactive | 0.674 |
| | µSAM (Generalist) | Interactive | 0.453 | | µSAM (Generalist) | Interactive | 0.756 |
| | µSAM (Generalist) | Automatic | 0.270 | | µSAM (Generalist) | Automatic | 0.455 |
| *PlantSeg (Root)* | µSAM (Default) | Interactive | 0.225 | *MitoEM (Rat)* | µSAM (Default) | Interactive | 0.709 |
| | µSAM (Generalist) | Interactive | 0.369 | | µSAM (Generalist) | Interactive | 0.833 |
| | µSAM (Generalist) | Automatic | 0.165 | | µSAM (Generalist) | Automatic | 0.706 |
| | | | | *MitoEM (Human)* | µSAM (Default) | Interactive | 0.564 |
| | | | | | µSAM (Generalist) | Interactive | 0.750 |
| | | | | | µSAM (Generalist) | Automatic | 0.606 |

**b**

Lucchi

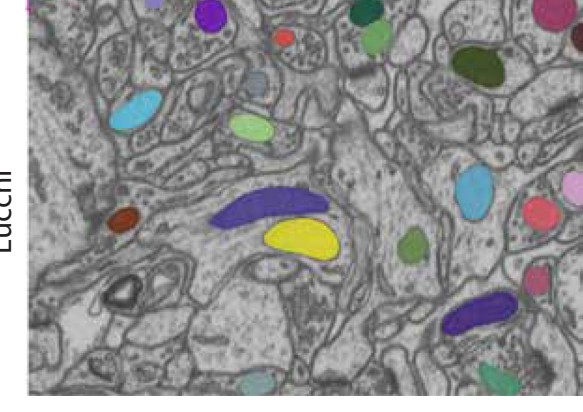 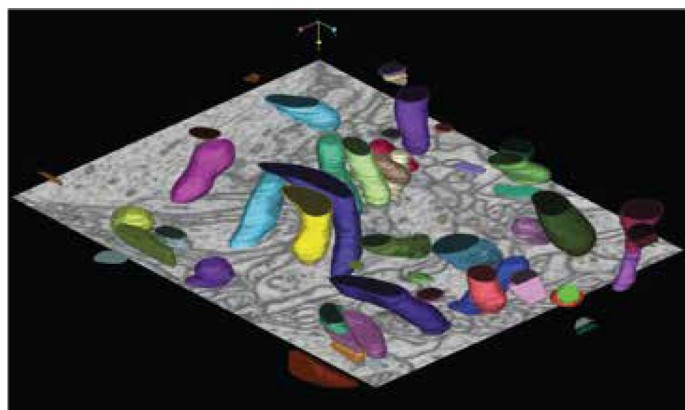

PlantSeg (Ovules)

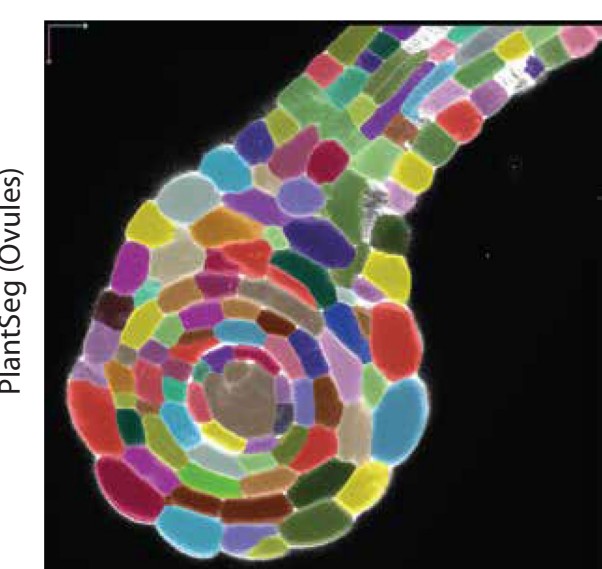 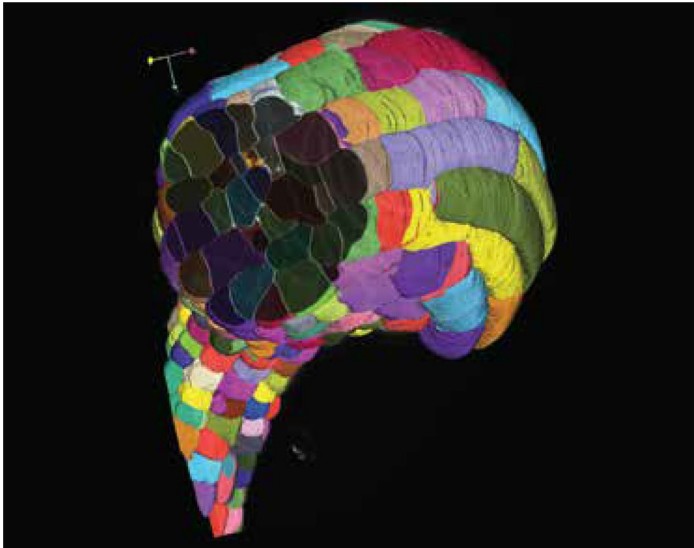

**Extended Data Fig. 9 | Volumetric segmentation results.** Interactive and automatic 3D segmentation. **a.** Quantitative evaluation for interactive and automatic segmentation with default SAM and the LM generalist for cell segmentation (left) / the default SAM and the EM generalist (right); using the ViT-B models. We use a confocal microscopy volume from PlantSeg (Ovules)[30] / a FIBSEM volume from Lucchi et al.[36] for the experiments. For interactive segmentation we derive a single prompt in the middle slice per object and then run our interactive volumetric segmentation approach based on projecting prompts to adjacent slices. For automatic segmentation we use the slice by slice segmentation approach, followed by merging of segments across slices. We report the result for AIS with our generalist models; 3D segmentation via AMG is too inefficient to run it here. We report the SA50 metrics (segmentation accuracy at an IOU of 50%) because we found that mean segmentation accuracy is too stringent for these 3D segmentation problems. **b.** 2D and 3D visualizations of the results for automatic segmentation for both datasets.

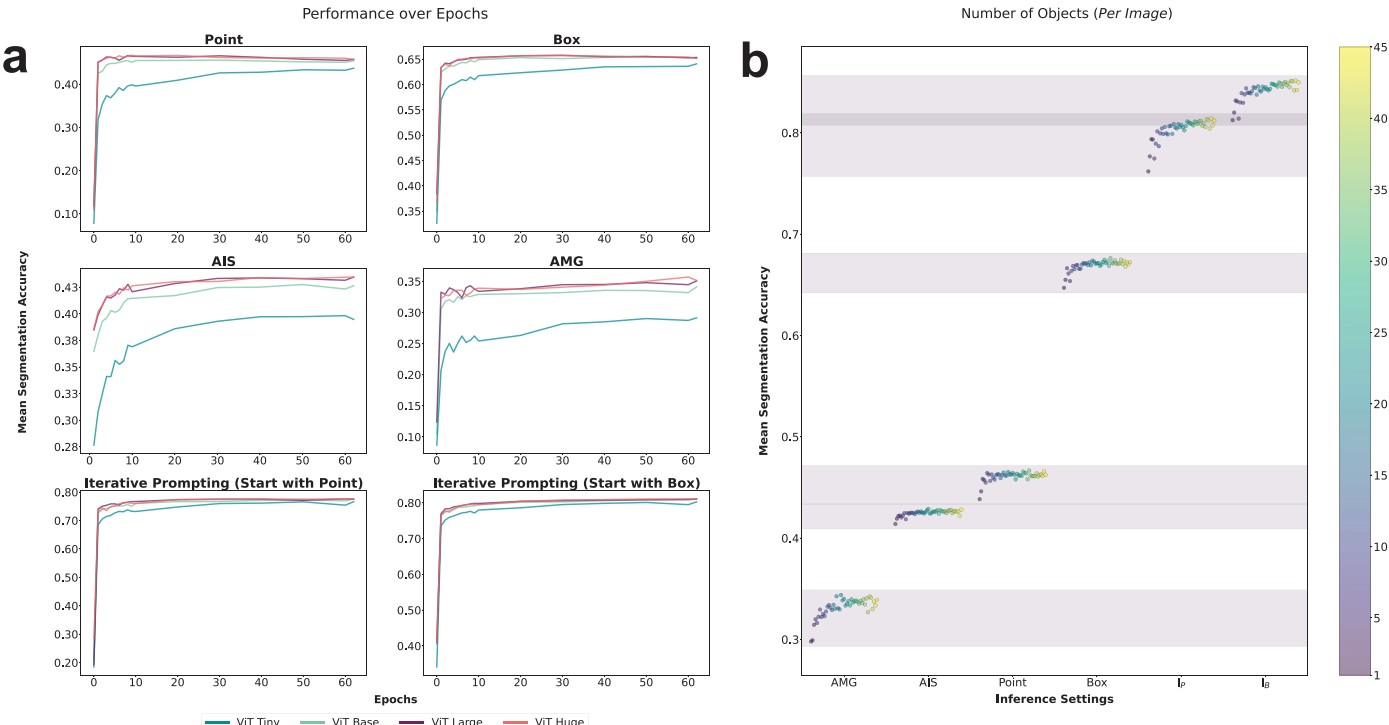

**c**

| Resource | Compute Capacity | Model | Batch Size | Finetuned Parts | No. of Objects |
|---|---|---|---|---|---|
| CPU | 32GB | ViT Base | 1 | *all* | 10 |
| CPU | 64GB | ViT Base | 1 | *all* | 15 |
| GTX1080 | 8GB | ViT Base | 1 | MD, PE | 10 |
| RTX5000 | 16GB | ViT Base | 1 | *all* | 10 |
| V100 | 32GB | ViT Base | 1 | *all* | 35 |
| A100 | 80GB | ViT Large | 2 | *all* | 30 |

**d**

| Resource | Model | Finetuning Stategy | Best Epoch | Train Time (in *hh:mm:ss*) |
|---|---|---|---|---|
| CPU (32G) | Default | *Full FT* | 24 | 5:41:31 |
| CPU (32G) | Default | *LoRA* | 13 | 2:57:08 |
| CPU (32G) | Generalist | *Full FT* | 6 | 2:01:30 |
| CPU (32G) | Generalist | *LoRA* | 7 | 1:58:57 |
| CPU (64G) | Default | *Full FT* | 15 | 3:51:02 |
| CPU (64G) | Default | *LoRA* | 19 | 5:20:02 |
| CPU (64G) | Generalist | *Full FT* | 5 | 1:28:26 |
| CPU (64G) | Generalist | *LoRA* | 15 | 5:42:34 |
| GTX1080 | Default | MD, PE | 40 | 1:18:05 |
| GTX1080 | Generalist | MD, PE | 13 | 0:15:05 |
| RTX5000 | Default | *Full FT* | 43 | 0:46:55 |
| RTX5000 | Default | *LoRA* | 16 | 0:17:37 |
| RTX5000 | Generalist | *Full FT* | 3 | 0:04:22 |
| RTX5000 | Generalist | *LoRA* | 32 | 0:34:04 |
| V100 | Default | *Full FT* | 20 | 0:26:24 |
| V100 | Default | *LoRA* | 42 | 0:51:10 |
| V100 | Generalist | *Full FT* | 2 | 0:03:48 |
| V100 | Generalist | *LoRA* | 5 | 0:07:11 |

**Extended Data Fig. 10 | Model finetuning in resource-constraint settings.**
Resource constrained finetuning. **a**. Improvement of different segmentation
settings with training epochs for finetuning ViT-T,B,L,H on LIVECell. We train for
100,000 iterations, otherwise using the same settings as in Fig. 2a. We see that
the majority of improvements happen early, motivating the use of early stopping
in resource constrained settings. **b**. Influence of the number of objects per image
used during finetuning, which is the most important training hyperparameter
and also determines the VRAM required for training. The experiments are for a
ViT-B trained for 100,000 iterations on LIVECell with 1-45 objects per image and
we show evaluations for the usual segmentation settings. We see that increasing
the number of objects initially strongly improves results and then plateaus or
improves results with a smaller slope. **c**. Best hyperparameter settings for the

hardware configurations we have tested. For each configuration we first looked
if training ViT-L is possible (only for A100), using ViT-B otherwise, then how many
objects could fit. For A100 we use a batch size of 2 and for all other settings a
batch size of 1. For the GTX 1080 it is not possible to fine-tune the full ViT-B model
and it is only possible to fine-tune mask decoder (MD) and prompt encoder (PD),
which limits the model improvements, see also Fig. 2b. **d** Training times in epochs
and minutes for finetuning models on Covid IF (Supplementary Fig. 4) using
the different hardware configurations and best settings according to **c**, when
updating all weights (*Full FT*) or using parameter-efficient training (*LoRA*) We use
early stopping after 10 epochs without improvement and start training either
from the default model or LM generalist.

# Reporting Summary

## Statistics

For all statistical analyses, confirm that the following items are present in the figure legend, table legend, main text, or Methods section.

| n/a | Confirmed | |
|---|---|---|
| ☐ | ☒ | The exact sample size (*n*) for each experimental group/condition, given as a discrete number and unit of measurement |
| ☒ | ☐ | A statement on whether measurements were taken from distinct samples or whether the same sample was measured repeatedly |
| ☒ | ☐ | The statistical test(s) used AND whether they are one- or two-sided *Only common tests should be described solely by name; describe more complex techniques in the Methods section.* |
| ☒ | ☐ | A description of all covariates tested |
| ☒ | ☐ | A description of any assumptions or corrections, such as tests of normality and adjustment for multiple comparisons |
| ☐ | ☒ | A full description of the statistical parameters including central tendency (e.g. means) or other basic estimates (e.g. regression coefficient) AND variation (e.g. standard deviation) or associated estimates of uncertainty (e.g. confidence intervals) |
| ☒ | ☐ | For null hypothesis testing, the test statistic (e.g. *F*, *t*, *r*) with confidence intervals, effect sizes, degrees of freedom and *P* value noted *Give P values as exact values whenever suitable.* |
| ☒ | ☐ | For Bayesian analysis, information on the choice of priors and Markov chain Monte Carlo settings |
| ☒ | ☐ | For hierarchical and complex designs, identification of the appropriate level for tests and full reporting of outcomes |
| ☒ | ☐ | Estimates of effect sizes (e.g. Cohen's *d*, Pearson's *r*), indicating how they were calculated |

*Our web collection on statistics for biologists contains articles on many of the points above.*

## Software and code

Policy information about availability of computer code

| | |
|---|---|
| Data collection | We have created a new open source software tool for microscopy image segmentation. The software is available at https://github.com/computational-cell-analytics/micro-sam and documented at https://computational-cell-analytics.github.io/micro-sam/micro_sam.html. This includes installation instructions with all required dependencies. |
| Data analysis | We rely on the scientific python tools for analysis of our experiment results and plotting, in particular numpy (1.26), pandas (2.1) and matplotlib (3.8). The version numbers given in parenthesis are for the main python environment where the analysis was run. Note that these software packages are stable so the analysis output is expected to be stable across versions. |

For manuscripts utilizing custom algorithms or software that are central to the research but not yet described in published literature, software must be made available to editors and reviewers. We strongly encourage code deposition in a community repository (e.g. GitHub). See the Nature Portfolio guidelines for submitting code & software for further information.

## Data

Policy information about availability of data

All manuscripts must include a data availability statement. This statement should provide the following information, where applicable:

- Accession codes, unique identifiers, or web links for publicly available datasets
- A description of any restrictions on data availability
- For clinical datasets or third party data, please ensure that the statement adheres to our policy

Our software is available on github under a permissive open source license at
https://github.com/computational-cell-analytics/micro-sam. It is documented at
https://computational-cell-analytics.github.io/micro-sam/micro_sam.html. The version at
submission of this manuscript is 1.1.1. Our LM and EM generalist models are available on
BioImage.IO and Zenodo. Please refer to our model documentation for the ids and dois of the
individual models. Additional models are deposited on Zenodo and the corresponding links are
given in the documentation. The tables and code for generating quantitative plots are available on github. We
make use of publicly available datasets for most experiments. They are listed in Supp. Table 1
and 2. We use new datasets for the 2D and 3D annotation user studies. These datasets are avaialble on zenodo,
please see the data availability section for their DOIs.

## Human research participants

Policy information about studies involving human research participants and Sex and Gender in Research.

| | |
|---|---|
| Reporting on sex and gender | NA |
| Population characteristics | NA |
| Recruitment | NA |
| Ethics oversight | NA |

Note that full information on the approval of the study protocol must also be provided in the manuscript.

# Field-specific reporting

Please select the one below that is the best fit for your research. If you are not sure, read the appropriate sections before making your selection.

☒ Life sciences        ☐ Behavioural & social sciences        ☐ Ecological, evolutionary & environmental sciences

For a reference copy of the document with all sections, see nature.com/documents/nr-reporting-summary-flat.pdf

# Life sciences study design

All studies must disclose on these points even when the disclosure is negative.

| | |
|---|---|
| Sample size | We perform different kinds of experiments, with different definitions of sample size. To evaluate segmentation algorithms on existing datasets we use approximately 30 different datasets. Here, a larger size is desirable to provide a coverage of relevant experimental settings, but requires additional experimental and computational effort. In the user study we provide a multi annotator study with 5 subjects to investigate inter annotator effects. There are no clear statistical criteria for the number of annotators, five was the number we could recruit and coordinate among the co-authors. |
| Data exclusions | We did not exclude any data from experiments. |
| Replication | We performed replications for one of the segmentation evaluation experiments (Extended Data Figure 1) to investigate the effect of randomness in the interactive segmentation procedure. We tested this for five independent replications. |
| Randomization | Randomization of experiments is not applicable for our study. |
| Blinding | Blinding of experiments is not applicable for our study. |

# Reporting for specific materials, systems and methods

We require information from authors about some types of materials, experimental systems and methods used in many studies. Here, indicate whether each material, system or method listed is relevant to your study. If you are not sure if a list item applies to your research, read the appropriate section before selecting a response.

## Materials & experimental systems

| n/a | Involved in the study |
|-----|----------------------|
| ☒ | Antibodies |
| ☒ | Eukaryotic cell lines |
| ☒ | Palaeontology and archaeology |
| ☒ | Animals and other organisms |
| ☒ | Clinical data |
| ☒ | Dual use research of concern |

## Methods

| n/a | Involved in the study |
|-----|----------------------|
| ☒ | ChIP-seq |
| ☒ | Flow cytometry |
| ☒ | MRI-based neuroimaging |

