## [Peer Review File · Nature Methods]

Segment Anything for Microscopy

Corresponding Author: Professor Constantin Pape

Version 0:

Decision Letter:

26th Oct 2023

Dear Constantin,

Your Article, "Segment Anything for Microscopy", has now been seen by 3 reviewers. As you will see from their comments below, although the reviewers find your work of considerable potential interest, they have raised a number of concerns. We are interested in the possibility of publishing your paper in Nature Methods, but would like to consider your response to these concerns before we reach a final decision on publication. We therefore invite you to revise your manuscript to address these concerns.

Generally speaking, we thought the concerns raised by the referees were reasonable. We ask that you focus your revision on providing convincing benchmarking relative to other tools (and explaining why your method may sometimes underperform relative to generalist models), clarifying computational costs/runtimes, improving installation/user guides, and either making a stronger case that works well for EM or removing those claims and limiting the paper to optical imaging.

Please note, while we think the majority of concerns can be addressed, we cannot promise to send the paper back to reviewers until we've seen the new data.

Link Redacted

We hope to receive your revised paper within three months. If you cannot send it within this time, please let us know. In this event, we will still be happy to reconsider your paper at a later date so long as nothing similar has been accepted for publication at Nature Methods or published elsewhere.

OPEN SCIENCE REQUIREMENTS

REPORTING SUMMARY AND EDITORIAL POLICY CHECKLISTS

DATA AVAILABILITY

All novel DNA and RNA sequencing data, protein sequences, genetic polymorphisms, linked genotype and phenotype data, gene expression data, macromolecular structures, and proteomics data must be deposited in a publicly accessible database, and accession codes and associated hyperlinks must be provided in the "Data Availability" section.

CODE AVAILABILITY

Please include a "Code Availability" subsection in the Online Methods which details how your custom code is made available. Only in rare cases (where code is not central to the main conclusions of the paper) is the statement "available upon request" allowed (and reasons should be specified).

MATERIALS AVAILABILITY

ORCID

Nature Methods is committed to improving transparency in authorship. As part of our efforts in this direction, we are now requesting that all authors identified as 'corresponding author' on published papers create and link their Open Researcher and Contributor Identifier (ORCID) with their account on the Manuscript Tracking System (MTS), prior to acceptance. This applies to primary research papers only. ORCID helps the scientific community achieve unambiguous attribution of all scholarly contributions. You can create and link your ORCID from the home page of the MTS by clicking on 'Modify my Springer Nature account'. For more information please visit <http://www.springernature.com/orcid>.

Sincerely,
Rita

Rita Strack, Ph.D.
Senior Editor
Nature Methods

Reviewers' Comments:

Reviewer #1:

Remarks to the Author:

A. Summary of the key results

In their paper titled "Segment Anything for Microscopy," the authors introduce a software project that extends the capabilities of the original Segment Anything Model (SAM, Kirillov et al., 2023) in several ways. They accomplish this by:

1. Fine-tuning for specific microscopy datasets: The authors have fine-tuned SAM for various public microscopy datasets, including both Light Microscopy (LM) and Electron Microscopy (EM) public datasets.
2. User-friendly Interface: They have developed a user interface based on napari, enabling the interactive use of the fine-tuned models on new images.
3. 3D image Segmentation and cell tracking: The authors have extended SAM's capabilities to segment 3D images and perform cell tracking on sequences of 2D images.

Building on the original SAM framework, the authors employ bounding boxes as prompts, which consistently outperform point annotations, as previously established in the SAM paper. They further evaluate the performance of the default SAM model, trained on a large dataset of natural images, against specialized models fine-tuned on specific EM or LM datasets and generalist models trained on various EM or LM datasets. In general, their results indicate that both fine-tuned models consistently match or surpass the performance of the default model when using the same prompts. For LM datasets, they also benchmark their results against Cellpose, another generalist method for cell segmentation using deep learning, with the fine-tuned models frequently outperforming a pretrained model provided within Cellpose.

Additionally, the authors provide an analysis of their interactive annotation tool for 2D and 3D segmentation, as well as for tracking tasks. Across all cases, the use of fine-tuned models accelerates the annotation process while maintaining or even improving performance compared to competing methods, such as Cellpose for 2D segmentation, Ilastik for 3D segmentation, and TrackMate in combination with Cellpose for tracking.

In summary, this paper represents one of the first efforts to adapt the SAM framework to the domain of bioimage analysis. Through fine-tuning on major public datasets and the provision of user-friendly tools and libraries, the authors have expanded the applicability and accessibility of SAM for bioimage analysts.

B. Originality and significance

The authors have maintained the core architecture of SAM while retraining it with custom microscopy datasets, using an object-by-object training approach inspired by the original SAM paper. Consequently, the improved performance of the fine-tuned models compared to the original model is an anticipated outcome, given the models' training on more specific and relevant data. This aligns with findings reported in the Cellpose 2.0 paper (Pachitariu & Stringer, 2022), where it was

demonstrated that pretrained models only need a reduced subset of training data to achieve the same performance improvement as with the full training dataset.

The novelty of this work lies primarily in the software tools and utilities introduced to facilitate the integration of SAM into the field of EM and LM image analysis. Notably, the authors have introduced a graphical user interface (GUI) leveraging napari, which simplifies the usage of SAM by introducing interactive functionalities. These interactive tools, combined with a system for segmenting large 2D images by breaking them into patches and the ability to project 2D segmentation results onto 3D objects or perform tracking of 2D objects over time, are the key contributions of this work.

C. Data & methodology: validity of approach, quality of data, quality of presentation

The authors have conducted an extensive validation of their tool using a total of 22 publicly available datasets of both LM and EM images. Their evaluation involved benchmarking against the original SAM model, that serves as reference for comparison with their own fine-tuned models. Furthermore, for some LM datasets, the authors included two additional pretrained Cellpose models, namely, LiveCELL and cyto, in their assessments.

For the evaluation of segmentation results, the authors have chosen a single metric, initially defined as "segmentation accuracy metric" but also referred to as "segmentation quality" in their figures. While this simplifies the comparison of different methods within their manuscript, it may limit the direct comparability of their results with those from previously published articles.

Segmentation results are presented both quantitatively (based on their selected segmentation metric) and qualitatively. Qualitative results are often illustrated using single-point prompts, showcasing the performance of both default and fine-tuned specialized and generalist models. This is a bit surprising, since the analysis presented in the manuscript indicates a much better performance across all datasets and models while using bounding boxes as prompts.

The usability of the proposed interactive GUI is also evaluated by measuring average annotation times for 2D and 3D segmented objects, as well as the annotation of complete tracks within the tracking tool. Those times are compared with publicly available software alternatives, including human-in-the-loop Cellpose, Ilastik carving (which does not employ machine learning), and TrackMate (leveraging Cellpose segmentations).

D. Appropriate use of statistics and treatment of uncertainties

The manuscript relies on straightforward statistical measures such as averages and standard deviations of metrics to present the results. No complex statistical tests or equivalent methods for comparing the performance of various approaches are employed within the manuscript. This approach is worth mentioning as, throughout the text, the term "significant" is used somewhat broadly when describing improvements or differences between values.

It's worth noting that while the term "significant" is used, its usage may not necessarily imply statistical significance in the context of hypothesis testing. This aspect might benefit from clarification to ensure a more precise interpretation of the results. Additionally, considering the broad use of the term "significant," the inclusion of statistical tests or alternative methods for establishing the significance of observed improvements could enhance the robustness of the presented findings and provide stronger support for the reported results.

In the assessment of the annotation tools, the manuscript does not specify the number of annotators involved in the evaluations. Additionally, the presentation is based only on average values for objects per time or tracks per time. The inclusion of standard deviations within these values would enhance the comprehension of the tool's impact, providing insights into the variability and consistency of results achieved with each tool.

E. Conclusions: robustness, validity, reliability

The authors have introduced a method to fine-tune the original SAM model for microscopy datasets, demonstrating its utility across various datasets to enhance or maintain segmentation results achieved with the original model, particularly when using bounding boxes as prompts.

The inclusion of interactive annotation and tracking tools within napari represents a valuable contribution to the bioimage analysis community. These tools offer significant benefits, particularly for non-expert users, who can accelerate their annotation processes in a semi-automatic manner. However, there is room for improvement in the evaluation of these tools, as mentioned previously.

F. Suggested improvements: experiments, data for possible revision

From a methodological point of view, I recommend the following improvements:

1. Further documentation of the training process:

* It would be beneficial to provide a more detailed account of the training process. This could involve describing the alternative methods explored for fine-tuning, even if they did not yield successful results. Documenting these unsuccessful attempts can provide valuable insights for other researchers, contributing to the collective knowledge base.

* Moreover, some of the datasets used are 2D and some are 3D, but those details, together with their image sizes, the patch size used for training and their evaluation (2D or 3D), are missing. Including these specifics would provide readers with a more complete understanding of the experimental setup and facilitate reproducibility.

2. Display of training and validation plots:

* To enhance transparency and understanding of the training process, consider including training and validation plots for key metrics such as loss and Dice score. These plots can offer readers insights into the convergence behavior of models, helping them gauge when and how the models reached optimal performance.

3. Incorporation of additional evaluation metrics:

* Expanding the repertoire of evaluation metrics would enhance the comprehensiveness of the presented results. For instance, simply including metrics such as accuracy at various IoU thresholds (e.g., 0.3, 0.5, 0.7) can provide readers with a more detailed and informative perspective on model performance than just the mean segmentation accuracy.

4. Diversification of annotators and reporting standard deviations:

* In the analysis of the interactive tools, it is advisable to involve more than one annotator, ideally two or more, who are not affiliated with the `micro_sam` project. This diversified perspective can yield more robust insights into the differences between tools. Additionally, providing standard deviation values alongside the average annotation times for these annotators would offer a clearer understanding of the variations and consistency in tool performance.

From a presentation point of view, I recommend the following improvements:

1. In the figures displaying instance segmentation results, consider employing bounding box prompts. While the current use of single-point prompts may be more straightforward for visualization, they often correspond to the lowest segmentation quality in the results, potentially giving the impression of "cherry-picked" examples. For instance, in the MouseEmbryo dataset, single-point prompts yield segmentation quality values below 0.1 for both the default and generalist models, yet the results display a flawless segmentation for the generalist model. Employing bounding box prompts could provide a more representative and informative view of segmentation outcomes.

2. Consider including fully segmented results alongside single-instance examples. Presently, qualitative results are showcased for individual instances, yet many datasets contain densely populated areas of objects requiring segmentation.

3. The results of the analysis of interactive annotation tools require clarification, particularly regarding the measurement of time for each tool. There is ambiguity in how times are measured and reported. For example, the human-in-the-loop GUI of Cellpose 2.0 allows retraining the model based on newly introduced annotations after completing a full image. However, in this study, three images are annotated and then the model is updated "twice," which appears somewhat arbitrary and needs further explanation. Additionally, when reporting "annotation times for the default model and the mean over annotation times after fine-tuning," it's unclear whether these times include retraining, how many images are used (e.g., the 20 with ground-truth), and whether these factors affect the measurements. Similarly, in the "tracking user study," it remains unclear how times are measured and whether TrackMate is used interactively with Cellpose or in conjunction with a segmentation output from a pretrained model from the Cellpose Zoo.

4. Consider revising Figure 1a for improved clarity. The current representation may be unclear in terms of where the process begins and ends. A more explicit depiction would enhance the understanding of the workflow.

Regarding the experiments performed in this work, although the authors have conducted an extensive analysis with a very large number of datasets, their models have been specialized for LM or EM modalities exclusively. Given the manuscript's title, "Segment Anything for Microscopy," it raises questions about the potential effects of training SAM across different modalities and whether a more generalized foundation model for microscopy could be established. Experiments in that direction, maybe with tests on held-out datasets of various modalities or even on the 2018 Data Science Bowl dataset, could substantially increase the impact of this work.

Minor comments:

* On page 5, it would be beneficial to provide at least a brief definition of the segmentation metric used. Currently, it's referred to as the "segmentation accuracy metric" on page 5 and as "segmentation quality" in Figure 4, which can be confusing.

* On page 5, the "automated instance segmentation" method is mentioned but not explained. It would be helpful to provide some context or explanation for readers who may not be familiar with the original SAM paper. Additionally, maintain consistency in the naming, also referred to as "automatic instance segmentation" through the text.

* In Figure 1b, the caption should explain the meaning of colors, specifying what the red mask, yellow contours, and blue box represent.

* In the caption of Figure 2b, please correct the color discrepancy, "box annotations (blue)" should be replaced with "box annotations (red)". Additionally, there is a typo in the caption: "Vit-B" should be corrected to "ViT-B."

* In Figure 2, the acronym "amg" for automatic instance segmentation might be confusing without an explanation. Although it originates from the SAM paper, it should be clarified for clarity in this context.

* Ensure consistent writing of "Cellpose" throughout the text and figures, using the correct capitalization. Similarly, maintain "TrackMate" as the correct spelling instead of "Trackmate."

* In Figure 4a, the legend should not include "amg" since it is not present in those experiments.

* In the caption of Figure 4, it reads "We [...] do not evaluate the automatic instance segmentation, since this is ill-defined given segmentation tasks for specific objects". This requires further explanation in the text, since that approach is used for LM densely populated images.

- * In the results of the "2D user study," for completeness, consider including the segmentation metric values obtained with each approach. This information could be presented as supplementary material.
- * On page 15, the phrase "with versa" should be corrected to "vice versa."
- * Some descriptions related to implementation details, such as specific code-related terms or names of napari layers, might be too technical for a general reader. Simplify these explanations to improve readability.
- * Details about the linear motion model used for the projection of segments during tacking are missing. Please, add a short explanation or citation.
- * Consider reorganizing Figure 5 to present the measured times in tables alongside the plots and figures for improved clarity.
- * Clarify whether the IoU and stability thresholds applied for the automated segmentation experiments (Figures 2 and 3) were optimized using the mentioned grid search.

Comments about the code:

I have successfully installed and tested the project code on both a Linux system and a MacOS system. The GUI functions correctly, as mentioned in the manuscript, with occasional minor bugs, which are typical for a software application in its early development stages.

Regarding the documentation, I suggest the inclusion of a step-by-step tutorial designed to assist inexperienced users in navigating and using the software effectively. Additionally, if feasible, consider incorporating tooltip texts for each button within the GUI. These tooltips can provide brief explanations or hints to users, further facilitating their understanding and usage of the software. Such user-friendly enhancements would contribute to a more accessible and user-oriented software experience.

G. References: appropriate credit to previous work?

The manuscript appropriately acknowledges previous work through accurate and well-placed citations throughout the text.

H. Clarity and context: lucidity of abstract/summary, appropriateness of abstract, introduction and conclusions

The text of the reviewed paper exhibits a generally high level of clarity, ensuring that its content is accessible to readers. The language used is precise, technical, and appropriate for the scientific context. Furthermore, the paper maintains a logical and well-structured flow, making it easy for readers to follow the research methodology and findings. To further enhance its quality, attention to defining technical terms, providing contextual information, and offering more detailed explanations of the experimental setup would contribute to a more informative and well-rounded paper.

Reviewer #2:

Remarks to the Author:

Archit et al. present a tool for interactive cell segmentation and tracking, which is build upon the Segment Anything (SAM) approach proposed by Kirillov et al. 2023. The tool supports 2D, 3D and 2D+t annotation, with provided segmentation models being finetuned to light and electron microscopy image data. To facilitate processing of 3D and timeseries data, the interactive functionality is improved by propagating labels to adjacent layers/frames, thereby reducing annotation efforts. Furthermore, offering the ability to pre-compute image encodings allows for faster runtimes. The employed finetuning techniques helped to achieve results that surpass the results obtained with the default SAM model and the results obtained with state-of-the-art approaches used in microscopy data processing, such as Cellpose, Ilastik and Trackmate. Furthermore, the authors conducted various experiments analyzing different levels of segmentation instructions in form of manually provided point or box annotations. The results differ on different datasets, but generally a higher amount of instructions let to more accurate segmentation outcomes. All software is made available online and the tool comes with a convenient GUI, that can be easily set up and used. Although the idea to finetune the SAM approach is not fundamentally new, the proposed work includes multiple helpful additions for microscopy image processing and dedicated finetuned models, which emphasizes the potential usefulness for the microscopy image community. Nevertheless, I have some remarks and suggestions that are listed in the following.

MAJOR POINTS

- The order in which information is provided within the paper does sometimes lead to confusion. For example, "automatic instance segmentation" was referred to without beforehand giving an explanation of what it means. Furthermore, when referring to the metric used for evaluation of the segmentations results, an explanation is missing and the reader does not know if larger or smaller values are better. Some of these explanations are given in later parts of the paper, but would rather benefit the clarity of earlier sections. I understand that detailed explanations might indeed belong into a separate section, but low-level explanations might be beneficial to prevent that confusion and the necessity to skip ahead. Similarly, it was mentioned that tracking results were evaluated using the Cell Tracking Challenge metric, but a brief low-level explanation would benefit readability and again prevent the reader from skipping to the source/reference.
- It was mentioned that the point annotations were placed at the center of the object to mimic user behavior. However, for the example given in Figure 4 b top left, I would expect the user to put the point into the bulky part of the structure. This raises the question of how sensitive the outcomes are to different

positioning of the point annotations. Do changes have a significant impact on the results? Could that explain why sometimes giving less point annotations is beneficial (Fig. 2 various cases, Fig. 3a Lizard Generalist, Fig. 4a CREMI Default,...)? Maybe it could also explain why only the nucleus was segmented for the CovidIF data in Supp. Fig. 3?

- In my opinion, a more detailed explanation of the runtime evaluation given in Figure 5 would be beneficial. On which machine was this performed and is this comparable to common user hardware? How many different users were involved when doing annotations and could this induce a subjective bias due to different levels of expertise with the different tools? What did contribute to the calculation of the runtime: for example, the finetuned `micro_sam` was reported to have an average annotation time of 0.3 seconds, but does this also include providing user point/box annotations? Even 3.4 seconds for manual annotations seems to be pretty fast, assuming that accurate annotations were obtained.
- While testing the software, I experienced that the pre-computation of image encodings is a must in order to effectively use the tool. This is especially crucial for 3D and timeseries data. Although computation of encodings can be done beforehand in a fully-automated fashion, I think this is an important point when comparing the different approaches and should at least be explicitly mentioned when discussing runtimes. Other approaches like Cellpose don't have this requirement and can be directly applied, which is not necessarily a (dis-)advantage, but certainly important information to interpret results.
- Figure 4 a shows results for different EM datasets. When referenced in text, interpretations are no longer associated with the datasets, but with the structures they show (cells, neurites, nuclei, mitochondria). However, without knowing these datasets, it is hard to follow this interpretation, which is why I would recommend to keep a consistent description and either use the dataset name or the structure specification.
- Figure 5 seems a bit chaotic and could be visually improved. My main concern is that the different blocks (a,b,c) range into one another, which makes it harder to separate them and get a clear view of the different results.
- It seems unclear if there is a difference in processing RGB image data (e.g. histopathology) and grayscale image data. Does the approach take a 3-channel image as input for all cases or is there any pre-processing involved?

MINOR POINTS

- The abbreviation "amg" is not properly explained, especially at its first appearance, and hard to associate with "automated instance segmentation". An appropriate explanation or another abbreviation might improve clarity.
- The naming `micro_sam` seems a little out of place. I personally would have preferred the naming that was used in the repository/documentation: `\muSAM`.
- The following was mentioned regarding the evaluation of runtimes on page 12: "We then finetune the model on the segmentation results from interactive segmentation and also report the annotation times". It is unclear to me if this means that the same data was used for finetuning and testing? Using the same data would probably be the case in a practical scenario, but giving this information would still help in interpreting the results.
- In relation to the previous point, the term "semi-automatic approaches" was used. It would help readability if it would be explicitly mentioned which of those approaches are indeed semi-automatic.
- Supp. Figure 3: The naming "finetuned" and "generalist" seem to be interchangeable here. A more consistent description would improve clarity.
- Overall some sentences could be reformulated. Some suggestions for reformulations and typos are the following:
 - "To train the generalist, we assemble..." → "...generalist model,..."?
 - Maybe consider reformulating the following to improve clarity: "However, this difference is negligible compared to the performance improvement compared to the default model"
 - Supp. Fig. 3: "This" → "This"
 - "Supp. Figure 2 show" → "...shows"

SOFTWARE-RELATED POINTS:

- It would be helpful for annotation speed, if there would be a shortcut for switching between positive and negative prompts.
- Personally, I think placing point annotations is the fastest option. However, it appears that the segmentation process necessitates an iterative approach, meaning that it seems impossible to place multiple independent point annotations to process results for multiple cells collectively and then initiate the processing in one go (unlike for box annotations). Unfortunately, this can significantly extend the runtime when using the tool in practical scenarios. For example, in an image scene with several hundred cells, it would entail a substantial number of manual interactions and clicks. Ideally, it would be beneficial to be able to place common negative prompts and independent positive prompts, even for very dense scenes of

touching cells or scenes showing cellular membranes.

• Placing multiple box annotations at once before initiating the segmentation process does not seem to work for 3D image data. Unless it only was a bug, adding this feature would significantly improve the practical usability for 3D image data.

Reviewer #3:

Remarks to the Author:

The manuscript titled “Segment Anything for Microscopy” (uSAM from now on) by Archit et al describes a computational tool based on the Segment Anything Model (SAM) released recently by Meta << <https://doi.org/10.48550/arXiv.2304.02643> >>. Vision transformers in SAM – as opposed to convolutional approaches – have shown exciting results for segmentation tasks, so there is merit in extending SAM to segment features in microscopy datasets. Further, with SAM one can load “embeddings” of the model and then produce image annotations very rapidly. This is a little like having a loaded gun ready to go and “pulling the trigger” for any instance of, say, a cell in an image, will produce instantaneous segmentation on that image. Essentially the user experience would boil down to – click on, or box out, a feature, and bam: the feature is accurately segmented out.

This would be a powerful tool for users, however, for a variety of reasons, we feel that this work is not ready for prime time, and it may merit consideration for publication at Nature Methods only after a substantial revamp. The annotator GUI itself is a useful tool (perhaps it can be used in conjunction with other tools) and it has the advantage of being “first out of the gate”, but there are four main drawbacks:

- a. The tool does not work well for electron microscopy data
- b. Shortcomings inherent to the SAM limit the usability of this tool
- c. The comparisons made to benchmark uSAM performance flatter to deceive
- d. The installation was not easy and cannot be done by an end user

Here we describe our experience with uSAM.

We used Linux [Ubuntu 18.04LTS] env with 256Gib RAM, 12 CPU, 4 1080Ti GPUs – a reasonably beefy server setup (i.e. above what a user + laptop would use, but not the top-of-the-line A100 GPUs that the authors have).

- a. The given anaconda installation steps are very slow for a linux environment and could not be performed without significant troubleshooting, meaning that this tool can only be installed by computational analysts, not by end users.
 - i. Various individual dependencies had to be individually installed from source (not just napari, pyqt, but also SAM and torch-em)
 - ii. Installation failed because of ZARR and Vigna (an odd and exotic dependency, we urge authors to find an alternative).
 - iii. Finally installed using Linux bash script, which is still in dev
 - iv. Training/finetuning should allow multi GPU support
- b. In fact when troubleshooting, we noticed that the github repo was updated in the past days several times, meaning that this is a project still in development. We urge the authors to upload a fully stable linux/PC/mac compatible release on Zenodo as a v1. We understand that future iterations will follow, but a stable v1 is a must.

Now with regards to the approach.

- a. We note that the approach of “fine-tuning” a SAM to a specific domain is not novel – it has been done before. [<https://doi.org/10.48550/arXiv.2304.13785>]
- b. That means that the authors’ work was to train the SAM model on a handful of LM and EM datasets, update weights, and then release an annotation tool that allows rapid segmentation based on this new uSAM model. Here are three issues with this:
 - b.i. We agree that hard work has been done, but from a computational point of view, there really isn’t much that’s new or novel (see Pt 2a).
 - b.ii. The Achilles heel of this approach is that WYSIWYG – the “generalist uSAM” is meant to be used as-is, further retraining on LM or EM data is very slow, taking days/weeks even on A100 GPUs, and even longer on weaker systems.
 - b.iii. Running uSAM on 3D data or stacks is extremely slow. We recorded ~ 8s per image for a 1k x 1k image, meaning that even modest image stacks require hours to load the model embeddings. Note that even though appropriate cuda libraries are installed it used CPUs.
- c. The SAM approach requires that the user clicks on various features to get a segmentation. Yes, this makes segmentation easy, but it also means that for n instances in m images, the user will have to click n x m times minimum. This is a burden which is not required for models such as CellPose.
- d. There’s another issue as far as we could tell – for multiple instances of the same class (say n cells in an image) clicking multiple points does not work, you have to box out instances. And this fails when you have convoluted structures where one cell “envelops” another, so the boxes completely overlap.
- e. Overall, unfortunately the time and compute power required to finetune the uSAM models is such a massive stumbling block (the authors should clarify exact numbers on resources/time needed for fine-tuning to the point of getting a useful model), it just renders this approach pretty inflexible.

3. Regarding the data presented:

- a. Some plots of the model training must be included (loss over iterations, hyperparameter grid search, accuracy over iterations)

- etc). Otherwise it appears as if the authors simply chose a set of parameters and went with it.
- b. The quality of the segmentation output (and if there was manual cleanup required) must be visually compared with other models such as CellPose and StarDist.
 - c. The Y axis must be set at 1.0 across all graphs, otherwise it is misleading
 - d. Which CellPose model was used for comparison? Since there is for example CellPose for TissueNet, why aren't the authors using that to compare?
 - e. The authors should re-label "segmentation quality" to "mean segmentation accuracy" (if that's what it is)
 - f. What does "in-loop finetuning" mean, is it about finetuning SAM model with user annotation? because we can't see any finetuning implementation in the annotator GUI.

Specific Figure notes:

Figure 3 shows that uSAM can beat CellPose at a magic number of 12 prompts (8n + 4p) but very strangely the specialist model does not perform better than the generalist in some cases (why?). Further, only the LiveCELL is a true apples-to-apples comparison of models' strength. But again, CellPose is then good to go out of the box, whereas uSAM still needs all those user prompts or seeds for annotating. And fine-tuning is onerous for uSAM (and again even then you need the seeds!) Finally, when y axes are set to 1.0, the results are revealed to be modest overall.

Figure 4 reveals the weakness of uSAM with EM data – it basically does not generalize (and automated annotation amg is SQ=0?)

Figure 5. The CellPose numbers are missing from the graph, and the 3D results must be compared also (e.g., against StarDist)

In all cases, the images presented appear to be "best case" segmentation, and we got far worse results in our hands unfortunately.

4. The "generalist model" doesn't work well for EM! To be fair, the authors are modest about their claims with EM data, but even for a basic task – segmenting neurites and cells from their own shared dataset – uSAM failed badly on stacks of images <<see attached >>

a. One limitation is that for volume EM datasets, running inference purely in the imaging plane gives the well-known "pancake artifact". We suggest the authors to run inference in all three planes, which should improve results <<REF Conrad and Narayan 2023 <https://doi.org/10.1017/S143192762002053X> >> and they should show results.

Overall, this is a commendable effort, but either the authors must drop claims of electron microscopy and just release the LM uSAM (in which case the model does not reliably perform favorably with CellPose), or alternatively, the authors must train extensively on EM data to get a better performing uSAM for EM data.

Regards,

Abhishek Bhardwaj & Kedar Narayan
Center for Molecular Microscopy
Frederick National Lab, NCI, NIH

Version 1:

Decision Letter:

26th Jun 2024

Dear Constantin,

Thank you for your letter detailing how you would respond to the reviewer concerns regarding your Article, "Segment Anything for Microscopy". We have decided to invite you to revise your manuscript as you have outlined, before we reach a final decision on publication. We ask that you update the main text of the paper to outline where the SAM fits into the current tool space as you so nicely did in your summary. We also ask that you add in the technical details requested by the reviewer.

Regarding the remaining concerns from reviewer 3, we think your responses are appropriate. In all cases, please add relevant discussions to the Discussion section.

Link Redacted

We hope to receive your revised paper within six weeks. If you cannot send it within this time, please let us know. In this event, we will still be happy to reconsider your paper at a later date so long as nothing similar has been accepted for publication at Nature Methods or published elsewhere.

OPEN SCIENCE REQUIREMENTS

REPORTING SUMMARY AND EDITORIAL POLICY CHECKLISTS

DATA AVAILABILITY

CODE AVAILABILITY

Please include a "Code Availability" subsection in the Online Methods which details how your custom code is made available. Only in rare cases (where code is not central to the main conclusions of the paper) is the statement "available upon request" allowed (and reasons should be specified).

MATERIALS AVAILABILITY

ORCID

Sincerely,
Rita

Rita Strack, Ph.D.
Senior Editor
Nature Methods

Reviewers' Comments:

Reviewer #1:

Remarks to the Author:

I believe the new version of both the manuscript and its associated software application (and documentation) have improved much since its initial submission of this work. In particular, the new version of microSAM contains a new option for automatic instance segmentation (AIS) that requires fine-tuning on annotated data and everything is now implemented as a napari plugin, facilitating its use. Moreover, the authors now provide more insights about the training and fine-tuning of the model, as well as more details about the limitations of their approach.

Regarding the initial review of the manuscript, the authors have addressed most of my comments and suggestions in a satisfactory manner. Nevertheless, I still have some concerns about a few of the points I raised:

1. In the evaluation of the annotation tools, I believe it would be very positive for the paper to include (1) the average annotation times and their standard deviation of several annotators at least for a few images of the 2D annotation user study (they are only 20 in total), (2) the rest of important times on each experiment (embedding calculation, retraining/fine-tuning) and (3) the achieved segmentation accuracy in each experiment. This information is crucial for the potential users to decide on which tool to use.

2. In the information about the datasets used (Supplementary Tables 1 and 2) I would include their size in total number of GB but also their image dimensions. This will help the readers understand the scale of the experiments made.

3. Regarding my previous comments about the specialization of microSAM on LM and EM modalities, I didn't mean the number of datasets wasn't large enough. I was pointing out that the very first impression a reader gets when seeing the title of the manuscript (Segment Anything for Microscopy), or at least that was my case, is that we will be presented with a foundation model for segmentation of microscopy images. However, the paper jumps directly into creating generalist or specialist models per image modality. Following that line of thought, one has to wonder if it is possible to have a generalist model for different modalities, but there is no experiment in that direction. My suggestion about using the test set of the DSB dataset was double: on the one hand, it's a very diverse dataset in terms of modalities, and, on the other hand, it would allow direct comparison with state-of-the-art methods applied in the same dataset.

Related to this last comment, I do believe the manuscript would benefit from better defining its objectives and reach. As I see it, microSAM is a powerful software application that exploits SAM to create specific solutions for the instance segmentation and tracking of microscopy images. While this could seem obvious, along the text, microSAM is referred to in a sometimes ambiguous way such as "a tool for interactive and automatic segmentation and tracking" (singular), "tools for interactive and automatic segmentation and tracking" (in plural), "a napari plugin for interactive annotation and automatic segmentation", an improvement of "SAM for microscopy data [that] implements data annotation tools", an extension of "SAM for more efficient automatic instance segmentation", etc. I think this is understandable after so many changes in the manuscript and the application itself, but I'm sure the readers would appreciate it if this aspect is clearly stated. Maybe at the beginning you can

briefly summarize the structure of your software, with its backend (accessible library) and frontend (GUI in the form of a napari plugin).

Specific comments about the current version of the manuscript:

- * I would rewrite the abstract taking my comments above into account. There is some repetition and ambiguity (tool vs tools) about the definition of microSAM itself.
- * Figure 1a is a bit confusing, since we don't see very well where the model is. Also, the caption itself talks about different models, a napari plugin and speed ups, but none of that is depicted in the figure. I really like the logo though!
- * In the part where you describe your contributions, I would rewrite the text to clearly state (1) if the training method you implement is novel, (2) that you extend the *architecture* of SAM with a new encoder, and (3) that your software application includes a frontend in the form of a napari plugin and pretrained models ready to use. Again, please refer to my comment above about properly defining microSAM and its reach.
- * From the results you presented, one would conclude that not using the previous mask prediction as an additional prompt to the model is an important limiting factor for the segmentation results of the interactive segmentation method (Figure 2a, left vs right). I understand that keeping track of the previous mask of each annotated object could be an implementation challenge, but could it be less of a hurdle than fine-tuning the whole model? The former could be seamlessly executed in CPU, while the latter requires a GPU to be used in a real scenario.
- * In all the experiments using fine-tuned models (generalists or specialists) in Figures 2, 3 and 4, it's unclear what prompts are used to extract the objects under the I_P and I_B frameworks. I understand the training was performed with 7 iterations of point/box annotations but, is it also the case for testing? Or a single prompt is used in that case?
- * Regarding the AIS method proposed here, which is frequently the best performing method among the automatic ones presented in the paper, one has to wonder if a lighter U-Net-like convolutional model predicting the same channels wouldn't perform in an equivalent or better way when fine-tuned in the same images. This is an important question for a potential user interested in using (or not) microSAM for the automatic instance segmentation of their data.

Minor comments:

- * Page 1, Main text: dimensionality => dimensionalities.
- * Page 1, Main text: I would move the paragraph mentioning Cellpose 2.0 to the later part where you describe the state of the art limitations.
- * Page 2, Main text: The sentence "Vision foundation models have recently been introduced for image analysis tasks in natural images, echoing developments in natural language processing" requires being supported by a citation.
- * Page 3, Results text: To avoid confusion, since both specialist and generalist models are "finetuned" models, do not use "finetuned" to refer to the specialist ones. Instead, I would stick to specialist and generalist, which are very telling.
- * Page 4: I would rename the section "Finetuning Segment Anything improves cell segmentation for phase-contrast microscopy" to "Finetuning SAM improves cell segmentation for phase-contrast microscopy", since "Segment Anything" alone is an ambiguous term.
- * Page 4: Where you state that SAM "predicts an object mask based on point or box inputs", I would be more exact and describe all possible prompts to SAM.
- * Page 4: You state that the AMG method does automatic segmentation "segmenting all points with SAM". While I understand what you mean, I would rephrase that part of the text to avoid confusion about "segmenting points" to state each point in the grid is used as an individual prompt.
- * Page 4: I would rewrite the sentence "We evaluate SAM finetuning and different finetuning settings on LIVECell25". Apart from the typo ("finetuing"), it's not clear what you mean by evaluating SAM finetuning and finetuning settings. I assume you mean you evaluated different ways of fine-tuning SAM.
- * In Figure 2a: One would expect to see a line too in the right plot for a fine-tuned version of Cellpose, the same way AIS is a somehow fine-tuned version of AMG (with the new encoder).
- * In Figure 2a: The caption implies the right plot "shows the mean segmentation accuracy for interactive segmentation" but, if I understood correctly from the text, the right plot shows the accuracy results at each iteration of the fine-tuning process. This is different from interactive segmentation since previous masks are reused on each iteration, clearly benefiting the final result. This aspect should be clearly stated. Moreover, it's not clear if the results associated with that plot come from the training set or a held-out test set.
- * Page 9: The use of the dataset NucMM (X-Ray images) seems quite surprising given the section is about EM. Maybe justify its inclusion here.
- * Page 9: Both datasets MitoEM and NucMM contain official test partitions in their corresponding challenges. If the evaluation is performed on those sets, results could be submitted to each official challenge so they can also be compared with the current state-of-the-art specialized methods.
- * Page 11: The text about using micro SAM in resource constraints is somehow confusing. First, it says that you "compare the inference times for all relevant operations: computing image embeddings, inference for one object with a box or point annotation and automatic segmentation via AMG and AIS, for CPU and GPU" (related to Figure 5a), and then you say that you "use pre-computed image embeddings for Point, Box, AMG and AIS measurements". Point and Box measurements are in the same plot and I understand the embeddings computation time is not included. However, the "Time per Image" plot compares Embeddings, AMG and AIS times as if they were independent, although AMG and AIS do need the embeddings calculation in order to work. Also, the way time is measured for these three options (Page 13) seems a bit arbitrary (the minimum value of the average over 10 images after 5 repetitions). Showing the average time values with the standard deviations would be much more informative.
- * Page 13: The sentence "Training on the CPU in this setting took 5.3 hours, training on the GPU ca. 30 minutes" is repeated, that very same information is in the caption of Figure 5. I'd remove it from the caption.
- * Page 14: In the "User study 1", is there any reason to fine-tune the default SAM model and not the generalist LM model? This seems a bit counterintuitive, since one would expect the generalist model to be finetuned easier, given it has already seen similar data.

- * Figure 6: I would include the final metric reached with each method in each of the user studies. In the same figure, I'd include (1) qualitative results of more of the methods used in each study, maybe zooming over areas to see better the differences, and (2) other related computation times (embedding calculation for instance) in the corresponding tables.
- * Page 18, Limitations: The text reads "While our methods and tools provide versatile and powerful functionality for interactive and automatic microscopy segmentation, it has some limitations". Again this requires rewriting to better specify microSAM as a single application (not methods and tools).
- * Page 20, Automatic Instance Segmentation: Please cite the corresponding watershed method used.
- * Page 28, User Study: The sentence "The automated segmentations are of high quality, resulting in minimal correction effort." could be supported by some numbers (i.e., how many errors in total or per image).
- * Page 30: When commenting about the different TRA scores of each method, it would be nice to give an intuition of what those numbers mean in practice, i.e. missing tracks or extra tracks.

In summary, as I mentioned at the beginning of my review, I believe the manuscript and its associated software have really improved from their previous versions. However, apart from providing more information about the different results and addressing the questions I pointed out, I really think the paper would benefit from (1) a better definition of the software itself and (2) a clear message of its immediate application and reach. In my opinion, as it is right now, microSAM is a very useful tool for semi-automatic/interactive annotation, but it's difficult to see it competing with the current state-of-the-art models for fully automatic instance segmentation. Notice I'm not saying it has to compete with them, but it would be better to emphasize that aspect in the text. Moreover, it needs a bit of work to make it faster and more user-friendly so it gets accepted by the large community of non-expert users in need of such methods.

Reviewer #2:

Remarks to the Author:

I thank the authors for responding to all of my remarks and improving the code repository. I am happy to recommend publication.

Reviewer #3:

Remarks to the Author:

Summary of Key Results:

The authors present Segment Anything for Microscopy (μ SAM), a tool for interactive and automatic segmentation and tracking of objects in multi-dimensional microscopy data. The key results are:

Finetuning the Segment Anything Model (SAM) on microscopy datasets significantly improves segmentation quality across various imaging modalities like light microscopy (LM) and electron microscopy (EM).

They provide generalist models for LM and EM (for mitochondria and nuclei) that outperform the default SAM and achieve comparable or better performance than specialized tools like CellPose and MitoNet.

They implement tools for interactive annotation, automatic segmentation, and tracking within the napari plugin, enabling efficient data annotation compared to established tools.

User studies demonstrate the utility of μ SAM for organoid segmentation, 3D nucleus segmentation in EM, and nucleus tracking in fluorescence microscopy.

They investigate finetuning and inference in resource-constrained settings, making μ SAM accessible on consumer hardware.

Originality and Significance:

The application of vision foundation models like SAM to microscopy data is a novel and significant contribution. While some prior works have explored SAM for specific biomedical applications, this work presents a comprehensive framework for finetuning, interactive annotation, and automatic segmentation across multiple microscopy modalities. The generalist models and user-friendly tools enable widespread adoption of these techniques in the bioimaging community.

Data & Methodology:

The authors employ a rigorous methodology for finetuning SAM on diverse microscopy datasets, including iterative training schemes and architectural modifications like the automatic instance segmentation (AIS) decoder. The use of publicly available datasets and internal datasets for user studies ensures the validity of the approach and the quality of data.

The presentation of the methodology, including training details, evaluation metrics, and tool implementation, is clear and comprehensive, enabling reproducibility.

Appropriate Use of Statistics and Treatment of Uncertainties:

The authors use appropriate evaluation metrics like mean segmentation accuracy and the tracking metric from the Cell Tracking Challenge. They also investigate the influence of randomness in the interactive evaluation scheme and report standard deviations (Supp. Fig. 1b).

Conclusions:

The conclusions drawn from the results are robust and well-supported by the extensive quantitative and qualitative evaluations. The authors acknowledge the limitations of their approach, such as the larger computational footprint compared to CNN-based methods and the lack of a single model for all microscopy domains.

Suggested Improvements:

Exploring more efficient architectures or parameter-efficient finetuning techniques could further reduce the computational cost and enable real-time interactive segmentation on consumer hardware.

Investigating semantically aware models or training procedures to handle ambiguous segmentation cases in EM and enable a unified model for multiple domains could be a valuable future direction.

Implementing automatic tracking based on the initial frame-by-frame segmentations from AIS could further improve the tracking annotation tool's efficiency.

References:

The authors provide appropriate credit to previous work by citing relevant publications throughout the manuscript. The references cover foundational works on vision transformers, the original SAM paper, and related works on biomedical applications of SAM. The datasets used are also properly referenced.

Clarity and Context:

The abstract and summary provide a clear and concise overview of the work, highlighting the key contributions and findings. The introduction effectively sets the context and motivation for applying vision foundation models to microscopy data analysis. The conclusions summarize the main results, acknowledge limitations, and outline future directions, providing a well-rounded perspective on the work's significance and potential impact.

Version 2:

Decision Letter:

Our ref: NMETH-A53595B

8th Oct 2024

Dear Constantin,

Thank you for submitting your revised manuscript "Segment Anything for Microscopy" (NMETH-A53595B). It has now been seen by the original referees and their comments are below. The reviewers find that the paper has improved in revision, and therefore we'll be happy in principle to publish it in Nature Methods, pending minor revisions to satisfy the referees' final requests and to comply with our editorial and formatting guidelines.

Please provide a point-by-point response to the remaining concerns when you resubmit.

TRANSPARENT PEER REVIEW

ORCID

Sincerely,
Rita

Rita Strack, Ph.D.
Senior Editor
Nature Methods

Reviewer #1 (Remarks to the Author):

I would like to thank the authors for their extensive efforts in improving both the manuscript and the accompanying tool. The revisions have made the text clearer, and it now better reflects the true contributions of the work, while also acknowledging its limitations and future directions. However, I have a few comments based on both my observations and the authors' responses:

* Regarding the DSB Nuclei dataset, I had assumed that the training was carried out exclusively on the training and validation partitions, which is why I suggested providing results from the test partition. Additionally, the number of images listed in Supplementary Table 1 for that dataset (497) does not match the "official" count from the BBBC (<https://bbbc.broadinstitute.org/BBBC038>). Please clarify this discrepancy in the text.

* I agree with the new terminology ("tool" and "widgets") for describing your software. However, there are still some places in the manuscript where corrections were missed (e.g., on page 3 it still reads "In summary, our tools [...]"). Please review the text to ensure consistency throughout.

* I appreciate the improvements to Figure 1, but you might want to adjust the placement of the "Napari Plugin" box to center it for better clarity.

* Concerning the impact of using the previous predicted mask as a prompt for subsequent predictions, I understand from your results (Supplementary Figure 1) that the effect is negligible in a fine-tuning context compared to an interactive context with a frozen model. My earlier comment about the caption of Figure 2a (apologies for the lack of clarity) was intended to address the separation between the effect of adding more prompts (which I see as interactive segmentation) and fine-tuning. As presented, you seem to perform both fine-tuning and adding more points ("iterations") simultaneously. I suggest clarifying this distinction, as I would expect a fine-tuned but frozen model to improve with additional points, especially when using masks.

* I remain unconvinced about including NucMM in the EM section, but at the very least, please provide justification for this inclusion in the manuscript, as you did in the rebuttal.

* Finally, in Figure 6b and 6c, the evaluation metrics for each method are missing. Please add these for completeness.

Reviewer #1 (Remarks on code availability):

My students and I have reviewed the code. The software is still far from being user-friendly, it requires a GPU for better use and important information is somehow hidden in the terminal. However, it has great potential.

Reviewer #2 (Remarks to the Author):

Thanks for addressing all concerns that have been raised. I think the revised version of the script is again an improvement over the prior submissions. The unification of how the tool is addressed indeed increased readability and conciseness. Evaluation of different backbone networks adds more depths to the evaluation of the learning process and the required capacity of the network. The evaluations now feature not only a detailed assessment of segmentation accuracy, but also detailed insights into runtimes, which is equally, if not more, important for the user. I have only a few minor comments regarding the recent additions, and aside from these, I have no further concerns.

1. The number of images used in the user study with different annotators is somewhat limited, which offers a rough estimate for practical scenarios rather than a comprehensive evaluation. Nevertheless, I believe the results should be published as they are, given the clear effort and valuable insights provided. For future research with the tool, it could be worthwhile to consider expanding this aspect to provide an even more robust analysis.

2. The inclusion of dimensional information in Supplementary Tables 1 and 2 is a good addition. However, I would suggest providing clarification on the order of the dimensions listed. I assume the order is (z, y, x), but explicit confirmation would be helpful.

3. The hyperlink provided for "user study v3" did not work for me. Unless it was planned to only make it publicly available upon acceptance, this should be fixed.

Reviewer #2 (Remarks on code availability):

The tool's codebase is regularly updated, with growing amounts of information on usage and installation. I believe that this increasingly enhances accessibility for users across various disciplines, and I have no further comments.

Version 3:

Decision Letter:

26th Nov 2024

Dear Constantin,

I am pleased to inform you that your Article, "Segment Anything for Microscopy", has now been accepted for publication in Nature Methods. The received and accepted dates will be August 21, 2023 and Nov 26, 2024. This note is intended to let you know what to expect from us over the next month or so, and to let you know where to address any further questions.

Over the next few weeks, your paper will be copyedited to ensure that it conforms to Nature Methods style. Once your paper is typeset, you will receive an email with a link to choose the appropriate publishing options for your paper and our Author Services team will be in touch regarding any additional information that may be required. It is extremely important that you let us know now whether you will be difficult to contact over the next month. If this is the case, we ask that you send us the contact information (email, phone and fax) of someone who will be able to check the proofs and deal with any last-minute problems.

Please note that *Nature Methods* is a Transformative Journal (TJ). Authors may publish their research with us through the traditional subscription access route or make their paper immediately open access through payment of an article-processing charge (APC). Authors will not be required to make a final decision about access to their article until it has been accepted. [Find out more about Transformative Journals](https://www.springernature.com/gp/open-research/transformative-journals)

If you are active on Twitter/X, please e-mail me your and your coauthors' handles so that we may tag you when the paper is published.

Best regards,
Rita

Rita Strack, Ph.D.
Senior Editor
Nature Methods

Visit the Springer Nature Editorial and Publishing website at http://editorial-jobs.springernature.com?utm_source=ejP_NMeth_email&utm_medium=ejP_NMeth_email&utm_campaign=ejp_Nmeth or www.springernature.com/editorial

and-publishing-jobs for more information about our career opportunities. If you have any questions please click here.**

Open Access This Peer Review File is licensed under a Creative Commons Attribution 4.0 International License, which permits use, sharing, adaptation, distribution and reproduction in any medium or format, as long as you give appropriate credit to the original author(s) and the source, provide a link to the Creative Commons license, and indicate if changes were made. In cases where reviewers are anonymous, credit should be given to 'Anonymous Referee' and the source.

Screenshots for 3D cell segmentation Lucchi++ dataset

Response

We thank all three reviewers for their constructive and overall positive feedback. Based on the reviews and other feedback we have received about our preprint and our tool we have improved several aspects of our methods, our experiments and of our tools. The most important changes are the following:

1. We have improved automated instance segmentation by training a new decoder that predicts outputs for watershed-based instance segmentation, which we call AIS. It provides faster, and in most cases better, segmentation results than the automatic mask generation (AMG) approach from SAM. We have included this approach in all relevant quantitative experiments and also make it available in our tools, see also point 5.
2. We have improved the training algorithm for interactive segmentation. In particular, we found that whether the previous mask prediction of the model is used as an additional prompt during training has a big influence on the model's behavior. If this additional prompt is always given, as in the training algorithm described in Kirilov *et al.* and in our initial implementation, the model relies on it and will only improve segmentations consistently with more point prompts if the mask prompt is also provided. We have updated the training algorithm to use the additional mask prompt with a probability of 50%, improving the model for interactive segmentation without it. The updated training methodology is described in detail in the methods section. We have also added Supp. Fig. 10 to evaluate its impact. We have also updated the model evaluation to better account for this fact, see the next paragraph on changes in the manuscript.
3. We have investigated how to finetune models on user data in more detail. For this we have finetuned models on different hardware resources, especially on a CPU and on a consumer grade GPU, find parameters that allow training on these resources and report on the results. We find that finetuning on 10 or less annotated images can improve results and that it takes about 30 minutes on a consumer-grade GPU and about 6 hours on a CPU (the exact times may vary depending on the dataset). See Fig. 5, Supp. Fig 12 and 13 for details. We also measure the runtimes for applying the model for interactive and automatic segmentation for different encoder sizes, see Fig. 5 a.
4. We have redone the work related to EM. We now train a generalist model for mitochondrion and nucleus segmentation, by using the training data for mitochondria segmentation provided by the MitoLab / MitoNet publication. We find consistent performance improvements compared to default SAM and compare to MitoNet for automatic instance segmentation (see details for method comparisons below). We also investigate how these models work for other organelles (in particular ER) and neurites. See Fig. 4 and Supp. Figs. 5-8.
5. Our tools are now implemented as a napari plugin (they were implemented as “standalone tools” before). This enables integration with other napari plugins, for example to measure morphological features based on the segmented objects or to use them for retraining another neural network. Further, we have improved the user interface and added new features, in particular faster automatic instance segmentation thanks to AIS, automatic 3D segmentation and a GUI for model finetuning. In addition, we now

provide our latest LM and EM generalist models on [Biolmage.IO](https://biolm.io) to enable other tools supporting this community standard to use them.

We have thoroughly updated the manuscript to reflect the changes discussed above and to address several other comments raised by the reviewers. The most important changes in the manuscript, beside the points 1-5, are:

- We use a different methodology for evaluating the interactive segmentation results. Instead of randomly sampling a fixed number of points and/or a bounding box, providing it as prompts to the model and evaluating the result, we now use an iterative approach. It follows a similar logic as in training to simulate interactive correction of the model outputs (derived from errors in the model predictions when compared to the ground-truth). The results of this evaluation procedure better reflect the model's practical utility, and we found this especially important to evaluate the influence of the improved training algorithm, see also point 2 above. Consequently the evaluation experiments and associated figures have changed: they were redone with the new evaluation method and the new models that were trained with the improved training method. We explain the new evaluation in detail in the methods section.
- We are now more specific in the comparison to other methods for automatic instance segmentation: CellPose and MitoNet (which we have added for mitochondrion segmentation). We are not claiming that our models are superior for the respective automatic segmentation tasks, but rather that we provide results of similar quality in most practical settings, while providing a tool that combines **interactive** and automatic segmentation and that is applicable for a wider range of modalities. We also highlight the longer runtimes of SAM compared to these methods in the limitation paragraph, which we have added at the end of the discussion section.
- We include a new figure, Fig. 5, to show the results for inference runtimes of different SAM encoder sizes and for finetuning on a small dataset. The figure about the user studies is now Fig. 6. We have added more supplementary figures to provide further model evaluations, including for all sizes of the image encoder, to show qualitative instance segmentation results, to show 3D segmentation results and to further investigate the nuances of different training strategies.
- We have updated the user studies to make use of our new models and some of the new features we have introduced. For the tracking user study we have replaced the dataset, since the one used in the previous version of the manuscript is now part of the training dataset for our light microscopy generalist model. We now focus on measuring the annotation times and have removed the plots for quantitative measurements that were provided in the first manuscript. We show qualitative annotation results instead. We have also added a methods section that describes the user studies in more detail. Please note that each user study was undertaken by a single annotator familiar with the data and the tools used. While we agree that studying the effect of variation in annotation times depending on the annotator would be of interest, recruiting annotators that are familiar enough with the tools and data would be a large effort. We consider this to be out-of-scope for our contribution. It would be worthwhile to have a more systematic effort to benchmark the practical application of segmentation and annotation tools, but it is

beyond our current resources to do this. Ideally such an effort should be organized independently from tool developers or include several different tool developers. Nevertheless, we believe that the current user studies provide valuable insight into how our tools can be used in practice.

The changes to the manuscript are quite extensive, so we do not provide a version where they are highlighted. Please see the point-by-point response below for detailed answers on how we have addressed specific reviewer comments. Together with the resubmission of this manuscript we have made a 1.0 release of our tool. The annotation tool and core functionality is now stable. We have already seen some adoption of the tool, and believe that it will grow further with the stable version and improved documentation that we have published alongside it. While there are some limitations compared to existing tools for data annotation and segmentation (see also the new limitations paragraph), we believe that our tool offers the most versatile solution for microscopy segmentation currently available. We plan on further improvements to address these limitations and to provide more efficient finetuning, more general models etc. Since these improvements will not fundamentally change the functionality from a user perspective and due to the high utility of our tool for many applications we strongly believe that it is ready to be published in the current form.

Point-by-point response to the three reviews:

Reviewer 1: Thank you for the correct summary of our contribution and the generally positive assessment of our work.

For the evaluation of segmentation results, the authors have chosen a single metric, initially defined as “segmentation accuracy metric” but also referred to as “segmentation quality” in their figures. While this simplifies the comparison of different methods within their manuscript, it may limit the direct comparability of their results with those from previously published articles.

We now refer to this metric as “mean segmentation accuracy” in all figures. We have added more details on it in the methods section and also follow the reporting guidelines from Hirling *et al*¹, who provide best practices for segmentation evaluation.

Qualitative results are often illustrated using single-point prompts, showcasing the performance of both default and fine-tuned specialized and generalist models. This is a bit surprising, since the analysis presented in the manuscript indicates a much better performance across all datasets and models while using bounding boxes as prompts.

Using only point prompts in the original publication for the qualitative results was not intentional. We have updated all figures showing qualitative results and they now also show bounding box prompts, especially in Supp. Figs. 3, 4, 6 and 7. Note that the improvement of our models compared to default SAM is often more pronounced for point prompts, because the default

¹ <https://www.nature.com/articles/s41592-023-01942-8>

models often segment cell clusters or even the whole image when given a single point prompt, but we see a clear improvement for box prompts as well.

It's worth noting that while the term "significant" is used, its usage may not necessarily imply statistical significance in the context of hypothesis testing. This aspect might benefit from clarification to ensure a more precise interpretation of the results. Additionally, considering the broad use of the term "significant," the inclusion of statistical tests or alternative methods for establishing the significance of observed improvements could enhance the robustness of the presented findings and provide stronger support for the reported results.

Thank you for pointing this out. Indeed we did not mean to use “significant” in the strict statistical sense. We have replaced it with other appropriate terms in the revised manuscript.

In the assessment of the annotation tools, the manuscript does not specify the number of annotators involved in the evaluations. Additionally, the presentation is based only on average values for objects per time or tracks per time. The inclusion of standard deviations within these values would enhance the comprehension of the tool's impact, providing insights into the variability and consistency of results achieved with each tool.

We have added a method section on the user study to clarify these points. Briefly we had a single annotator that is familiar with the data, task and tools, see also the response above. Regarding the standard deviations: for two of the user studies (3d and tracking) we compute the times by annotating objects / tracks in a single volume / timeseries. We compute the annotation times by taking the time it takes to annotate all objects / tracks and then divide by the number of objects. For the 2d user study we measured the annotation time per image and used it to compute the time per object. For this reason we cannot easily give standard deviations. Note that we chose this approach to not introduce measurement errors that occur when stopping time for short intervals (such as annotating a single cell in 2d).

It would be beneficial to provide a more detailed account of the training process. This could involve describing the alternative methods explored for fine-tuning, even if they did not yield successful results. Documenting these unsuccessful attempts can provide valuable insights for other researchers, contributing to the collective knowledge base.

We have now extended the description of the training process in the methods section. This also includes the updated training strategy described in this response. We describe other training strategies we have experimented with, including their downsides in more detail too.

Moreover, some of the datasets used are 2D and some are 3D, but those details, together with their image sizes, the patch size used for training and their evaluation (2D or 3D), are missing. Including these specifics would provide readers with a more

complete understanding of the experimental setup and facilitate reproducibility.

Thank you for pointing out this oversight. We now list the patch shapes used for the different experiments in the methods section together with other important hyperparameters. We treat 3D datasets as a series of images during training.

To enhance transparency and understanding of the training process, consider including training and validation plots for key metrics such as loss and Dice score. These plots can offer readers insights into the convergence behavior of models, helping them gauge when and how the models reached optimal performance.

We have included this in Supp. Fig. 12 a.

Expanding the repertoire of evaluation metrics would enhance the comprehensiveness of the presented results. For instance, simply including metrics such as accuracy at various IoU thresholds (e.g., 0.3, 0.5, 0.7) can provide readers with a more detailed and informative perspective on model performance than just the mean segmentation accuracy.

We have not included additional evaluation metrics in the figures. While we agree that these would be informative we could not find a good strategy to include them while keeping the figures easy enough to understand. However, we provide the numerical data for all plots together with the plotting scripts online², so it would in principle be easy to redo these plots for different settings. Note that we make use of different IOU thresholds for some supplementary experiments where we found them to be more appropriate (see the methods section for details).

In the analysis of the interactive tools, it is advisable to involve more than one annotator, ideally two or more, who are not affiliated with the micro_sam project. This diversified perspective can yield more robust insights into the differences between tools. Additionally, providing standard deviation values alongside the average annotation times for these annotators would offer a clearer understanding of the variations and consistency in tool performance.

Please refer to the answers on this above.

In the figures displaying instance segmentation results, consider employing bounding box prompts. While the current use of single-point prompts may be more straightforward for visualization, they often correspond to the lowest segmentation quality in the results, potentially giving the impression of "cherry-picked" examples. For instance, in the MouseEmbryo dataset, single-point prompts yield segmentation quality values below 0.1 for both the default and generalist models, yet the results display a flawless segmentation for the generalist model. Employing bounding box prompts could provide a more representative and informative view of segmentation

² <https://github.com/computational-cell-analytics/micro-sam/tree/master/scripts/plotting>

outcomes.

We have updated these plots, see also comment above.

Consider including fully segmented results alongside single-instance examples. Presently, qualitative results are showcased for individual instances, yet many datasets contain densely populated areas of objects requiring segmentation.

We now include qualitative examples for automatic instance segmentation in Fig. 5 c and Supp. Fig. 11.

The results of the analysis of interactive annotation tools require clarification, particularly regarding the measurement of time for each tool. There is ambiguity in how times are measured and reported. For example, the human-in-the-loop GUI of Cellpose 2.0 allows retraining the model based on newly introduced annotations after completing a full image. However, in this study, three images are annotated and then the model is updated "twice," which appears somewhat arbitrary and needs further explanation. Additionally, when reporting "annotation times for the default model and the mean over annotation times after fine-tuning," it's unclear whether these times include retraining, how many images are used (e.g., the 20 with ground-truth), and whether these factors affect the measurements. Similarly, in the "tracking user study," it remains unclear how times are measured and whether TrackMate is used interactively with Cellpose or in conjunction with a segmentation output from a pretrained model from the Cellpose Zoo.

We describe the methodology of the user study in more detail now in the newly introduced methods section. We address all these points there. Note that there was a miscommunication regarding the retraining in CellPose, we indeed use the in-the-loop finetuning and retrain the model after each image, for five images. We are sorry about this oversight.

Consider revising Figure 1a for improved clarity. The current representation may be unclear in terms of where the process begins and ends. A more explicit depiction would enhance the understanding of the workflow.

We have updated Figure 1 to provide an easier to understand depiction of the workflow.

Regarding the experiments performed in this work, although the authors have conducted an extensive analysis with a very large number of datasets, their models have been specialized for LM or EM modalities exclusively. Given the manuscript's title, "Segment Anything for Microscopy," it raises questions about the potential effects of training SAM across different modalities and whether a more generalized foundation model for microscopy could be established. Experiments in that direction, maybe with tests on held-out datasets of various modalities or even on the 2018 Data Science Bowl dataset, could substantially increase the impact of this

work.

There are different ways to address this comment: we **already do** provide experiments on a wide range of microscopy datasets, including many held-out datasets that were not used for training the model in any way and including modalities that are not part of the training dataset. Supp. Fig. 2 shows all 12 datasets (some including a mixture of modalities) we have tested for light microscopy, which include two histopathology datasets (PanNuke and Lizard). We see clear improvements across all these datasets; except for ViT-T, which is a very small model and which we only recommend to use in very resource constrained settings. For the case of histopathology we see that the improvement for a single point prompt is not always clear, but all other interactive segmentation settings clearly improve. Experiments on the hold-out set of the 2018 DSB would not provide further value here, since we include the training split of this dataset in the generalist's training dataset. Similarly, we test the new EM generalist for a total of 15 datasets, see Supp. Fig. 5, including a high energy X-Ray dataset (NucMM (Mouse)) and datasets containing organelles the model was not trained on (Sponge EM, Platynereis (Cilia)). We see improvements due to our generalist model in all cases. For structures that are significantly different from mitochondria (ER, neurites), we observe that our generalist model for mitochondria and nuclei does not improve quality compared to default SAM, but finetuning on specific data can be used in such cases, see Supp. Fig. 8.

The underlying question raised is: can we build a single model that works optimally for all microscopy data modalities. The answer is currently no, especially due to the semantic ambiguity in the case of EM data (see more detailed discussions in the manuscript). However, we can provide a family of models (default SAM model, generalist LM / EM models) that share the same architecture and can be used in exactly the same way, which already offers a more unified approach to microscopy segmentation than prior deep learning based tools. We discuss this in the new limitations paragraph as well.

Regarding the minor comments: thank you for raising these points. We have taken them into account for the revised manuscript.

Regarding the documentation, I suggest the inclusion of a step-by-step tutorial designed to assist inexperienced users in navigating and using the software effectively.

Thank you for this suggestion. We now provide an introduction video (Supp. Video 1) that explains how to install the tool and how to start it after the installation.

Additionally, if feasible, consider incorporating tooltip texts for each button within the GUI. These tooltips can provide brief explanations or hints to users, further facilitating their understanding and usage of the software. Such user-friendly enhancements would contribute to a more accessible and user-oriented software experience.

Thank you for the suggestion. We have integrated tooltips into the napari plugin.

Reviewer 2: Thank you for the good summary of our contribution, the good feedback on improvements and the generally positive assessment of our work.

The order in which information is provided within the paper does sometimes lead to confusion. For example, “automatic instance segmentation” was referred to without beforehand giving an explanation of what it means.

We have addressed this in the revised manuscript and now explain the two approaches for automatic instance segmentation (AMG and AIS, the second one being new) early in the result section and also introduce the abbreviations there.

Furthermore, when referring to the metric used for evaluation of the segmentations results, an explanation is missing and the reader does not know if larger or smaller values are better.

We now give a short explanation of the metric early in the result section and have extended the descriptions of the metric in the method section.

It was mentioned that the point annotations were placed at the center of the object to mimic user behavior. However, for the example given in Figure 4 b top left, I would expect the user to put the point into the bulky part of the structure. This raises the question of how sensitive the outcomes are to different positioning of the point annotations. Do changes have a significant impact on the results?

Note that we have now updated the evaluation procedure for interactive segmentation (see beginning of the response for details). In the new scheme we now randomly sample the first point annotation, and the subsequent points are sampled in places where the model predictions were wrong (compared to ground-truth). We investigated the influence of randomness on this procedure in Supp. Fig. 1 b. Here, we run the experiment five times with different random seeds and report the standard deviation over these five runs in the bar plots. We found that the overall results were not affected much.

Could that explain why sometimes giving less point annotations is beneficial (Fig. 2 various cases, Fig. 3a Lizard Generalist, Fig. 4a CREMI Default,...)?

This behavior was due to the reliance on a mask prompt for interactive segmentation in our previous training scheme. It is now fixed and our models consistently improve segmentation results with more point prompts in almost all cases. See the discussion of this at the beginning of this response for details.

In my opinion, a more detailed explanation of the runtime evaluation given in Figure 5 would be beneficial. On which machine was this performed and is this

comparable to common user hardware? How many different users were involved when doing annotations and could this induce a subjective bias due to different levels of expertise with the different tools? What did contribute to the calculation of the runtime: for example, the finetuned micro_sam was reported to have an average annotation time of 0.3 seconds, but does this also include providing user point/box annotations? Even 3.4 seconds for manual annotations seems to be pretty fast, assuming that accurate annotations were obtained.

We have added a new methods section that explains the (updated) user study in more detail; see also the response above. Regarding the specific question on annotation time: annotation with the finetuned models (both ours and CellPose) was mainly spent on visual inspection since both models yield very good segmentation results. This explains the low annotation time in these cases. Regarding manual annotation: the organoids are of fairly round shape, which makes it possible to annotate them relatively quickly. This time would indeed be much higher for more complex shapes; and would result in an even clearer advantage for interactive and automated annotation.

While testing the software, I experienced that the pre-computation of image encodings is a must in order to effectively use the tool. This is especially crucial for 3D and timeseries data. Although computation of encodings can be done beforehand in a fully-automated fashion, I think this is an important point when comparing the different approaches and should at least be explicitly mentioned when discussing runtimes. Other approaches like Cellpose don't have this requirement and can be directly applied, which is not necessarily a (dis-)advantage, but certainly important information to interpret results.

This is true. We have now included time measurements for the different inference steps (computing image embeddings, automatic segmentation and interactive segmentation) in Fig. 5 a. We also address the comparison to other tools in terms of runtimes in the new limitations paragraph.

Figure 4 a shows results for different EM datasets. When referenced in text, interpretations are no longer associated with the datasets, but with the structures they show (cells, neurites, nuclei, mitochondria). However, without knowing these datasets, it is hard to follow this interpretation, which is why I would recommend to keep a consistent description and either use the dataset name or the structure specification.

Thank you for pointing this out. We provide a new set of EM experiments, but have kept this comment in mind for the description of the new experiments.

Figure 5 seems a bit chaotic and could be visually improved. My main concern is that the different blocks (a,b,c) range into one another, which makes it harder to separate them and get a clear view of the different results.

We have updated Figure 6 (which now shows the user study results) to give a more concise overview of the results. We have also decided to focus on the timing results and to show qualitative results for each study. We discuss the evaluation of annotation quality in the (new) user study method section.

It seems unclear if there is a difference in processing RGB image data (e.g. histopathology) and grayscale image data. Does the approach take a 3-channel image as input for all cases or is there any pre-processing involved?

The vision transformer that processes the image data in SAM requires RGB input data. Hence, for grayscale data we duplicate the single channel three times and provide it to the model. For RGB data like histopathology data we provide the data as is. Data with two channels, e.g. TissueNet, proves a bit more difficult, and we average the channels and then duplicate the resulting single channel three times. We have added a paragraph describing this in more detail to the first subsection of Methods.

Thank you for raising the minor points about the manuscript. We have taken these into account for the revised version.

It would be helpful for annotation speed, if there would be a shortcut for switching between positive and negative prompts.

There is a keyboard shortcut for this already: “t”. Thank you for pointing out that this was not sufficiently documented. We have added this to the online documentation, the (new) tooltip and the updated video for 2d annotation.

Personally, I think placing point annotations is the fastest option. However, it appears that the segmentation process necessitates an iterative approach, meaning that it seems impossible to place multiple independent point annotations to process results for multiple cells collectively and then initiate the processing in one go (unlike for box annotations). Unfortunately, this can significantly extend the runtime when using the tool in practical scenarios. For example, in an image scene with several hundred cells, it would entail a substantial number of manual interactions and clicks. Ideally, it would be beneficial to be able to place common negative prompts and independent positive prompts, even for very dense scenes of touching cells or scenes showing cellular membranes.

There are two points to address here: first, for many cells in an image it is advisable to start with automatic segmentation, remove wrongly segmented cells, and then correct the results via interactive prompting. Our automatic segmentation support has improved a lot compared to the prior version due to the new AIS methodology (much faster and better results in most cases). We have updated Supp. Video 2 to demonstrate this procedure.

We also agree that point based segmentation for multiple objects at a time is a good idea. We have added support for this via the “batched” segmentation option for 2D annotation, following

the procedure you suggest, including sharing negative prompts. This is explained in more detail in Supp. Video 2, Supp. Video 5 and in our online documentation.

Placing multiple box annotations at once before initiating the segmentation process does not seem to work for 3D image data. Unless it only was a bug, adding this feature would significantly improve the practical usability for 3D image data.

We agree that this is desirable, but it would introduce significant technical complexity to enable interactive segmentation of 3D segmentation results for multiple objects. We decided not to implement this feature for now due to a large potential for bugs and a potentially confusing experience for users. Instead, we now support automatic 3d segmentation, based on AIS. This can speed up 3D segmentation a lot, as shown in the second user study on 3D nucleus segmentation. This new feature is also shown in Supp. Video 3.

Reviewer 3: Thank you for the feedback on our tool and the positive evaluation of its potential impact. We believe that some of the negative user experience has been due to unforeseen installation issues. Based on this feedback we have thoroughly updated the tool and believe that all the major points raised have been addressed:

- a. The tool does not work well for electron microscopy data*
- b. Shortcomings inherent to the SAM limit the usability of this tool*
- c. The comparisons made to benchmark uSAM performance flatter to deceive*
- d. The installation was not easy and cannot be done by an end user*

We have updated the EM models to address a) by providing a generalist model for the segmentation of mitochondria, nuclei and other roundish organelles. We further demonstrate that finetuning yields improved models for other tasks such as neurite segmentation. In summary, these changes make the tool very useful as is.

We are not quite sure about the exact meaning behind b), but have included timing benchmarks and experiments on model training with constrained resources to provide a better overview of how the tool can be used in practical applications.

We disagree with c). We believe that the benchmarks accurately reflect the performance of the tool and especially show its benefits for interactive segmentation. Please note that many aspects of these experiments have now been updated. We also highlight in the manuscript now that we don't claim superiority to other tools for automatic segmentation, but that we provide automatic segmentation at an overall similar quality for practical purposes while also providing interactive segmentation functionality and offering a tool that addresses different microscopy modalities at once.

The installation problems observed in d) are unfortunate, but are not the norm according to our experience with users and are also not reflected in the report of the other two reviewers.

Nevertheless, we have simplified and updated the installation instructions to address this.

- a. The given anaconda installation steps are very slow for a linux environment and could not be performed without significant troubleshooting, meaning that this tool*

can only be installed by computational analysts, not by end users.

i. Various individual dependencies had to be individually installed from source (not just napari, pyqt, but also SAM and torch-em)

ii. Installation failed because of ZARR and Vigna (an odd and exotic dependency, we urge authors to find an alternative).

iii. Finally installed using Linux bash script, which is still in dev

As mentioned above we have not observed these errors ourselves, but have simplified the installation procedure and updated the instructions to address this:

i) the conda package now includes all required dependencies and no additional software has to be installed. We initially decided to not include napari and pyqt due to the fact that they are not needed to use the micro_sam python library and to provide a more lightweight dependency for developers. To simplify the installation for users of the tool we now include them and provide explanations for developers how to install the software without napari (and in the future may provide a separate package for this if interest arises).

ii) We are not quite sure why this happened and have not observed these issues ourselves. Our best guess is an old conda version. However, the fact that installation via conda takes long is known in general and is by no means restricted to our package. For this reason we now recommend using mamba for installing the package, which provides the same functionality as conda but installs the package much faster.

iii) The installer is now fully supported, but does not enable GPU support, which we clearly state in the documentation.

You can find the updated installation instructions here:

https://computational-cell-analytics.github.io/micro-sam/micro_sam.html#installation

iv. Training/finetuning should allow multi GPU support

This is on our roadmap (see <https://github.com/constantinpape/torch-em/issues/229>), but we have focused on other improvements for the resubmission. To make training on user's own data easier we now provide:

- A new GUI that comes with our napari plugin to enable training for users without any coding experience.
- A notebook that explains how to use the python library for training a model:
<https://github.com/computational-cell-analytics/micro-sam/blob/master/notebooks/micro-sam-finetuning.ipynb>
- An extensive study on finetuning with different hardware configurations, see Fig. 5 b and Supp. Fig. 12 + 13. Finetuning is possible both on a CPU (where it takes ca. 6 hours for our experiments, though this depends on the dataset due to early stopping) and a consumer-grade GPU with 16 GB VRAM or more (ca. 30 minutes). The limit of 16 GB VRAM is fulfilled by the latest high-end consumer grade GPUs (3090 and 4090 both have 24 GB), but not by a 1080Ti. For a 1080Ti we found that training is only possible with settings that lead to worse results (either freezing the image encoder or using a ViT-Tiny). In this setting training on the CPU will likely be better. Note that multi GPU support for training would not help in this case, as it does not decrease the VRAM

requirements. We plan to integrate parameter efficient finetuning based on LoRA and related approaches in the future, to enable faster training on the CPU and using GPUs with less VRAM.

b. In fact when troubleshooting, we noticed that the github repo was updated in the past days several times, meaning that this is a project still in development. We urge the authors to upload a fully stable linux/PC/mac compatible release on Zenodo as a v1. We understand that future iterations will follow, but a stable v1 is a must.

We have now released a stable v1 version of our tool, which includes all improvements described in the revised version of the manuscript and in this response.

*a. We note that the approach of “fine-tuning” a SAM to a specific domain is not novel – it has been done before. [<https://doi.org/10.48550/arXiv.2304.13785>]
b. That means that the authors’ work was to train the SAM model on a handful of LM and EM datasets, update weights, and then release an annotation tool that allows rapid segmentation based on this new uSAM model. Here are three issues with this:
b.i. We agree that hard work has been done, but from a computational point of view, there really isn’t much that’s new or novel (see Pt 2a).*

Our work is substantially different from MedSAM³ or other work that finetunes SAM for medical image data: to the best of our knowledge these approaches all only finetune the model for bounding box based segmentation. This makes the segmentation objective much simpler and improves segmentation quality only for the narrow case of segmentation with box prompts, but decreases performance in other settings, especially iterative interactive segmentation where the user corrects the segmentation masks predicted by SAM with prompts. See discussion of the training objective in the updated methods section. Furthermore, we now include finetuning a separate decoder for better instance segmentation (compared to the AMG procedure of SAM), so our work is the first contribution that combines finetuning for interactive and automatic segmentation in the same model. We have extended the introduction to cite more related work and to make these distinctions clear.

b.ii. The Achilles heel of this approach is that WYSIWYG – the “generalist uSAM” is meant to be used as-is, further retraining on LM or EM data is very slow, taking days/weeks even on A100 GPUs, and even longer on weaker systems.

Further retraining is possible on user hardware, even the CPU, in a reasonable amount of time. See the comment on this above. We agree that we did not highlight this fact enough in the first version of the manuscript and that it would have been very difficult for a user to achieve because it required detailed knowledge of training hyperparameters. We have rectified this in the revised version of the manuscript and provide a convenience function that selects an appropriate configuration for the user’s hardware, see the training notebook we already linked above for details.

³ <https://www.nature.com/articles/s41467-024-44824-z>

b.iii. Running uSAM on 3D data or stacks is extremely slow. We recorded ~ 8s per image for a 1k x 1k image, meaning that even modest image stacks require hours to load the model embeddings. Note that even though appropriate cuda libraries are installed it used CPUs.

Computing the image embeddings for 3d data on a CPU indeed takes some time. This is why we enable caching the embeddings to file so that they can be reused; note that intermediate annotation results can also be reloaded. The runtime can be reduced by choosing a model with vit_b backend (which does not lead to a big loss of segmentation) or a vit_t backend (which is much faster, but performs poorly for some data and needs to be tested on 2d data beforehand). See the updated experiments comparing all models in the manuscript. If a GPU is available the embedding computation is much faster, see also Fig. 5 a for the comparison of runtimes for different operations. Note that the preprocessing time is purely “passive”, so even with a CPU SAM can be used for fast interactive annotation since the embedding computation can run “unattended” in the background. In your case the fact that only the CPU was used is due to the installation (see answer above).

c. The SAM approach requires that the user clicks on various features to get a segmentation. Yes, this makes segmentation easy, but it also means that for n instances in m images, the user will have to click $n \times m$ times minimum. This is a burden which is not required for models such as CellPose.

We provide automatic segmentation support in the 2D and 3D annotator now, so this statement is not true for the revised versions. For both tools it is now possible to first compute an automated segmentation, correct or remove wrongly segmented objects with napari, and then to interactively segment the remaining objects. Note that we already provided this feature for 2D segmentation in the previous version of the manuscript using AMG, but now provide automatic segmentation via AIS (based on an additional decoder), which is much faster and provides better quality in many cases. We use it for 3D segmentation by segmenting the volume slice by slice and then merging the segmentation across slices.

d. There's another issue as far as we could tell – for multiple instances of the same class (say n cells in an image) clicking multiple points does not work, you have to box out instances. And this fails when you have convoluted structures where one cell “envelops” another, so the boxes completely overlap.

We don't fully understand this comment. We'd like to point out that segmentation with multiple points has improved with our new models due to the improvements in the training methodology mentioned in the beginning of the response. We now also support annotation with polygons (see Supp. Video 2 for how to use these), which may help to segment objects with a complex shape.

e. Overall, unfortunately the time and compute power required to finetune the uSAM models is such a massive stumbling block (the authors should clarify exact numbers

on resources/time needed for fine-tuning to the point of getting a useful model), it just renders this approach pretty inflexible.

This statement is not correct, finetuning is possible with standard resources, see the prior answers on this. We agree that the corresponding experiments were missing and that the corresponding documentation was previously not sufficient, but have rectified this.

Note that we have also added a short discussion of runtimes in comparison to other tools in the limitations section, where we highlight the disadvantages compared to CNN based tools. (Which are present but not as dramatic).

a. Some plots of the model training must be included (loss over iterations, hyperparameter grid search, accuracy over iterations etc). Otherwise it appears as if the authors simply chose a set of parameters and went with it.

We have included additional figures to report on this: Supp. Fig. 12 a shows the evaluation measures over training iterations, Supp Fig 12 b. Shows the influence of the most important training hyperparameter, the number of objects sampled per image per batch. We have also extended the methods section on training where we describe additional experiments, including for choosing learning rate and optimizer, comparing different training schemes for interactive and automatic segmentation and initial experiments on the interactive segmentation objective.

b. The quality of the segmentation output (and if there was manual cleanup required) must be visually compared with other models such as CellPose and StarDist.

We now provide quantitative comparison of automatic segmentation results in Fig. 5 c and Supp. Fig. 11. We have revamped the figure on the user study (Fig. 6) and also show qualitative results for all three datasets there.

c. The Y axis must be set at 1.0 across all graphs, otherwise it is misleading

We disagree with this point. Showing the Y-axis set to 1.0 across all graphs would make the plots less legible due to increased whitespace and setting it not 1.0 is not misleading. We compare across a set of very different datasets with segmentation problems of different difficulty using a quite stringent metric. This means that results cannot be compared across datasets, which would be the main reason to set the Y-axis to 1.0. We make sure to set the same limit for the Y-axis for experiments on the same datasets in our plots.

d. Which CellPose model was used for comparison? Since there is for example CellPose for TissueNet, why aren't the authors using that to compare?

This is updated in the revised version, we compare to the CellPose specialist models for LIVECell and TissueNet and cyto2 otherwise.

e. The authors should re-label “segmentation quality” to “mean segmentation accuracy” (if that’s what it is)

We have done this and also explain the metrics used in more detail in the methods section.

f. What does “in-loop finetuning” mean, is it about finetuning SAM model with user annotation? because we can’t see any finetuning implementation in the annotator GUI.

We have removed “in-the-loop finetuning” when referring to our tool because we agree that this is not quite accurate. We do provide a GUI based tool for finetuning now, see details above.

Figure 3 shows that uSAM can beat CellPose at a magic number of 12 prompts ($8n + 4p$) but very strangely the specialist model does not perform better than the generalist in some cases (why?).

The experiments are updated now, our models have improved due to the updated training procedure and the evaluation procedure was also updated to mimic iterative interactive annotation better. See explanations at the start of the response. We would also like to point out that comparing CellPose (automatic segmentation) to interactive segmentation (based on prompts) is not very meaningful and now make this clear in the manuscript.

Further, only the LiveCELL is a true apples-to-apples comparison of models’ strength. But again, CellPose is then good to go out of the box, whereas uSAM still needs all those user prompts or seeds for annotating. And fine-tuning is onerous for uSAM (and again even then you need the seeds!) Finally, when y axes are set to 1.0, the results are revealed to be modest overall.

We now compare both automated (AIS, AMG) and interactive segmentation with uSAM with two methods for automated segmentation (CellPose for LM, MitoNet for mitochondria in EM). We agree that the distinction of automatic versus interactive segmentation is important and clearly explain this in the updated manuscript. Thanks to our new instance segmentation contribution (AIS), the automatic instance segmentation capabilities of our tool are much improved compared to earlier. To be clear, we again want to draw attention to the fact here that we do not claim to be superior to CellPose or MitoNet for **automatic instance segmentation**, but rather that we provide results of comparable quality in most practical settings thanks to AIS, combined with the **interactive segmentation functionality** of SAM, which can speed up data annotation massively. We comment on this both in the results section and the new limitations section at the end of the discussion.

Figure 4 reveals the weakness of uSAM with EM data – it basically does not generalize (and automated annotation amg is SQ=0?)

We have completely updated the EM contribution, provide a generalist model for mitochondria, nuclei and other roundish organelles. We demonstrate that this model improves the segmentation performance for the respective tasks and study how finetuning for other segmentation tasks improves results. See details in the responses above.

Figure 5. The CellPose numbers are missing from the graph, and the 3D results must be compared also (e.g., against StarDist) In all cases, the images presented appear to be “best case” segmentation, and we got far worse results in our hands unfortunately. 4. The “generalist model” doesn’t work well for EM! To be fair, the authors are modest about their claims with EM data, but even for a basic task – segmenting neurites and cells from their own shared dataset – uSAM failed badly on stacks of images <<see attached >>.

We have updated the user study now and focus on the annotation times and the qualitative results in the Result section. We discuss the quantitative results in more detail in the corresponding methods section.

a. One limitation is that for volume EM datasets, running inference purely in the imaging plane gives the well-known “pancake artifact”. We suggest the authors to run inference in all three planes, which should improve results <<REF Conrad and Narayan 2023 <https://doi.org/10.1017/S143192762002053X> >> and they should show results.

Running inference on all three planes would significantly increase runtimes so this approach cannot be implemented in the current context. Ultimately we believe that a full 3D model is the best solution for this approach. We plan to include this in future work but it is out of scope for this contribution. Another alternative to running inference in three planes is to rectify resulting artifacts via post-processing. We already do this for automatic 3D segmentation and use a closing operation to avoid gaps due to missing or incomplete segmentations in a few slices. Similarly, boundary placement in 3D could be rectified with a local edge filter. We had good experiences with this in other contexts and may look into adding this as an additional post-processing feature to alleviate the artifact until a full 3D model is available. We did not have time to look into this yet due to the significant work that went into improving the methodology and tool for the revised version. For now we address this point in the limitation paragraph.

Overall, this is a commendable effort, but either the authors must drop claims of electron microscopy and just release the LM uSAM (in which case the model does not reliably perform favorably with CellPose), or alternatively, the authors must train extensively on EM data to get a better performing uSAM for EM data.

We strongly believe to have addressed these points with the new models and updated experiments. Regarding the comparison to CellPose we again want to point out that we are not claiming to provide a better tool for automatic segmentation (though the automated segmentation quality is not very different for many practical purposes according to our

experiments), but rather that we provide a more versatile tool that also supports interactive segmentation.

Dear editor and reviewers,

We have now submitted a revised version of our “Segment Anything for Microscopy” manuscript. In this accompanying letter we address questions raised by the editor and reviewers after our last submission, summarize additional experiments and changes to the manuscript and provide a point by point response to the reviewer's remarks. We have highlighted all relevant changes in the resubmitted manuscript that were made to address these points in red.

First, we would like to address two important questions raised by the editor, which also stem from points raised by reviewer 1:

(1) Where does μ SAM fit into the toolspace and who do you think its users will be?

μ SAM addresses both interactive and automatic instance segmentation, with the main distinguishing aspect compared to other microscopy segmentation tools that interactive segmentation is performed by the deep learning model itself. For example, ilastik supports several options for interactive annotation, but none of these are directly tied to a deep learning model, leading to a diminished performance for difficult segmentation problems. We clearly see this in our comparison to the carving workflow, which is used frequently for 3D annotation, where our approach is faster and provides better results. The existing deep learning based tools do not provide any model-based interactive annotation. For example, CellPose “human-in-the-loop” relies on manual correction of model predictions and retraining. This is a very good approach if the initial segmentation quality is satisfactory and annotation times for individual objects are not too high. For (2D) cell segmentation this is often the case, however there are still many applications, from 3D cell segmentation to EM organelle segmentation to other modalities like X-Ray, where this does not hold true. For these cases, generating segmentation annotations (either for training a segmentation model or direct analysis) takes very long and often even prohibits quantitative analysis. In these cases μ SAM can be transformative, as the combination of interactive and automatic segmentation functionality speeds up the data annotation and processing dramatically. Consequently, some of the use-cases that we envision, and are increasingly also seeing in practice, are:

- Data annotation and (semi-)automatic segmentation for 3D instance segmentation, both in LM, EM and other modalities. This may include finetuning a custom SAM model for the given data to further speed up annotation or to implement high-quality automatic segmentation.
- Annotation and segmentation of diverse structures, for example organelles in EM. Due to the high diversity a specific pretrained model is often not available for these applications, resulting in challenges in quantitative analysis alleviated by μ SAM.
- Cell segmentation (both interactive and automatic) in LM. μ SAM is a very useful tool in this space and we believe is competitive with existing tools.

In our opinion, μ SAM is transformative for the first two applications, where there is an urgent lack of existing tools. The main limitations we see compared to established tools are in the runtime and scalability. We would like to stress that this still does not preclude automatic processing with μ SAM, it mainly leads to longer times for preprocessing for automated prediction. An exception may be large volumetric data, but here our tool can be very valuable in generating training data for a more scalable automatic segmentation approach.

To give concrete examples, we have seen external uptake for the following applications:

- Segmentation of chromatin in volume EM using interactive volumetric segmentation.
- Segmentation of chloroplasts in plankton in 2D TEM images using automatic segmentation with some interactive corrections.
- Data annotation for ground-truth generation in several modalities in an image analysis facility.

In all these cases the users had previously tried other tools that could not solve their tasks, were able to use our tool without initial help and only contacted us later regarding specific optimizations. I want to stress that neither the list of envisioned applications nor of specific user examples are exhaustive, but should rather indicate typical applications. We have adapted the main section of the manuscript to better highlight some of these aspects.

(2) Will the tool be easy/fast enough for users to benefit from practically?

Yes, we believe that this is the case, as we have seen some external adoption already (see above). In more detail, we currently see three relevant points for usability:

- Installation of the software. Using conda/mamba can be challenging for life scientists, but is currently needed for the tool's full functionality. To alleviate these issues we provide an extensive FAQ and the standalone installers.
- Software usage. The complexity is inherently higher compared to tools that do not support interactive annotation since this additional functionality requires new UI elements. To address this we provide an extensive FAQ and the tutorial videos.
- Computation times. These are higher than other tools, especially on a CPU. We implement embedding precomputation to alleviate some of these issues. The precomputation step is passive, so that other work can be conducted in that time. The usage is interactive afterwards. Note also that other tools suffer from similar problems. E.g. precomputation with carving for a 3D volume takes several minutes (comparable to precomputing image embeddings for a 3D volume on a CPU). In-the-loop finetuning with CellPose also takes several minutes **after each image**, resulting in frequent interruptions of the annotation process.

To address the questions raised by the reviewers we have performed additional experiments and extended the manuscript to report on their results:

- We have redone the 2D segmentation user study with multiple annotators and an updated experiment design. To better understand different aspects of how μ SAM compares to CellPose for this data we now report average annotation times, annotation quality compared to consensus annotations and generalization performance measured on a separate test split. Overall, we find that μ SAM is competitive with CellPose for these experiments, with advantages for CellPose in initial segmentation quality but disadvantages in generalization of finetuned models.
- We have performed an additional experiment that compares automatic segmentation using our distance-based approach (AIS) for different SAM-derived architectures and a U-Net. This experiment shows that using our UNETR architecture outperforms other approaches for small training datasets, with only marginal differences for large datasets.
- We perform an initial exploration of LoRA-based parameter-efficient training. Here, we find that the segmentation quality is only marginally affected, but do not see any

substantial efficiency gains and on average even longer training times. We report on these results, but do not integrate LoRA further with our tool due to our findings.

- We perform initial experiments on automated tracking for the user study that show that a tracking result of good quality can be obtained based on automatic segmentation with μ SAM. This prepares implementation of automated tracking in our tool, which is pending broader availability of third-party software.

Below you can find a detailed description of these experiments and detailed answers to the other points raised by the reviewers

Answer to reviewer 1 (R1):

> R1: I believe the new version of both the manuscript and its associated software application (and documentation) have improved much since its initial submission of this work. In particular, the new version of microSAM contains a new option for automatic instance segmentation (AIS) that requires fine-tuning on annotated data and everything is now implemented as a napari plugin, facilitating its use [...]. Nevertheless, I still have some concerns about a few of the points I raised:

Thank you for the overall positive assessment of our contribution and the confirmation that our first revision improved the manuscript and the tool. We are confident that our second revision now addresses the remaining concerns.

> R1: 1. In the evaluation of the annotation tools, I believe it would be very positive for the paper to include (1) the average annotation times and their standard deviation of several annotators at least for a few images of the 2D annotation user study (they are only 20 in total), (2) the rest of important times on each experiment (embedding calculation, retraining/fine-tuning) and (3) the achieved segmentation accuracy in each experiment. This information is crucial for the potential users to decide on which tool to use.

We have redone the user-study to address these points. The experiments are now performed by five different annotators and we report the average annotation times, the quality of the annotations compared against a consensus annotation and the generalization on a separate test split with ground-truth annotations. Please note that we have a few important observations in this experiment: (i) interactive segmentation with default SAM works better for this dataset compared to the LM generalist, because the LM generalist enlarges the segmentation masks too much. We have not observed this phenomenon for other data, so it is likely dataset specific. We aim to provide an improved generalist model to address this in the future. (ii) CellPose in the loop training is a bit unstable: the model sometimes overfits leading to bad segmentations for the next image. In these cases we use the initial *cyto2* model to segment the next image. (iii) the finetuned CellPose models show a worse generalization performance on the test set compared to *cyto2*. In contrast, the microSAM models improve on the test set after finetuning. We don't have a good explanation for this fact, but have double checked the training and inference code for CellPose several times to ensure that this is not due to a bug. Qualitatively we find that the finetuned CellPose models miss organoids with somewhat different morphologies than in the training set. Note that the initial automatic segmentation performance of *cyto2* is better compared to the LM generalist. We report the additional timing information (2) in the

methods section. In summary, we are convinced that this study now provides a more exhaustive overview of the practical aspects of model application, shows that microSAM is competitive with CellPose for this application and highlights different practical concerns for both tools.

> R1: 2. In the information about the datasets used (Supplementary Tables 1 and 2) I would include their size in total number of GB but also their image dimensions. This will help the readers understand the scale of the experiments made.

Thank you for this suggestion. We agree that this information is helpful and have added the following data to the supplementary tables: the size of the dataset in GB, the number of samples in the dataset (either images or volumes) and the average dimensions of a sample in the dataset.

> R1: 3. Regarding my previous comments about the specialization of microSAM on LM and EM modalities, I didn't mean the number of datasets wasn't large enough. I was pointing out that the very first impression a reader gets when seeing the title of the manuscript (Segment Anything for Microscopy), or at least that was my case, is that we will be presented with a foundation model for segmentation of microscopy images. However, the paper jumps directly into creating generalist or specialist models per image modality.

Creating a segmentation foundation model for all of microscopy is currently not possible in our context given the Segment Anything architecture. This is due to the high diversity of microscopy modalities and our goal to provide a model that addresses interactive and automatic instance segmentation. SAM does not have a semantic awareness, e.g. to differentiate cellular structures and organelles in EM, which would be required to provide such a unified model. This fact is apparent from the electron microscopy experiments. This point is addressed both in the EM experiment section ("An electron microscopy model improves segmentation for mitochondria and nuclei") and the Limitations section. To further clarify it, we have now added two sentences to the start of the Results section. Note that we don't consider this limitation to be significant from a practical point of view: a user would always know if they deal with EM or LM data.

> R1: Following that line of thought, one has to wonder if it is possible to have a generalist model for different modalities, but there is no experiment in that direction.

We certainly believe that this would be possible, given that our current experiments prove this for "n=2" (EM and LM). However, assembling a large and diverse enough dataset for another modality is a time consuming task, so we consider this to be out-of-scope for our current contribution. Note that we already address models for other modalities as possible future work in the conclusion; we have now slightly extended the corresponding sentence.

> R1: My suggestion about using the test set of the DSB dataset was double: on the one hand, it's a very diverse dataset in terms of modalities, and, on the other hand, it would allow direct comparison with state-of-the-art methods applied in the same dataset.

Thank you for the clarification. As stated before, DSB (without histopathology data) is used as part of the training set of our generalist model. Hence we cannot use it for an out-of-domain validation experiment without changing the training dataset and retraining the generalist. Given the already large diversity of our test datasets we opted not to perform this experiment.

> R1: Related to this last comment, I do believe the manuscript would benefit from better defining its objectives and reach. As I see it, microSAM is a powerful software application that exploits SAM to create specific solutions for the instance segmentation and tracking of microscopy images. While this could seem obvious, along the text, microSAM is refer to in a sometimes ambiguous way such as "a tool for interactive and automatic segmentation and tracking" (singular), "tools for interactive and automatic segmentation and tracking" (in plural), "a napari plugin for interactive annotation and automatic segmentation", an improvement of "SAM for microscopy data [that] implements data annotation tools", an extension of "SAM for more efficient automatic instance segmentation", etc. I think this is understandable after so many changes in the manuscript and the application itself, but I'm sure the readers would appreciate it if this aspect is clearly stated. Maybe at the beginning you can briefly summarize the structure of your software, with its backend (accessible library) and frontend (GUI in the form of a napari plugin).

Thank you for raising this point. We agree that the previous manuscript was too ambiguous in the description of our tool. We have addressed this by rewriting the contribution section and unifying how we refer to our contribution throughout the manuscript. In particular we now use the singular "tool". When referring to individual components of the tool we use the term "widget", which matches the corresponding napari nomenclature.

> R1: * I would rewrite the abstract taking my comments above into account. There is some repetition and ambiguity (tool vs tools) about the definition of microSAM itself.

We have rewritten the abstract to remove the ambiguity between tool and tools as well as to avoid repetition. We have not done further significant changes, as we believe that the abstract correctly summarizes our contribution.

> R1: * Figure 1a is a bit confusing, since we don't see very well where the model is. Also, the caption itself talks about different models, a napari plugin and speed ups, but none of that is depicted in the figure. I really like the logo though!

We have updated the figure to better separate model and workflow and have updated the caption to better describe the typical workflow in our tool and better reference elements of the figure..

> R1: * In the part where you describe your contributions, I would rewrite the text to clearly state (1) if the training method you implement is novel, (2) that you extend the *architecture* of SAM with a new encoder, and (3) that your software application includes a frontend in the form of a napari plugin and pretrained models ready to use. Again, please refer to my comment above about properly defining microSAM and its reach.

We have rewritten this part of the manuscript following your suggestions..

> R1: * From the results you presented, one would conclude that not using the previous mask prediction as an additional prompt to the model is an important limiting factor for the segmentation results of the interactive segmentation method (Figure 2a, left vs right). I understand that keeping track of the previous mask of each annotated object could be an implementation challenge, but could it be less of a hurdle than fine-tuning the whole model? The former could be seamlessly executed in CPU, while the latter requires a GPU to be used in a real scenario.

We have already addressed this in a supplementary experiment that was part of the last submission, see Suppl. Fig. 10. In summary: while providing the previous predicted mask as prompt does improve the interactive segmentation performance of the default SAM model, the finetuned model still performs clearly better in this setting. So the value of finetuning goes far beyond just making the interactive segmentation independent of the mask prompt, both by improving interactive segmentation and providing better automatic segmentation through the new instance segmentation decoder. Nevertheless, it would be desirable to enable tracking the previous mask prompt in our napari plugin. This is however not trivial, due to our functionality for segmenting multiple objects with prompts at a time. The book-keeping required for this functionality would become more complicated if mask prompts are also used and would need to be implemented and tested carefully. So this is currently out-of-scope for our revision work, but we will try to implement this in the future.

> R1: * In all the experiments using fine-tuned models (generalists or specialists) in Figures 2, 3 and 4, it's unclear what prompts are used to extract the objects under the I_P and I_B frameworks. I understand the training was performed with 7 iterations of point/box annotations but, is it also the case for testing? Or a single prompt is used in that case?

We follow a similar procedure to training when evaluating a model for interactive segmentation: we derive an initial prompt (either a point or a box) from the ground-truth segmentation mask, followed by iteratively sampling point prompts where the mask predicted by the model contains errors. In each iteration we add a positive point where the predicted mask does not cover the true object and a negative point where it “spills over” the true object. The initial prompt (point or mask) and the point prompts from previous iterations are kept. We found that this procedure best evaluates the interactive segmentation capability of the model by mimicking iterative corrections of model predictions by the user. We perform seven of these iterations after the initial prompt. Fig. 2 a, Supp Fig. 1 b and Supp Fig. 10 show the evaluation results over all iterations. For Fig. 2 b, Fig. 2 c and other figures that report interactive segmentation results we perform the same experiment, but report only a subset of the results. Here, I_B denotes the result for the last iteration when starting from an initial box prompt (corresponding to the red bar at iteration 7 in Fig. 2 a) and I_P denotes the result for the last iteration when starting from an initial point prompt (corresponding to the green bar at iteration 7 in Fig. 2 a). We also report the results for the initial point / box prompt (“Point” / “Box”). We have made this reporting choice for interactive segmentation to provide a concise overview. We have added some explanation to further clarify the evaluation

procedure to the corresponding part of the results section.

> R1: * Regarding the AIS method proposed here, which is frequently the best performing method among the automatic ones presented in the paper, one has to wonder if a lighter U-Net-like convolutional model predicting the same channels wouldn't perform in an equivalent or better way when fine-tuned in the same images. This is an important question for a potential user interested in using (or not) microSAM for the automatic instance segmentation of their data.

We have added a new experiment to address this point, training both our UNETR model (which corresponds to the SAM image encoder and an additional convolutional decoder), another architecture derived from SAM and a U-Net to predict the foreground and distance channels for instance segmentation. We perform this experiment both on LIVECell and Covid-IF to investigate how the performance differs with a large and a small training dataset respectively. The results of this experiment are shown in the new Supp. Fig. 14 a) and b). In summary, we see quite similar segmentation quality when training on a large dataset, but clear advantages for the methods that use a SAM encoder with initialized weights given small training data. This observation may not be surprising given that we do not have pretrained weights for the UNet encoder. However, as a consequence weight initialization is crucial for applications like user-based finetuning on small datasets. Performing a truly fair experiment with a UNet encoder pre-trained on the SAM segmentation dataset is out-of-scope for our work, as it is unclear how to exactly pre-train this encoder and as this would require significant computational resources. Note that the current experiments show the advantage of our approach in practice.

> R1: Minor comments: [...]

Thank you for raising these points, we have taken them into account while revising the manuscript. Below are a few more clarifications for specific points:

> R1: * In Figure 2a: One would expect to see a line too in the right plot for a fine-tuned version of Cellpose, the same way AIS is a somehow fine-tuned version of AMG (with the new encoder).

We use the CellPose model trained on LIVECell, which is provided already with the CellPose software, for all experiments on LIVECell. So both the lines in Fig. 2 a for CellPose correspond to a finetuned CellPose model. This was already stated in the text and we have now added this information to the caption as well.

> R1: * In Figure 2a: The caption implies the right plot "shows the mean segmentation accuracy for interactive segmentation" but, if I understood correctly from the text, the right plot shows the accuracy results at each iteration of the fine-tuning process. This is different from interactive segmentation since previous masks are reused on each iteration, clearly benefiting the final result. This aspect should be clearly stated. Moreover, it's not clear if the results associated with that plot come from the training set or a held-out test set.

We do not quite understand the first part of this comment. The results in Fig. 2 a are

computed for the default SAM model and the model we have finetuned on LIVECell. Our answer to “**In all the experiments using fine-tuned models [...]**” given above explains the evaluation procedure for interactive segmentation that is used here. Regarding the second part of this remark: all results are obtained on the test set defined in the LIVECell publication, which is fully distinct from the training set. We indeed had missed to specify this and have added a sentence to the main text and caption.

> R1: * Page 9: The use of the dataset NucMM (X-Ray images) seems quite surprising given the section is about EM. Maybe justify its inclusion here.

This dataset shares image characteristics with EM, so it makes sense to include it here as data from a related domain.

> R1: * Page 9: Both datasets MitoEM and NucMM contain official test partitions in their corresponding challenges. If the evaluation is performed on those sets, results could be submitted to each official challenge so they can also be compared with the current state-of-the-art specialized methods.

The evaluations in this section are only done in 2D, as stated in the caption and the text. These challenges are for 3D segmentation so we cannot directly submit the results. The main focus here is on comparing the improvements of generalist to the default SAM model and showing MitoNet as a reference tool. Given the large effort we already had to invest for experiments to address the other requests for this revision we did not have the time to look into applying our 3D segmentation approach to these datasets and submitting results.

> R1: * Page 11: The text about using micro SAM in resource constraints is somehow confusing. First, it says that you "compare the inference times for all relevant operations: computing image embeddings, inference for one object with a box or point annotation and automatic segmentation via AMG and AIS, for CPU and GPU" (related to Figure 5a), and then you say that you "use pre-computed image embeddings for Point, Box, AMG and AIS measurements". Point and Box measurements are in the same plot and I understand the embeddings computation time is not included. However, the "Time per Image" plot compares Embeddings, AMG and AIS times as if they were independent, although AMG and AIS do need the embeddings calculation in order to work.

We have reformulated the text to describe the procedure more clearly. We have chosen to not include the image embedding computation in time per image as our tool also makes use of the pre-computed embeddings for the instance segmentation steps and to better show efficiency gains through AIS. Note that we do take the embedding computation into account when comparing runtimes to CellPose in the text.

> R1: Also, the way time is measured for these three options (Page 13) seems a bit arbitrary (the minimum value of the average over 10 images after 5 repetitions). Showing the average time values with the standard deviations would be much more informative.

Minimum time measurements are often used to determine the “true” runtimes as differences in execution are due to background load. The deviations for the runtimes are small.

> R1: In summary, as I mentioned at the beginning of my review, I believe the manuscript and its associated software have really improved from their previous versions. However, apart from providing more information about the different results and addressing the questions I pointed out, I really think the paper would benefit from (1) a better definition of the software itself and (2) a clear message of its immediate application and reach. In my opinion, as it is right now, microSAM is a very useful tool for semi-automatic/interactive annotation, but it's difficult to see it competing with the current state-of-the-art models for fully automatic instance segmentation. Notice I'm not saying it has to compete with them, but it would be better to emphasize that aspect in the text. Moreover, it needs a bit of work to make it faster and more user-friendly so it gets accepted by the large community of non-expert users in need of such methods.

Points (1) and (2) are now addressed in the manuscript as we have improved the description of the tool based on your overall feedback and further highlighted μ SAM's distinguishing features, especially the data annotation aspects. Regarding the comparison to tools for automatic segmentation, we see our main disadvantage in the runtimes and scalability, but see our tool competitive in segmentation quality for most cases. Nevertheless, we agree that we don't see a clear advantage of using μ SAM over CellPose for many cell segmentation tasks, but want to highlight its application to a more diverse set of tasks. With respect to the user-friendliness we want to highlight our efforts to provide good documentation, tutorial videos and FAQs. We have already seen pick-up by external users and are providing support on image.sc and GitHub. We will use feedback from these interactions to improve the tool, but conclude that it is being used by the community in the current state. Please also see the beginning of this letter for a more detailed answer pertaining to the scope of our contribution and usability of our tool. Finally, we strongly believe that our tool and manuscript are ready for publication in the current state.

Answers to reviewer 2 (R2):

> R2: I thank the authors for responding to all of my remarks and improving the code repository. I am happy to recommend publication.

Thank you for this positive feedback!

Answers to reviewer 3 (R3):

> R3: The authors present Segment Anything for Microscopy (μ SAM), a tool for interactive and automatic segmentation and tracking of objects in multi-dimensional microscopy data. [...]. The authors acknowledge the limitations of their approach, such as the larger computational footprint compared to CNN-based methods and the lack of a single model for all microscopy domains.

Thank you for this positive and correct evaluation of our contribution.

> R3: Suggested Improvements:

Exploring more efficient architectures or parameter-efficient finetuning techniques could further reduce the computational cost and enable real-time interactive segmentation on consumer hardware.

We have now performed initial exploratory experiments of parameter-efficient training with LoRA and have added the corresponding results to our figures where applicable:

- Fig. 5 b and Supp. Fig. 13 now also contain training curves with LoRA (dashed lines).
- Supp. Fig. 12 d now also contains training times for LoRA.
- Supp. Fig. 14 c shows results for a new study that evaluates the influence of the LoRA rank.

In summary, we find that segmentation quality is only slightly diminished when training with LoRA. However, the training times with LoRA are longer in most cases (see Supp. Fig. 12 d) due to more iterations / epochs needed until early stopping. Hence, using LoRA as default option for training in our tool does not currently make sense. We plan to look into more advanced methods for parameter-efficient finetuning in the future, but believe that this is not in scope for the current publication, as parameter-efficient finetuning for architectures based on vision transformers is an active and fast evolving field of research.

> R3: Investigating semantically aware models or training procedures to handle ambiguous segmentation cases in EM and enable a unified model for multiple domains could be a valuable future direction.

Adding semantic awareness to the Segment Anything Model would require fundamental changes to the architecture as well as a suitable dataset for training on linked pairs of microscopy images and text annotations. Such a contribution would require significant additional research and dataset building and is thus out of scope for the current publication.

> R3: Implementing automatic tracking based on the initial frame-by-frame segmentations from AIS could further improve the tracking annotation tool's efficiency.

We have performed an initial experiment on this, and implemented automated tracking based on an initial frame-by-frame segmentation from AIS with motile (<https://github.com/funkelab/motile>). Motile is, to the best of our knowledge, the most advanced and configurable tracking software available in python. Our implementation proceeds by computing overlaps between objects implemented in adjacent frames and using these overlaps to compute costs for a graph-based tracking problem, which is solved by motile. We evaluate this approach on the data from the tracking user study, based on AIS segmentation results from the ViT-B LM Generalist model and provide the other results from the user study for reference. The results are measured with the TRA score from CTC (see manuscript for details):

- **Automatic Tracking (AIS μ SAM LM Generalist): 0.959**
- “ μ SAM (Default)”: 0.984
- “ μ SAM (Generalist)”: 0.982
- “ μ SAM (Finetuned)”: 0.974
- “TrackMate (Stardist)”: 0.950

Here, the new automated tracking result comes from a fully automated approach, whereas all other approaches result either from interactive segmentation (other μ SAM approaches) or from automated tracking followed by manual correction (TrackMate). Despite this fact, the automated tracking result is already better than the TrackMate result after correction. Note that more extensive experiments would be necessary to make more conclusive claims about our tracking method, but the results here indicate that this approach is very promising for integrating an automated tracking method into the μ SAM tool. Some additional work is necessary to finalize this integration. First of all, some work on the GUI and user interaction is necessary to enable correction of the automated tracking results. More importantly, motile, which underlies our implementation, can currently not be installed purely via conda and is thus not compatible with our tool's installation in a stable manner. We are working with the developers of the tool to overcome this limitation and will integrate the automated tracking solution as soon as this is done.

Dear editor and reviewers,

We would like to express our gratitude for the positive evaluation of our manuscript and the opportunity to publish it in Nature Methods. We have now prepared a revised submission based on the last feedback from reviewers and the editorial guidance on preparing the manuscript for submission. We have lightly edited our manuscript to address the remaining comments from reviewers and adhere to the editorial standards. Details on the edits in response to editorial guidance are explained in the second column of the editorial guidance document.

We wish to participate in transparent peer review, assuming this applies to our paper.

In the remainder of this cover letter we comment on the remaining reviewer remarks, including how we have addressed these in the final edits:

Reviewer #1:

**** Regarding the DSB Nuclei dataset, I had assumed that the training was carried out exclusively on the training and validation partitions, which is why I suggested providing results from the test partition. Additionally, the number of images listed in Supplementary Table 1 for that dataset (497) does not match the "official" count from the BBBC (<https://bbbc.broadinstitute.org/BBBC038>). Please clarify this discrepancy in the text.***

We use a version of the dataset that excludes histopathology data and only includes fluorescence data. This version of the data is provided by StarDist, <https://github.com/stardist/stardist>. This version of the data only provides a train and validation split, which we use accordingly for training the generalist. We have added a comment about this in the text.

**** I agree with the new terminology ("tool" and "widgets") for describing your software. However, there are still some places in the manuscript where corrections were missed (e.g., on page 3 it still reads "In summary, our tools [...]"). Please review the text to ensure consistency throughout.***

Thank you for pointing this out. We have corrected the remaining occurrences of *tools* and now refer to our tool in a consistent manner.

**** I appreciate the improvements to Figure 1, but you might want to adjust the placement of the "Napari Plugin" box to center it for better clarity.***

We have centered the boxes in Figure 1.

**** Concerning the impact of using the previous predicted mask as a prompt for subsequent predictions, I understand from your results (Supplementary Figure 1) that the***

effect is negligible in a fine-tuning context compared to an interactive context with a frozen model. My earlier comment about the caption of Figure 2a (apologies for the lack of clarity) was intended to address the separation between the effect of adding more prompts (which I see as interactive segmentation) and fine-tuning. As presented, you seem to perform both fine-tuning and adding more points ("iterations") simultaneously. I suggest clarifying this distinction, as I would expect a fine-tuned but frozen model to improve with additional points, especially when using masks.

The results in Fig 2a compare the effect of interactive segmentation (adding more prompts) between the default SAM model (left) and the finetuned model (right). We believe that this is clear from the caption and did not further change it.

**** I remain unconvinced about including NucMM in the EM section, but at the very least, please provide justification for this inclusion in the manuscript, as you did in the rebuttal.***

We have added a comment in the text.

**** Finally, in Figure 6b and 6c, the evaluation metrics for each method are missing. Please add these for completeness.***

We do not have evaluation metrics in the case of 6b, as this would require an independent volume with ground-truth that we do not have for this dataset. Creating this would take a significant amount of time for this 3D dataset and starting from our annotations and using a corrected version would introduce a bias. For 6c we have added the tracking metric to the figure.

Reviewer #2:

1. The number of images used in the user study with different annotators is somewhat limited, which offers a rough estimate for practical scenarios rather than a comprehensive evaluation. Nevertheless, I believe the results should be published as they are, given the clear effort and valuable insights provided. For future research with the tool, it could be worthwhile to consider expanding this aspect to provide an even more robust analysis.

We agree that more images would provide a better insight, but would like to point out that given the different annotation variants and cycles the user study already required significant effort. That being said, we would welcome an effort to benchmark different tools for interactive / user-based segmentation tasks in a more extensive manner. Ideally such an effort should be undertaken by a team independent of tool developers or in a collaborative effort between multiple tool developers.

2. The inclusion of dimensional information in Supplementary Tables 1 and 2 is a good addition. However, I would suggest providing clarification on the order of the dimensions listed. I assume the order is (z, y, x), but explicit confirmation would be helpful.

We have updated the column headings in the table to make the axis order explicit.

3. The hyperlink provided for “user study v3” did not work for me. Unless it was planned to only make it publicly available upon acceptance, this should be fixed.

We forgot to make this repository public. We are sorry about this oversight. It is public now.